# PROVABLE BENEFIT OF CURRICULUM IN TRANSFORMER TREE-REASONING POST-TRAINING

## ABSTRACT

Recent curriculum techniques in the post-training stage of LLMs have been widely observed to outperform non-curriculum approaches in enhancing reasoning performance, yet a principled understanding of why and to what extent they work remains elusive. To address this gap, we develop a theoretical framework grounded in the intuition that progressively learning through manageable steps is more efficient than directly tackling a hard reasoning task, provided each stage stays within the model's effective competence. Under mild complexity conditions linking consecutive curriculum stages, we show that curriculum post-training avoids the exponential complexity bottleneck. To substantiate this result, drawing insights from the Chain-of-Thoughts (CoTs) solving mathematical problems such as Countdown and parity, we model CoT generation as a states-conditioned autoregressive reasoning tree, define a uniform-branching base model to capture pretrained behavior, and formalize curriculum stages as either depth-increasing (longer reasoning chains) or hint-decreasing (shorter prefixes) subtasks. Our analysis shows that, under outcome-only reward signals, reinforcement learning fine-tuning achieve high accuracy with polynomial sample complexity, whereas direct learning suffers from an exponential bottleneck. We further establish analogous guarantees for test-time scaling, where curriculum-aware querying reduces both reward oracle calls and sampling cost from exponential to polynomial order.

## 1 INTRODUCTION

When a high-school student first encounters a problem in Riemannian geometry, it is unrealistic to start with a standard textbook such as Do Carmo & Flaherty Francis (1992). Instead, one typically progresses step by step through prerequisite subjects such as algebra, analysis, and geometry, gradually developing the reasoning skills required. A similar principle underlies one of the most effective post-training strategies for foundation models: enhancing mathematical reasoning by encouraging Chain-of-Thought (CoT) generations to evolve from easy to hard, a process known as Curriculum-style Post-training (Liu et al., 2025a; Lee et al., 2025; Bae et al., 2025; Meng et al., 2025; Shi et al., 2025; Wen et al., 2025b; Zhang et al., 2025; Amani et al., 2025; Zhou et al., 2025).

Despite empirical progress, the value of curriculum strategies in post-training remains largely intuitive and lacks rigorous theoretical grounding. Curriculum strategies, both implicit and explicit, have also been explored in pre-training where models are trained from scratch (Bengio et al., 2009; Graves et al., 2017; Narvekar et al., 2020; Wang et al., 2021; Soviany et al., 2022), and some theoretical results exist in specialized function classes such as convex regression and bi-classification (Weinshall et al., 2018; Weinshall & Amir, 2020), parity (Abbe et al., 2023c; Panigrahi et al., 2024), teacher–student perceptrons with sparse features (Saglietti et al., 2022), and $k$-fold composition (Wang et al., 2025). However, these results are highly problem-specific: their notions of "difficulty" and "performance," their handcrafted algorithms, and their proof techniques are tailored to their specific train-from-scratch settings. They cannot be directly applied to transformer-based post-training reasoning, which is characterized by a pretrained base model and explicit CoT generalization. This motivates the central research question:

*When, why, how, and in what sense can curriculum post-training strategies (gradually increasing reasoning difficulty) theoretically improve performance compared to direct learning?*

To bridge this gap, we observe that in post-training, "performance" is naturally measured by the probability of generating the correct CoT, while "difficulty" is often quantified by the *pass-rate*, i.e., the probability that the base model already produces the correct CoT (Tong et al., 2024; Parashar et al., 2025). A principled formalization of this "difficulty" uses the coverage coefficient Foster et al. (2025) $\|\frac{\pi^\star}{\pi_{\mathrm{ref}}}\|_\infty := \sup_x \|\frac{d\pi^\star(\cdot|x)}{d\pi_{\mathrm{ref}}(\cdot|x)}\|_{L^\infty(\pi_{\mathrm{ref}}(\cdot|x))} < \infty^1$ (assuming absolute continuity), which bounds how much rarer a correct CoT can be under the base model's predictive policy $\pi_{\mathrm{ref}}$ compared to the correct CoT policy $\pi^\star$, and thus inversely control the *pass-rate*. In this view, a curriculum corresponds to intermediate policies $\pi_0^\star = \pi_{\mathrm{ref}}, \pi_1^\star, \ldots, \pi_K^\star = \pi^\star$, with the "difficulty" $\|\frac{\pi_k^\star}{\pi_{\mathrm{ref}}}\|_\infty$ nondecreasing in $k \in [K]$.

The next question is: why should stepwise curriculum estimation (i.e., sequentially matching $\pi_1^\star, \ldots, \pi_K^\star$) be more effective than directly learning $\pi^\star = \pi_K^\star$ from $\pi_{\mathrm{ref}}$? Standard rejection-sampling shows that to sample $\pi_k^\star$ from $\pi_{\mathrm{ref}}$, one needs $\Theta(\|\frac{\pi_k^\star}{\pi_{\mathrm{ref}}}\|_\infty \log(\delta^{-1}))$ trials to obtain at least one correct CoT with confidence no less than $1 - \delta$ (Block & Polyanskiy, 2024). Directly targeting $\pi^\star = \pi_K^\star$ thus costs $\Theta(\|\frac{\pi_K^\star}{\pi_{\mathrm{ref}}}\|_\infty \log(\delta^{-1}))$, whereas a stepwise curriculum costs $\sum_{k=0}^{K-1} \Theta(\|\frac{\pi_{k+1}^\star}{\pi_k^\star}\|_\infty \log(\delta^{-1}))$. Hence curriculum shines out if $\sum_{k=0}^{K-1} \|\frac{\pi_{k+1}^\star}{\pi_k^\star}\|_\infty \leq \|\frac{\pi_K^\star}{\pi_{\mathrm{ref}}}\|_\infty$ (assuming **(A1: Realizability)**: $\|\frac{\pi_{k+1}^\star}{\pi_k^\star}\|_\infty, \|\frac{\pi_K^\star}{\pi_{\mathrm{ref}}}\|_\infty < \infty$ ).

Returning to the high-school student analogy: manageable, stepwise difficult courses promote learning efficiency, where appropriately challenging curricula should let the student master the reasoning thinking for Riemannian geometry best within several semesters. In our setting, assuming **(A2: Suitable Step-wise Difficulty)**: $\|\frac{\pi_{k+1}^\star}{\pi_k^\star}\|_\infty \leq \Theta(C^p), \forall k$ for some $p \ll K$ and constant $C > 1$, and **(A3: Intensive Direct Difficulty)**: $\|\frac{\pi_K^\star}{\pi_{\mathrm{ref}}}\|_\infty \geq \Theta(C^{bK-c})$ for the same constant $C > 1$ with $1 \leq b \ll K, c \ll bK$, curriculum post-training can thus achieve an exponential improvement from $\Theta(C^{bK-c})$ to $\Theta(KC^p)$. This intuition is formalized in Thm. 1 and its follow-up discussion.

The natural question is when assumptions **(A1)–(A3)** hold in post-training reasoning. For **(A1)**, Foster et al. (2025) theoretically proved the necessity of base-model coverage (i.e., the probability of sampling the correct CoT), both for RL fine-tuning and reward-guided test-time scaling in linear realizable MDPs. Large-scale empirical evidence further shows that effective RL fine-tuning with sparse rewards merely *reinforces* existing tree-like reasoning patterns in the foundation model, and cannot generate CoTs with zero initial probability (Gandhi et al., 2025; Yue et al., 2025; Wu et al., 2024), underscoring the critical role of the pretrained base model and supporting our **(A1)**.

For **(A2)–(A3)**, we prove that in a state-conditioned autoregressive reasoning process (Def. 2) subsuming many reasoning tasks (Kim et al., 2025c; Nichani et al., 2025; 2024; Gandhi et al., 2024), assumptions **(A2)–(A3)** naturally hold in terms of the sample complexity in RL fine-tuning (Thm. 3), as well as reward-oracle complexity or computational complexity in test-time scaling (Thm. 4).

**Contributions**. We summarize our contributions as follows.

- We formalize curriculum post-training and establish a general bottleneck theorem (Thm. 1 in Sec. 2), showing that under mild coverage and complexity-alignment assumptions, stepwise curriculum post-training converts exponential depth dependence into polynomial order; we further clarify the role of base-model coverage and relax conditions.
- We instantiate the theory with an autoregressive reasoning tree (2S-ART), encompassing many graph-reasoning problems, and its transformer implementation (Sec. 2.1).
- We prove that curriculum strategies—both in RL fine-tuning and test-time scaling—achieve exponential-to-polynomial reductions in sample complexity and oracle-query complexity. We also discuss the limitations, extensions beyond our assumptions, broader applicability to diverse task families, and future directions in Sec. 3.

We delayed the detailed discussions of additional related work in App. A.

---

[1] i.e., the $\ell_\infty$-norm of the Radon–Nikodym derivative of $\pi^\star$ w.r.t. $\pi_{\mathrm{ref}}$. If $\mathcal{X} \times \mathcal{Y}$ are discrete space, it holds that $\|\frac{\pi^\star}{\pi_{\mathrm{ref}}}\|_\infty = \sup_{x \in \mathcal{X}, y \in \mathcal{Y}} \frac{\pi^\star(y|x)}{\pi_{\mathrm{ref}}(y|x)}$,

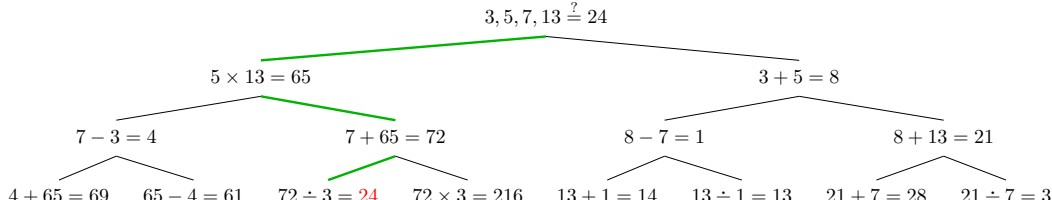

Figure 1: An illustration in Liu et al. (2025b) of the Chain-of-Thought for Countdown game, where the goal is to obtain 24 by applying basic arithmetic operations $(+, -, \times, \div)$ $(\Phi_l(\cdot, \cdot)$ in our Def. 2) between the current step's number (e.g., 13, 65 or 72) and some unused number (e.g., 5, 7 or 3) in $\{3, 5, 7, 13\}$, targeting 24 as the final outcome. Per Parashar et al. (2025), the difficulty measure of Countdown is the number of arithmetic operations required to solve an instance.

## 2 THEORETICAL FRAMEWORK FOR CURRICULUM POST-TRAINING ON REASONING TREES

**Preliminaries and Notations**. For each prompt $x$, a policy $\pi$ is a conditional probability measure $\pi(\cdot \mid x)$ over the output space $\mathcal{O}$, where $o \in \mathcal{O}$ denotes a CoT trajectory. For two policies $\pi$ and $\pi'$, if $\pi'(\cdot \mid x) \ll \pi(\cdot \mid x)$ for all $x$, we write $\left\|\frac{\pi'}{\pi}\right\|_\infty := \sup_{x \in \mathcal{X}} \left\|\frac{d\pi'(\cdot|x)}{d\pi(\cdot|x)}\right\|_{L^\infty(\pi(\cdot|x))} = \sup_{x \in \mathcal{X}} \operatorname{ess\,sup}_{o \sim \pi(\cdot|x)} \frac{d\pi'(\cdot|x)}{d\pi(\cdot|x)}(o)$. Landau symbols follow standard conventions. For $\mathbf{a} \in \mathbb{R}^m$, $\operatorname{softmax}(\mathbf{a})_i := \exp(a_i)/\sum_j \exp(a_j)$, and with temperature $\beta > 0$ we write $\operatorname{softmax}(\mathbf{a}/\beta)$. Finally, $\operatorname{ReLU}(t) := \max\{0, t\}$ elementwise.

As discussed in the introduction, pass rate is the operative notion of difficulty in post-training reasoning (Tong et al., 2024; Parashar et al., 2025), and the inverse of $\|\frac{\pi^\star}{\pi_{\text{ref}}}\|_\infty$ lower-bounds the pass rate and thus serves as a natural proxy for difficulty to master the target task. We formalize the discussions in the introduction as the theorem below.

**Theorem 1.** *Consider a curriculum of $K$ tasks $\pi_0^\star, \pi_1^\star, \ldots, \pi_K^\star$ with base model $\pi_{\text{ref}} := \pi_0^\star$ and target task $\pi^\star := \pi_K^\star$. Denote the learning complexity[2] from $\pi_k^\star$ to $\pi_{k'}^\star$ as $\mathcal{C}(\pi_{k'}^\star \mid \pi_k^\star)$. For any $0 \le k < k' \le K$, suppose $\|\frac{\pi_{k'}^\star}{\pi_k^\star}\|_\infty < \infty$ and*

$$\|\frac{\pi_{k+1}^\star}{\pi_k^\star}\|_\infty = \Theta(C^\star), \tag{1}$$

$$\|\frac{\pi_{k'}^\star}{\pi_k^\star}\|_\infty = \Theta\Big(\prod_{r=k}^{k'-1} \|\frac{\pi_{r+1}^\star}{\pi_r^\star}\|_\infty\Big), \tag{2}$$

$$\mathcal{C}(\pi_{k'}^\star \mid \pi_k^\star) = \widetilde{\Theta}\big(\|\frac{\pi_{k'}^\star}{\pi_k^\star}\|_\infty\big), \tag{3}$$

*for some constant $C^\star > 1$, where $\widetilde{\Theta}(\cdot)$ hide logarithmic factors in a confidence parameter. Then, the complexity of direct learning and step-wise curriculum learning satisfy:*

- *Direct estimation of $\pi_K^\star$ from $\pi_0^\star$ costs $\mathcal{C}_{\text{direct}} := \sum_{k=0}^{K-1} \mathcal{C}(\pi_{k+1}^\star \mid \pi_k^\star) = \Theta\big((C^\star)^K\big)$.*

- *Stepwise curriculum costs $\mathcal{C}_{\text{curriculum}} := \sum_{k=0}^{K-1} \mathcal{C}(\pi_{k+1}^\star \mid \pi_k^\star) = \Theta(KC^\star)$.*

*Consequently, $\mathcal{C}_{\text{direct}}/\mathcal{C}_{\text{curriculum}} = \Omega\big((C^\star)^{K-1}/K\big) \to \infty$ as $K \to \infty$.*

**Exponential–Polynomial Gaps Under Relaxed Conditions**. Even if the exact equalities above are relaxed in two ways: (i) some steps admit polynomial exponents $\mathcal{C}(\pi_{k+1}^\star \mid \pi_k^\star) = (C^\star)^{p_k}$ with $1 \le p_k \ll K$; (ii) the direct complexity satisfies $\mathcal{C}_{\text{direct}} = \Theta\big((C^\star)^{bK-c}\big)$ for some $1 \le b \ll K$ and $c \ll bK$; then writing $p_{\max} := \max_k p_k$ we still have

$$\mathcal{C}_{\text{curriculum}} = O\big(K(C^\star)^{p_{\max}}\big) \ll \Theta\big((C^\star)^{bK-c}\big) = \mathcal{C}_{\text{direct}}. \tag{4}$$

---

[2] The concrete complexity is left to be defined for certain specific scenario. For instance, it can be the sample Complexity of finetuning algorithms (Thm. 3), or the reward-oracle query complexity of test-time scaling (Thm. 4).

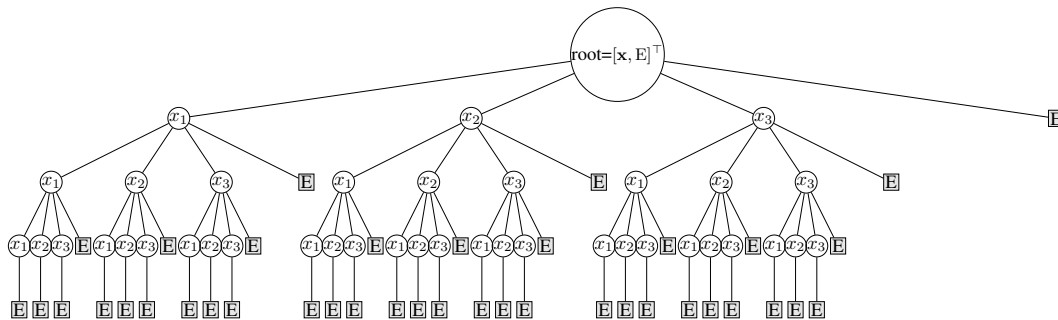

Figure 2: Reasoning tree for parity problems with $d = K = 3$ and input $x_1, x_2, x_3,$ EOS. The nodes on the 2nd–4th levels denote hypotheses about which index is the current secret index, corresponding to $x_{i_1}$, $x_{i_2}$, and $x_{i_3}$, respectively. In our parity CoT class $f \in \mathcal{F}_{\text{2S-ART}}^{\text{parity}}$, each step actually consists of two actions: (i) choose the next secret index $i_t$; (ii) after the choice, apply an XOR over $z_{t-1}$ and $x_{i_t}$ as $z_t = \Phi_l(z_{t-1}, x_{i_t}) := z_{t-1} \oplus x_{i_t}$, as formalized in Eq. (7). For visual clarity, the tree only displays the index-selection branches and omits the explicit XOR updates. E denotes the EOS token. For parity tasks, there are illegal children $\notin \mathcal{I}_l$ for each parent that violate the "legal" criteria: non-repeating indices and strictly increasing order ($i_1 < i_2 < i_3 < \cdots$); thus, for any parity problem, CoTs with duplicate variables or decreasing index order are illegal.

**Role of the Base Model**. The theorem presumes nontrivial coverage of the base model, namely $\pi_k^\star(\cdot \mid x) \ll \pi_{\text{ref}}(\cdot \mid x)$ for all $x$ and $C_{k,0} = \sup_x \left\| \frac{d\pi_K^\star}{d\pi_{\text{ref}}} \right\|_{L^\infty(\pi_{\text{ref}})} < \infty$. This does not hold for an untrained model from scratch but is plausible in post-training, aligning with theory showing that base-model coverage upper-bounds what fine-tuning or test-time scaling can achieve (Foster et al., 2025) and with empirical evidence that sparse-reward RL predominantly reinforces pre-existing tree-like reasoning patterns rather than creating support where the base model has zero probability (Snell et al., 2024; Yue et al., 2025; AI et al., 2025; Gandhi et al., 2025).

**Related Theoretical Context**. Indeed, the assumptions in Parashar et al. (2025) closely mirror those above in the case of Approximate Policy Iteration settings. We discuss this comparison in App. E. Furthermore, for linearly realizable Markov Decision Processes, leveraging the spanner-sampling framework of Foster et al. (2025), we can also prove an exponential improvement in inference-time computational complexity (Thm. 5) by similar assumptions; see App. E for details.

In subsequent sections, we show that this theorem naturally manifests in tree-reasoning settings, where the subtasks correspond to *prefix reasoning traces* (Parashar et al., 2025) or *hint-assisted reasoning* (Liu et al., 2025b; Amani et al., 2025) of the target task, and each parent state assigns comparable probability mass across its children, thereby satisfying Eq. (1) and Eq. (2). We further show the representation theorem where the transformer can subsume the autoregressive reasoning tree, and proved that Eq. (4) holds in both RL finetuning and test-time scaling in terms of sample complexity and oracle complexity.

### 2.1 Transformer Reasons over a States-Conditioned Reasoning Tree

Building on prior work that views post-training as reweighting over a pre-trained reasoning tree (Snell et al., 2024; Yue et al., 2025; AI et al., 2025; Gandhi et al., 2025; Liu et al., 2025b), we model the reasoning from a tree-causal, low-order autoregressive perspective, per formalized below.

**Definition 1** (2-States Conditioned Autoregressive Reasoning Tree (2S-ART)). *A 2S-ART $\mathcal{F}_{\text{2S-ART}}$ $(\{\Phi_l\}_{l\leq L}, \{\mathcal{I}_l\}_{l\leq L})$ is a function class of autoregressive tasks $f : [K]^d \times \{K+1\} \to [K+1]^{\leq L+1}$ such that, given input $\mathbf{x} = (x_1, \ldots, x_d) \in [K]^d$ with terminal token EOS $:= K+1$, generates a chain-of-thought $(z_1, \ldots, z_\ell,$ EOS$)$. At each step $l$, the model chooses an index $i_l$ from a legal set $\mathcal{I}_l$ of size $\Theta(d)$, reads the corresponding element $v_{i_l}$ from the current sequence, and updates its reasoning state by a two-states map $z_l = \Phi_l(z_{l-1}, v_{i_l})$, with $z_0 =$ EOS and $\Phi_l(z,$ EOS$) =$ EOS.*

*A task $f \in \mathcal{F}_{\text{2S-ART}}$ is identified with its index path $S = (i_1, \ldots, i_k, d+1)$, the associated curriculum subtask family $\{f_{S^1}, \ldots, f_{S^k}\}$ corresponds to the prefix paths $S^l = (i_1, \ldots, i_l, d+1), l \in [k]$.*

Fig. 2 is an example of parity with $d = K = L = 3$, and Fig. 1 is an instance of Coundown with $d = 4, K = \mathbb{Z}^+, L = 3$ without illustrating explicit EOS. Here EOS is the end-of-sequence token, and $z_l$ is the $l$-th reasoning state in the CoT. Depending on the task, $z_l$ may be a single token (e.g., in Eq. (7) where $\Phi_l(z_{l-1}, v_l(i_l)) = z_{l-1} \oplus v_l(i_l)$) or a short expression (e.g., in the countdown game where $\Phi_l(z_{l-1}, v_l(i_l))$ applies $(+, -, \times, \div)$ to $z_{l-1}$ and $v_l(i_l)$; see Fig. 1). The legal index set $\mathcal{I}_l$ encodes real-world selection rules, such as not reusing indices or numbers in parity or Countdown. The 2S-ART also subsumes prior abstractions of reasoning behavior:

- Markov-chain reasoning (Kim et al., 2025c) $(\Phi_l(z_{l-1}, v_l(i_l)) = \phi(z_{l-1})$ for some $\phi)$;
- Induction-head for associative recall (Nichani et al., 2025) $(\Phi_l(z_{l-1}, v_l(i_l)) = v_l(i_l))$;
- causal-graph reasoning (Nichani et al., 2024) $(\Phi_l(z_{l-1}, v_l(i_l)) = \phi(v_l(i_l))$ for some $\phi)$.

A detailed version of Def. 1 (Def. 3) appears in App. G. To rule out ambiguities where different index paths represent the same task, we adopt the uniqueness assumption below.

**Assumption 1.** *We assume the uniqueness of the* 2S-ART $\mathcal{F}_{\text{2S-ART}}$ $(\{\Phi_l\}_{l \leq L}, \{\mathcal{I}_l\}_{l \leq L})$ *in terms of the index path: for* $\forall f_{S_\star}, f_{S'_\star} \in \mathcal{F}_{\text{2S-ART}}$ *with* $|S_\star|, |S'_\star| \leq L + 1$, *if* $|S_\star| \neq |S'_\star|$, *then* $f_{S_\star} \neq f_{S'_\star}$.

A natural way to define a base model with general capability over the task class $\mathcal{F}_{\text{2S-ART}}$ is to suggest that, at each depth $l \in [L]$, the model assigns uniform probability to all children in the legal branch (Liu et al., 2025b) as below.

**Definition 2** (Probabilistic 2S-ART Base Model (PART)). *Consider a* 2S-ART $\mathcal{F}_{\text{2S-ART}}$ $(\{\Phi_l\}_{l \leq L}, \{\mathcal{I}_l\}_{l \leq L})$ *in Def. 2 with* $L \ll d$, *a* PART *samples* $S_\star$ *by uniformly drawing* $i_l \in \mathcal{I}_l(\mathbf{CoT}_{l-1}), \forall l \in [L]$, *i.e.* $P(i_l \mid z_{<l}) = \frac{1}{|\mathcal{I}_l(\mathbf{CoT}_{l-1})|} = \Theta(d^{-1})$.

This uniform selection rule in the 2S-ART base model leads directly to an exponential decay in success probability (*pass-rate*) of tasks with depth $l$, as summarized below.

**Corollary 1** (Exponential Decay of Success Probability with Depth). *Consider a* 2S-ART *sampler defined in Def. 1. For a fixed target* $f_{S_\star} \in \mathcal{F}_{\text{2S-ART}}$ *with associated curriculum subtask family* $\mathcal{F}_{S_\star} = \{f_{S_\star^1}, \ldots, f_{S_\star^{k^\star}}\}$, *the probability that* PART *samples the legal CoT for* $f_{S_\star^l}$ *is* $\Theta(d^{-(l+1)})$.

**Link to the *pass-rate* in Post-training**. The exponential-decay property of our autoregressive reasoning tree naturally satisfies Eqs. (1) (2). Consider the policy $\pi_{S^l_\star}$ for $f_{S^l_\star}$: given any $\mathbf{x} \in [K]^d$, $\pi_{S^l_\star}$ samples correct $i_1^\star, \ldots, i_l^\star$ in the correct order almost surely, thereby generating correct prefix $z_1^\star, \ldots, z_l^\star$ autoregressively, and from step $l + 1$ onward copies the behavior of PART. Denote $\pi_{\text{PART}}$ as the policy of PART. It is straightforward to verify that $\|\frac{\pi_{S^{l+1}_\star}}{\pi_{S^l_\star}}\|_\infty = \Theta(d)$ and $\|\frac{\pi_{S^l_\star}}{\pi_{\text{PART}}}\|_\infty = \Theta(d^{l+1})$ (**A1**), which align exactly with Eqs. (1) and (2) up to a $d$. In the next step, we demonstrate that a transformer can replicate PART, and then establish that Eq. (4) (corresponding to (**A2**)–(**A3**) in the introduction) also holds, both for the sample complexity of transformer's fine-tuning convergence and for the oracle-query complexity in test-time scaling.

**Transformer** ($\text{TF}(\cdot; \mathbf{W})$) **as Learning Model**. Let $\mathbf{U} = [\boldsymbol{\mu}_1, \ldots, \boldsymbol{\mu}_K, \boldsymbol{\mu}_{\text{EOS}}] \in \mathbb{R}^{d_X \times (K+1)}$ be the embedding vocabulary, where the $k$-th column is the embedding of token $k$, and $\boldsymbol{\mu}_{\text{EOS}}$ the embedding of EOS. Following Li et al. (2025), positional encodings $\mathbf{P} := [\mathbf{p}_1, \ldots, \mathbf{p}_{d+1}]$ are mutually orthogonal and satisfy $\mathbf{p}_i \perp \mathbf{U}$. They are concatenated with token embeddings as $\mathbf{E}[x_m] = [\boldsymbol{\mu}^{x_m \top}, \mathbf{p}_m^\top]^\top$, $\mathbf{E}[\text{EOS}] = [\boldsymbol{\mu}_{\text{EOS}}^\top, \mathbf{p}_{d+1}^\top]^\top$, and $\mathbf{E}[z_l] = [\boldsymbol{\mu}^{z_l \top}, \mathbf{p}_{i_l}^\top]^\top, \forall m \in [d], l \in [L + 1]$. The transformer $\text{TF}(\cdot; \mathbf{W})$ aims to recover the functions defined in the tree graph $f \in \mathcal{F}_G$, with embedded input $\mathbf{E}[x_1], \ldots, \mathbf{E}[x_d], \mathbf{E}[\text{EOS}] \in \mathbb{R}^{d_E}$, where $d_E = 2d_X$. Denote $\mathbf{E}[z_{-d}] := \mathbf{E}[x_1], \ldots, \mathbf{E}[z_{-1}] := \mathbf{E}[x_d], \mathbf{E}[z_0] := \mathbf{E}[\text{EOS}]$, the transformer performs next-token prediction via:

$$\text{TF}(\mathbf{E}[z_{-d}], \ldots, \mathbf{E}[z_{-1}], \mathbf{E}[z_0], \mathbf{E}[z_1], \ldots, \mathbf{E}[z_{l-1}]; \mathbf{W}) = \mathbf{E}[z_l] \tag{5}$$

Here, original data embeddings remain fixed during the reasoning process, while CoT state $\mathbf{E}[z_l]$ are generated by $\text{TF}(\cdot; \mathbf{W})$ ($\mathbf{W}$ denotes the model weights) via:

$$\hat{\mathbf{p}}_l = \sum_{j=-d}^{l-2} \mathbf{V}(\mathbf{E}[z_j] \, \text{softmax}(\mathbf{E}[z_j]^\top \mathbf{K}^\top \mathbf{Q} \mathbf{E}[z_{l-1}])) \in \mathbb{R}^{d_X}$$

$$\mathbf{p}_{i_l} \sim \text{softmax}(\hat{\mathbf{p}}_l^\top \mathbf{P} / \beta), \quad \text{The algorithm terminates with } \mathbf{E}[\text{EOS}] \text{ if } \mathbf{p}_{d+1} \text{ is sampled} \tag{6}$$

$$\hat{\boldsymbol{\mu}}^{z_l} = \text{FFN}_l(\boldsymbol{\mu}^{x_{i_l}}, \hat{\boldsymbol{\mu}}^{z_{l-1}}), \quad \mathbf{E}[z_l] = [\hat{\boldsymbol{\mu}}^{z_l}, \mathbf{p}_{i_l}]^\top \in \mathbb{R}^{d_E}.$$

Here, $\beta > 0$ is a temperature parameter, the causal masking is enforced by setting $\mathbf{E}[z_j]^\top \mathbf{K}^\top \mathbf{Q} \mathbf{E}[z_l] \leftarrow -\infty$ whenever $j \geq l$ or $l \leq d$. $\hat{\boldsymbol{\mu}}^{x_{i_l}}$ is depth-specific crucial token utilized for the $l$-th reasoning step, and $\hat{\boldsymbol{\mu}}^{z_l}$ is the resulting $l$-th reasoning step, formed by the depth-specific non-linear residual Feedforward layer $\mathrm{FFN}_l(\cdot, \cdot)$, which is tailored for task-specific reasoning operation $\Phi_l(\cdot, \cdot)$ defined in Def. 1, modeling after reasoning steps executed by the neural model such as the XOR operation computed by $h^\top \mathrm{ReLU}(W[\begin{smallmatrix} \mathbf{E}[z_{l-1}]:d_{\mathrm{X}} \\ \hat{\boldsymbol{\mu}}^{x_{i_l}} \end{smallmatrix}])$ in Wen et al. (2025a) or $\phi(\cdot)$ function in Kim & Suzuki (2025), and the calculation operation in each reasoning step of countdown game, per Fig. 1 (Liu et al., 2025b). The following shows that $\mathrm{TF}(\cdot)$ can replicate the PART per Def. 2.

**Theorem 2** (Base Model as PART ($\mathrm{TF}(\cdot; \mathbf{W}_{\mathrm{base}})$))**.** *Fix any 2S-ART ($\{\Phi_l\}_{l \leq L}, \{\mathcal{I}_l\}_{l \leq L}$) from Def. 1 and its probabilistic base model (PART) in Def. 2. Assume that for each depth $l$ there exists the map $\mathrm{FFN}_l$ that replicates the target operation on embeddings, i.e., $\boldsymbol{\mu}^{z_l} = \mathrm{FFN}_l\left(\boldsymbol{\mu}^{v_l(i_l)}, \mathbf{E}[z_{l-1}]:d_{\mathrm{X}}\right)$ whenever $z_l = \Phi_l(z_{l-1}, v_l(i_l))$. Then there exists a parameterization of a single-head attention transformer of the form in Eq. (6) to copy the PART behavior in Def. 2.*

The assumption that $\mathrm{FFN}_l$ exists for each depth $l$ is justified by the universal approximation theorem (Cybenko, 1989; Hornik, 1991). This establishes that feedforward networks can approximate any continuous function on compact subsets of $\mathbb{R}^n$ to arbitrary precision. Since continuous functions are dense in $L^p$ spaces for $1 \leq p < \infty$, this extends to approximating most practically relevant functions. Recent advances show that ReLU networks of width $d + 3$ and arbitrary depth can approximate any scalar continuous function of $d$ variables (Lu et al., 2017), while Suzuki (2018); Suzuki & Nitanda (2021) demonstrates optimal approximation rates for functions in Besov spaces, encompassing broader function classes beyond continuity.

0-1 **Outcome Signals**. In real-world mathematical reasoning and program-like tasks, supervision is often available only at the outcome level (correct vs. incorrect), while collecting step-by-step intermediate supervision is costly or infeasible. We therefore adopt 0-1 outcome-only supervision that evaluates only the *final pre-EOS prediction*. We use two oracles:

- $\mathbf{R}_{\mathbf{x}}^{f_{S_\star}}(\cdot)$: returns 1 iff, immediately before sampling EOS, $\mathrm{TF}(\cdot; \mathbf{W})$ outputs the token whose embedding equals $\boldsymbol{\mu}^{f_{S_\star}(\mathbf{x})}$; else 0.

- $\mathbf{R}_{\mathbf{x}}^{\mathcal{F}_{\mathbf{S}_\star}}(\cdot, \ell)$: for the curriculum subtask family $\mathcal{F}_{S_\star} = \{f_{S_\star^1}, \ldots, f_{S_\star^{k^\star}}\}$ per Def. 1, returns 1 iff the pre-EOS token equals $\boldsymbol{\mu}^{f_{S_\star^\ell}(\mathbf{x})}$ for $\ell \in [k^\star]$; else 0.

These outcome-only definitions are agnostic to any specific case study; in the parity case discussed in Sec. 2.1.1, outcome reward specializes to whether the result is correct per Eq. (7); in the multi-step language translation task studied in Abedsoltan et al. (2025) where the sampled word is translated to different language at different reasoning steps according to a fixed order, the outcome reward denotes whether the translation is correct for the task family.

**Challenge: Inherent Reward Hacking**. Outcome-only supervision on the final pre-EOS token allows many spurious CoTs to be accepted for a fixed input $\mathbf{x}$, because the oracle **only checks the surface token** and does not constrain the internal index-selection process of the 2S-ART ($\{\Phi_l\}_{l \leq L}, \{\mathcal{I}_l\}_{l \leq L}$). For instance, in parity, choosing any wrong index at some depth can still yield the correct final bit with probability $1/2$ under $\mathbf{x} \sim \mathrm{Unif}(\{0,1\}^d)$.

### 2.1.1 CONCRETE EXAMPLE: PARITY PROBLEM

**Sparse Parity Problem.** We remark that the Sparse Parity Problem can be seen as a $K = 2$ subcase of Def. 1, with the deterministic kernel $\Phi_l(z_{l-1}, x_{i_l})$ given by the XOR operation $z_{l-1} \oplus x_{i_l}$. Formally, given a $d$-dimensional binary input vector $\mathbf{x} = (x_1, \ldots, x_d) \sim \mathrm{Unif}\{0,1\}^d$, define a class of Boolean functions determined by a size-$k$ secret index set $S \subseteq [d]$. Each function computes the parity (XOR-sum) of the input components specified by $S$: $f_S(\mathbf{x}) = \bigoplus_{i \in S} x_i = x_{i_1} \oplus x_{i_2} \oplus \cdots \oplus x_{i_k}$, where $\oplus$ denotes XOR and $i_1 < i_2 < \cdots < i_k$ without loss of generality. The output satisfies $f_S(\mathbf{x}) = 1$ if $\sum_{i \in S} x_i$ is odd, and $f_S(\mathbf{x}) = 0$ otherwise. The function class $\mathcal{P}_{d,k} = \mathrm{Parity}(d, k)$ contains all such functions, with cardinality $|\mathcal{P}_{d,k}| = \binom{d}{k}$.

**CoT and Curriculum Subtask Family for Parity**. As a $K=2$ instance of Def. 1, parity's CoT is indexed by the path $S = (i_1, \ldots, i_k)$ with $i_k = d+1$ (EOS). The resulting XOR-based CoT (Wen

et al., 2025a; Abedsoltan et al., 2025) is:

$$z_1 = x_{i_1}, \quad z_2 = x_1 \oplus x_{i_2}, \quad \ldots, \quad z_k = z_{k-1} \oplus x_{i_k} = f_S(\mathbf{x}). \quad i_1 < i_2 < \ldots < i_k \tag{7}$$

In this case, the curriculum subtask family is the prefix-index family $\mathcal{F}_S = \{f_{S^1}, \ldots, f_{S^k}\}$ with $S^{k'} = (i_1, \ldots, i_{k'})$ and $f_{S^{k'}}(\mathbf{x}) = z_{k'}$ for $k' \in [k]$; early termination by EOS follows Def. 1. In particular, the legal sets $\{\mathcal{I}_l\}$ is defined to satisfy $i_1 < i_2 < \cdots < i_k$.

**Transformer** $(\mathrm{TF}(\cdot; \mathbf{W}))$ **as Learning Model**. We follow Eq. (5), Eq. (6), set the vocabulary as $\mathbf{U} = [\boldsymbol{\mu}^0, \boldsymbol{\mu}^1, \boldsymbol{\mu}_{\mathrm{EOS}}] = [\boldsymbol{e}_1, \boldsymbol{e}_2, \boldsymbol{e}_3]$ for $0, 1, \mathrm{EOS}$, and the $\mathrm{FFN}_l$ is instantiated for XOR operation:

$$\mathrm{FFN}_l = \mathbf{W}_2 \, \mathrm{ReLU}[\mathbf{W}_1(\hat{\boldsymbol{\mu}}^{x_{i_l}} + \mathbf{E}[z_{l-1}]_{:d_{\mathrm{X}}})], \quad \mathbf{W}_1 = \begin{bmatrix} \frac{1}{2} & \frac{1}{2} & \frac{1}{2} \\ 0 & 1 & 0 \\ -\frac{1}{2} & \frac{1}{2} & -\frac{1}{2} \end{bmatrix}, \quad \mathbf{W}_2 = \begin{bmatrix} 1 & -1 & 2 \\ 0 & 1 & -2 \\ 0 & 0 & 0 \end{bmatrix}, \forall m \in [k']. \tag{8}$$

It can be checked directly that this ensure our $\mathrm{FFN}_l$ replicates the $\Phi_l(z_{l-1}, v_l(i_l)) = z_{l-1} \oplus v_l(i_l)$:

$$\mathrm{FFN}_l(\boldsymbol{\mu}^0, \boldsymbol{\mu}^0) = \mathrm{FFN}_l(\boldsymbol{\mu}^1, \boldsymbol{\mu}^1) = \boldsymbol{\mu}^0, \quad \mathrm{FFN}_l(\boldsymbol{\mu}^0, \boldsymbol{\mu}^1) = \mathrm{FFN}_l(\boldsymbol{\mu}^1, \boldsymbol{\mu}^0) = \boldsymbol{\mu}^1, \quad \mathrm{FFN}_l(\boldsymbol{\mu}^{0/1}, \boldsymbol{\mu}^{\mathrm{EOS}}) = \boldsymbol{\mu}^{0/1}.$$

Per in previous Sec. 2.1, oracle $R^{f_{S_\star}}(\cdot)$ and $R^{\mathcal{F}_{S_\star}}(\cdot)$ for the target task $f_{S_\star}, |S_\star| = k + 1$, rather than direct intermediate supervision used in Kim & Suzuki (2025). Kim & Suzuki (2025) also considered data augmentation scheme (causal mask + random $d$-bit + ultra filter) circumvents the need for intermediate supervision, but is specifically tailored to their 2-parity–based hierarchical CoT when $k^\star = 2^v, v \in \mathbb{Z}^+$. By contrast, we adopt the more natural CoT formulation in Eq. (7).

### 2.1.2 Outcome Signal-based RL Finetunings by Gradient Descent

For mathematics benchmarks, the conventional REINFORCE objective is used to increase the probability that sampled CoTs yield correct answers Xiong et al. (2025); Setlur et al. (2025). In our setting, the training objective can be formulated as

$$\mathcal{J}_{\mathrm{REINFORCE}}^{k^\star}(\mathbf{W}^{k^\star}) = \mathbb{E}\left[ R^{k^\star}(\mathrm{TF}(\cdot; \mathbf{W})) \right], \tag{9}$$

where $R^{k^\star}(\cdot) \in \{R_{\mathbf{x}}^{f_{S_\star}}(\cdot), R_{\mathbf{x}}^{\mathcal{F}_{S_\star}}(\cdot)\}$, $\hat{p}_{\mathbf{W}^{k^\star}}(\cdot|\cdot)$ denotes the predictive distribution by $\mathrm{TF}(\cdot; \mathbf{W})$, and $\mathbf{W} \in \mathbb{R}^{(d_{\mathrm{E}} - d_{\mathrm{X}})^2}$ is the only matrix we consider trainable in the parametrization of Thm. 2:

$$\mathbf{K}^\top \mathbf{Q} = \begin{pmatrix} \mathbf{0}_{d_{\mathrm{X}} \times d_{\mathrm{X}}} & \mathbf{0}_{d_{\mathrm{X}} \times (d_{\mathrm{E}} - d_{\mathrm{X}})} \\ \mathbf{0}_{(d_{\mathrm{E}} - d_{\mathrm{X}}) \times d_{\mathrm{X}}} & \mathbf{W} \end{pmatrix}, \qquad \mathbf{V} = \begin{pmatrix} \mathbf{0} & \mathbf{I}_{d_{\mathrm{E}} - d_{\mathrm{X}}} \end{pmatrix}, \tag{10}$$

where the $\mathbf{0}, \mathbf{V}$, as well as the feedforwad map $\mathrm{FFN}_l$, is considered fixed during finetuning. This type of reparametrization is common in the transformer optimization literature to enable tractable analysis (Zhang et al., 2024b; Huang et al., 2024; Mahankali et al., 2023; Kim & Suzuki, 2025).

In mathematical datasets, the difficulty measure *pass-rate* often coincides with other task-specific measures Parashar et al. (2025): Blocksworld uses plan length of CoT (Valmeekam et al., 2023), Countdown counts the number of operations in the CoT (Gandhi et al., 2024), and in parity tasks, the number of XOR operations naturally reflects the CoT difficulty. Building on these observations, a natural curriculum for $\mathcal{F}_{S_\star}$ is to gradually increase reasoning depth from shallow to deep, where the number of reasoning operations grows with depth. Separately, Liu et al. (2025b); Amani et al. (2025) demonstrated the benefit of providing hints (partial CoT prefixes) and progressively shortening them so that the model completes longer suffixes. Accordingly, we consider two categories of curriculum finetuning under the Outcome Signal Oracle $\mathbf{R}_{\mathbf{x}}^{\mathcal{F}_{S_\star}}(\cdot)$:

- **Depth-increasing Curriculum** (Parashar et al., 2025): at the $l$-th stage, the algorithm truncate CoTs from $\mathrm{TF}(\cdot, \mathbf{W}^{(t)})$ with EOS to ensure length $l + 1$.

- **Hint-decreasing Curriculum** (Liu et al., 2025b): at the $l$-th stage, the algorithm provides a CoT prefix of length $k^\star + 1 - l$, letting $\mathrm{TF}(\cdot, \mathbf{W}^{(t)})$ generate the remaining steps.

Building on this, the following theorem establishes Eq. (4) (corresponding to **(A2)–(A3)** in the introduction) within the autoregressive reasoning tree setting, demonstrating that curriculum-based RL fine-tuning effectively alleviates the exponential bottleneck in sample complexity.

**Theorem 3** (Curriculum RL Finetuning Avoid Exponential Bottleneck). *Let* $\mathbf{U}$ *be an orthogonal matrix with* $d_{\mathrm{X}} = \Theta(K), d_{\mathrm{E}} = \Theta(d + L)$, $\mathrm{TF}(\cdot; \mathbf{W}_{base})$ *per in Thm. 2 with trainable* $\mathbf{W}$, *and a target* $f_{S_\star} \in \mathcal{F}_{\text{2S-ART}}$ *with* $|S_\star| = k^\star + 1$. *For any* $\varepsilon \in (0, 1)$, *using the RL objective in Eq. (9) and one gradient step per stage (a single step for no-curriculum;* $k^\star + 1$ *online steps for curricula), with probability no less than* $1 - \delta$, *there exist learning-rate choices* $\eta$ *such that*

1. *No-curriculum* ($R_{\mathbf{x}}^{f_{S_\star}}(\cdot)$ *as oracle, one-shot update with* $\eta = \tilde{\Theta}(\beta d^{k^\star + 1})$): *the sample complexity to achieve* $\mathbb{E}_{\mathbf{x} \sim \mathrm{Unif}([K]^d)}\left[R_{\mathbf{x}}^{f_{S_\star}}(\mathrm{TF}(\mathbf{x}; \mathbf{W}))\right] \geq 1 - \varepsilon$ *is at least* $n \geq \tilde{\Omega}(d^{2k^\star + 2})$.

2. *Depth-increasing Curriculum and Hint-decreasing Curriculum* ($\mathbf{R}_{\mathbf{x}}^{\mathcal{F}_{S_\star}}(\cdot)$ *as oracle,* $k^\star + 1$ *online updates with* $\eta = \tilde{\Theta}(\beta d^2)$): *the sample complexity to achieve* $\mathbb{E}_{\mathbf{x} \sim \mathrm{Unif}([K]^d)}\left[R_{\mathbf{x}}^{f_{S_\star}}(\mathrm{TF}(\mathbf{x}; \mathbf{W}))\right] \geq 1 - \varepsilon$ *is at most* $n \leq \tilde{O}((k^\star + 1)d^2)$.

*Here* $\tilde{\Omega}(\cdot)$ *and* $\tilde{O}(\cdot)$ *hide polylogarithmic factors in* $d, 1/\delta$ *and* $1/\varepsilon$, *and absolute constants (e.g., temperature* $\beta$) *as well as task-dependent spurious-acceptance parameters.*

**Remark**. Item (1) is not minimax-tight in certain cases of 2S-ART (for example, the minimax rate for the parity class up to EOS is $d^{k^\star}$). The gap arises because our analysis is based on vanilla gradient descent over the empirical estimate of outcome rewards Eq. (9), which disallows cross-sample referencing or structure-exploiting techniques such as Gaussian elimination (Raz, 2018; Abbe & Sandon, 2020). Nevertheless, our result already *suffices*: Item (2) establishes that curriculum post-training can reduce the sample complexity to polynomial order. The specific $d^2$ dependence stems from distinguishing margins of order $\Theta(d^{-1})$ in parallel attempts, which information-theoretically requires inverse-square costs. More sophisticated noisy SGD algorithms that enables cross-sample reference may improve this rate at the cost of a $\mathrm{poly}(d)$ time complexity (Cornacchia & Mossel, 2023; Abbe et al., 2023c).

**Related Work**. It is worth noting that one- or few-shot convergence under large learning rates has been widely discussed in prior optimization work, such as Cornacchia & Mossel (2023); Kim & Suzuki (2025). As mentioned in the introduction and Sec. A, there are also theoretical investigations of curriculum benefits in pre-training (train-from-scratch) settings for the parity function class (Cornacchia & Mossel, 2023; Abbe et al., 2023c; Panigrahi et al., 2024). In particular, Cornacchia & Mossel (2023) and Abbe et al. (2023c) construct curricula by mixing data distributions, where "difficulty" is characterized by the density of Hamming weight (fewer 1s than 0s are easier), while Panigrahi et al. (2024) study a teacher–student setup in which "difficulty" is defined by the signal strength of checkpoints provided by the teacher. Their analyses focus on 2-layer ReLU networks or MLPs trained by carefully designed stage-wise or layer-wise gradient descent algorithms. In contrast, our perspective builds on the difficulty measure *pass-rate* (see Cor. 1 and its remark) and its connection to the inherently probabilistic, tree-like CoT generation behavior in post-training (Liu et al., 2025b). This leads naturally to the depth-increasing Parashar et al. (2025) and hint-decreasing (Liu et al., 2025b; Amani et al., 2025) curriculum, and we consider GD updates of transformer without additional algorithmic tailoring, better mirroring the post-training scenarios.

**Proof Outline**. Intuitively, with a noiseless gradient oracle, the gradient magnitude in the projection space corresponding to the correct secret index strictly exceeds all others, so a single clean update suffices to create a decisive logits gap for softmax sampling. Thus the sample complexity is determined by how many samples are needed so that the estimated gradients reliably reflect this margin. A sufficient condition is that the confidence intervals of both the correct and incorrect indices are smaller than half of the smallest expected margin. From gradient calculations, the expected margin at step $\ell$ is $\Theta(\beta^{-1} d^{-(k+2-\ell)})$. To ensure confidence radius no larger than $\Theta(\beta^{-1} d^{-(k+2-\ell)}/2)$, Bernstein-type bounds with a union argument yield $n_\ell = \tilde{\Theta}(d^{2k+2-\ell})$. The first step dominates with $n_1 = \tilde{\Theta}(d^{2k+2})$, giving total sample complexity $n \geq \tilde{\Omega}(d^{2k+2})$. In contrast, under curriculum strategies, the expected margin at each step is $\Theta(\beta d^{-1}/2)$, so the required samples per step are at most $\tilde{\Theta}(d^2)$. Across $k + 1$ steps this totals $\tilde{O}((k + 1)d^2)$.

### 2.1.3 Outcome Signal-based Test-time Scaling

Consider a pretrained transformer that replicates PART's uniform legal-branching behavior (Thm. thm:tf-realizes-part). At test time, given $\mathbf{x} \sim \text{Unif}([K]^d)$, we have access only to the oracles $\mathbf{R}_{\mathbf{x}}^{f_{S_\star}}(\cdot)$ and $\mathbf{R}_{\mathbf{x}}^{\mathcal{F}_{S_\star}}(\cdot)$, which serve as Outcome Reward Models (ORM) for inference-time scaling. Following Foster et al. (2025), we denote by $T_{\text{data}}$ the *reward oracle-query complexity*, i.e., the total number of evaluations of $\mathbf{R}_{\mathbf{x}}^{f_{S_\star}}$ or $\mathbf{R}_{\mathbf{x}}^{\mathcal{F}_{S_\star}}$, and by $T_{\text{comp}}$ the *model-sampling complexity*, i.e., the total number of token emissions from $\text{TF}_{\text{base}}$. The theorem below shows that leveraging $\mathbf{R}_{\mathbf{x}}^{\mathcal{F}_{S_\star}}(\cdot)$ in a curriculum-based manner eliminates the exponential bottleneck in terms of $T_{\text{data}}$ and $T_{\text{comp}}$.

**Theorem 4** (Curriculum Test-time Scaling Avoid Exponential Bottleneck). *Let* $\text{TF}_{base} := \text{TF}(\cdot; \mathbf{W}_{base})$ *per defined in Thm. 2, and consider a target* $f_{S_\star} \in \mathcal{F}_{\text{2S-ART}}$ *with* $|S_\star| = k^\star + 1$. *Then, for identifying the ground-truth path* $S_\star$ *with confidence* $1 - \delta$:

1. *Using only* $\mathbf{R}_{\mathbf{x}}^{f_{S_\star}}(\cdot)$, *any procedure (including ones that force visible x-tokens by rejection sampling and then roll out) requires* $T_{data} \geq \tilde{\Omega}(d^{2k^\star})$ *and* $T_{comp} \geq \tilde{\Omega}(d^{2k^\star})$.

2. *Using curriculum queries adaptively to* $\mathbf{R}_{\mathbf{x}}^{\mathcal{F}_{S_\star}}(\cdot)$, *there exists a procedure with* $T_{data} \leq \tilde{O}((k^\star + 1)d^2)$ *and* $T_{comp} \leq \tilde{O}((k^\star + 1)d^3)$.

*Here* $\tilde{\Omega}(\cdot)$ *and* $\tilde{O}(\cdot)$ *hide polylogarithmic factors in* $d$, $1/\delta$ *and spurious-acceptance parameters.*

Akin to the remark of Thm. 3, without structure-exploiting procedure (Raz, 2018; Abbe et al., 2023a), our results match the information-theoretical bound in distinguishing the acceptance gap.

**Proof Outline**. We again illustrate the idea using the parity class. The algorithm's goal, given input $\mathbf{x}$, is to recover the secret index set $S = \{i_1, \ldots, i_k, d+1\}$. The key difficulty is that an incorrect index set still yields reward 1 with probability $1/2$, a hallmark of parity. At reasoning step $\ell$, conditioned on a correct prefix, both the correct index $j_\star$ and any wrong index $j \neq j_\star$ are selected with probability $\Theta(d^{-1})$. If $j_\star$ is chosen and the process continues until EOS, the final CoT obtains reward 1 with probability $\Theta(d^{-(k+1-\ell)}) + (1 - \Theta(d^{-(k+1-\ell)}))/2$, whereas if $j \neq j_\star$, the reward is $1/2$. Hence the probability gap is $\Theta(d^{-(k+1-\ell)}/2)$. Distinguishing such tiny gaps information-theoretically requires $\tilde{\Theta}(d^{2k+2-2\ell})$ trials, and summing over $\ell$ yields $\tilde{\Omega}(d^{2k})$ reward queries in total. By contrast, with curriculum (Alg. 6), the algorithm truncates the generation at length $\ell + 1$. Then, if $j_\star$ is chosen, the chance of reward 1 is $\Theta(d^{-1}) + (1 - \Theta(d^{-1}))/2$, while if $j \neq j_\star$, it is $1/2$. The resulting gap is $\Theta(d^{-1}/2)$, requiring only $\tilde{O}(d^2)$ trials to resolve. Across $k + 1$ stages this totals $\tilde{O}((k + 1)d^2)$ reward-oracle calls, and the additional transformer calls during rejection sampling after $S$ is identified contribute at most $\tilde{\Theta}(d)$ more.

## 3 Conclusion, Discussion, and Future Work

Unlike prior theoretical studies of pre-training curricula, where "difficulty" is task-specific, our framework for post-training is built on the widely used *pass-rate*. Based on this measure, we established Thm. 1, which characterizes when curriculum post-training avoids the exponential bottleneck. We substantiated this by modeling the autoregressive tree-like reasoning process, where reasoning length scales with subtask difficulty and can be replicated by a pretrained transformer. Considering both fine-tuning and test-time scaling, we showed exponential-to-polynomial reductions in sample and oracle-query complexity, serving as a concrete case study of Thm. 1.

A limitation is that our 2S-ART (PART) abstraction is restrictive. For instance, datasets such as Blocksworld involve $\Phi_l(x_{i_l^1}, x_{i_l^2})$ (choosing two states and stacking one on the other), beyond our one-state case. We believe the theory extends with more complex treatment. While our analysis focused on transformer post-training, the benefit of curricula in avoiding exponential bottlenecks is model-agnostic, as highlighted in Thm. 1 and Sec. E. Empirical studies also report easy-to-hard curricula to be effective in diffusion models (Kim et al., 2025a;b; Yu et al., 2023), and it is an interesting direction to investigate whether their underlying principles align with our theorem.

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

# Appendix: Provable Benefit of Curriculum in Transformer Tree-Reasoning Post-Training

## CONTENTS

## A  ADDITIONAL RELATED WORK

## B  LLM USAGE

In accordance with the conference policy on large language models (LLMs), we disclose that an LLM was used solely as a writing aid to improve the clarity and readability of certain paragraphs. The LLM did not contribute to research ideation, technical development, experimental design, analysis, or interpretation of results. All scientific ideas, methods, and conclusions presented in this paper are entirely the work of the authors.

## C  ETHICS STATEMENT

This work is purely theoretical and algorithmic in nature. It does not involve human subjects, personally identifiable data, or sensitive datasets, and therefore does not pose direct risks to privacy, security, or safety. Our study is conducted under the principles of academic integrity and transparency, adhering to the ICLR Code of Ethics. We further note that our methods and results do not introduce discriminatory or harmful applications, nor are they tied to any conflicts of interest or external sponsorship that may bias the research.

## D  REPRODUCIBILITY STATEMENT

We have taken concrete steps to ensure reproducibility of our results. All theoretical claims are accompanied by formal assumptions and complete proofs in the appendix. The complexity analyses and derivations are presented in full detail, ensuring transparency and verifiability. For experimental validations (if applicable), datasets are publicly available and preprocessing steps are explicitly described in the supplementary materials. To further support reproducibility, we provide pseudo-code and detailed algorithmic specifications in Alg. 2, 3, 1, 5, 4, 6, and will release anonymous source code in the supplementary materials to facilitate independent verification.

**Theoretical benefits of curriculum in post-training**. A growing body of work has provided empirical evidence for the effectiveness of curriculum strategies during post-training (Liu et al., 2025a; Lee et al., 2025; Bae et al., 2025; Meng et al., 2025; Shi et al., 2025; Wen et al., 2025b; Zhang et al., 2025; Amani et al., 2025). Zhang et al. (2025) offered a theoretical perspective, showing that intermediate-difficulty questions yield higher signal-to-noise ratios for gradient estimates. Closely related, Liu et al. (2025b) modeled LLM reasoning as a search tree (see Figure 1) and proved that uniformly sampling a hint depth (i.e., revealing the CoT up to that depth and letting the LLM complete the rest) reduces the exploration complexity of achieving $\geq 50\%$ pass@1 from exponential to polynomial (see Remark 8). In practice, they adopted a cosine scheduler to provide longer hints in early stages and gradually shorten them during fine-tuning. Similarly, Parashar et al. (2025) analyzed curriculum within Approximate Policy Iteration: under exponentially decaying assumptions on policy-approximation error and performance loss, they proved gains in sample complexity and empirically compared uniform, cosine, and Gaussian schedulers. However, none of these studies explicitly models the difficulty measure in post-training — namely, the *pass-rate* and its dependence on the base model — nor do they consider transformer's states-conditioned tree-like autoregressive reasoning. Their results largely focus on optimization endpoints argument (i.e., $\arg\min$ solutions assuming gradient methods have converged), without analyzing the underlying post-training dynamics or the transformer architecture. In contrast, our framework is built directly on the *pass-rate* and the probabilistic tree structure of post-training reasoning, leading to Theorem 1 and its case studies on transformers.

**Theoretical benefits of curriculum learning in pre-training for parity and beyond**. Curriculum Learning (CL) was first introduced by Bengio et al. (2009), and has since been empirically validated across vision, NLP, and reinforcement learning (Graves et al., 2017; Narvekar et al., 2020; Wang et al., 2021; Soviany et al., 2022). On the theoretical side, most analyses focus on *pre-training* with highly specific problem classes or architectures. A prominent line of work studies the *parity problem*: while parity can be solved efficiently by Gaussian elimination over $\mathbb{F}_2$ (Abbe & Sandon, 2020), gradient-based neural networks struggle on dense data due to precision barriers (Abbe et al., 2021). To circumvent this, Malach et al. (2021) showed that sparse parities become learnable

with a one-layer network augmented by a parity module, while Daniely & Malach (2020) analyzed two-layer fully connected nets where sparse inputs with *leaky labels* help learning. Cornacchia & Mossel (2023) proved that sparse parities can be learned by a two-layer net with one-step gradients but require a large learning rate, and Abbe et al. (2023b) gave empirical evidence that presenting sparse-to-dense samples as a curriculum improves generalization, though without formal guarantees. In convex settings, Weinshall et al. (2018); Weinshall & Amir (2020) proved that curriculum can accelerate the convergence of SGD. Building on these observations, Abbe et al. (2023c) analyzed a two-layer ReLU network trained with noisy SGD based on mixed data curriculyum where the sparse (lower hamming weights with fewer $-1$) data is first learned, and proved that a *layer-wise curriculum* starting from sparse samples reduces both time and sample complexity. In contrast to data curricula Panigrahi et al. (2024) considered teacher-student setting where the student would benefit from teacher model's checkpoints, serving as the internal signals for students to learn parity support. Statistically, Sarao Mannelli et al. (2024) showed the negative effect of overparameterization for curriculum learning, in the case of a XOR-like Gaussian Mixture problem. Beyond parity, Saglietti et al. (2022) studied a teacher–student model where the target depends on a sparse set of features, and curriculum is defined via the variance of irrelevant features (low variance = easy, high variance = hard). More recently, Wang et al. (2025) investigated curriculum learning for transformers in the *k-fold composition task* setting, where they showed that a carefully staged pre-training schedule—both in terms of data distribution and layer-wise progression—enables efficient learning. Taken together, these works demonstrate that curriculum can provably help in pre-training, but only under tailored data curricula (e.g., sparse input with lower Hamming weight where probability of $-1$ is mild, leaky labels, checkpoints), specialized architectures (e.g., augmented by a parity modules), or non-standard optimization schemes (e.g., noisy SGD, layer-wise training). In contrast, our work studies curriculum in the *post-training* regime, where we leverage the base model's coverage power and pre-trained *chain-of-thought reasoning* ability, *without exploiting handcrafted data correlations* or requiring algorithmic modifications.

**Theoretical Benefit of CoT.**    A substantial literature has emerged on the theoretical role of chain-of-thought (CoT) in enhancing transformer models. Several works demonstrate that incorporating polynomial-length intermediate steps expands the expressive capacity of transformers beyond constant-depth architectures (Feng et al., 2023; Li et al., 2024; Merrill & Sabharwal, 2023; Wei et al., 2022). Complementing these positive results, a parallel line of research investigates intrinsic barriers, proving lower bounds or rank-based constraints that necessitate non-trivial reasoning depth (Peng et al., 2024; Barceló et al., 2025; Amiri et al., 2025). Further, statistical and compositional analyses reveal how CoT encourages structured generalization and compositional filtering, thereby clarifying its benefit from a distributional perspective (Prystawski et al., 2023; Li et al., 2023). Building on these investigations, we studied the complexity improvement when the CoT evolution is obtained in a curriculum manner.

**Training Dynamics of Transformers.**    Beyond expressiveness, recent studies examine how transformers trained with CoT evolve during optimization. Early work explored emergent in-context learning and convergence under multi-head attention (Hahn & Goyal, 2023; Huang et al., 2024; Yang et al., 2024; Zhang et al., 2024a), while subsequent analyses provide provable characterizations of induction heads and sparse token mechanisms (Chen et al., 2024). Of particular relevance are studies on CoT-specific training dynamics, which analyze nonlinear or single-layer transformers solving structured tasks such as parity (Wen et al., 2024; Kim & Suzuki, 2025; Li et al., 2025). More recent work extends these insights to multi-step optimization and regular language recognition, uncovering implicit biases and algorithmic behaviors in gradient descent (Huang et al., 2025a;b;c). Wen et al. (2025a); Kim & Suzuki (2025) investigated the learning of transformers on the parity task with CoT, building on which Yin et al. (2025) studied the impact of data shift. Yang et al. (2025) further demonstrated that transformers can implement both forward and backward *tree-structured* symbolic reasoning using two attention heads. However, these works abstract away real-world sampling process during inference, relying instead on *atypical deterministic prediction of internal tokens/sentences*, and they primarily analyze pre-training rather than post-training.

## E    CASES BEYOND TRANSFORMER GRADIENT DYNAMICS OVER TREE

**Proof.** *Proof of Theorem 1.*

*By the scaling assumption equation 3, there exist absolute constants $m, M > 0$ such that for all $k < k'$,*

$$m \left\| \frac{\pi_{k'}^\star}{\pi_k^\star} \right\|_\infty \ \le \ \mathcal{C}(\pi_{k'}^\star \mid \pi_k^\star) \ \le \ M \left\| \frac{\pi_{k'}^\star}{\pi_k^\star} \right\|_\infty.$$

Direct route. *Taking $(k, k') = (0, K)$ and invoking equation 2 yields*

$$\mathcal{C}_{direct} = \mathcal{C}(\pi_K^\star \mid \pi_0^\star) \in \Theta\left( \left\| \frac{\pi_K^\star}{\pi_0^\star} \right\|_\infty \right) = \Theta\big( (C^\star)^K \big).$$

Curriculum route. *Consider the stepwise procedure that estimates $\pi_{k+1}^\star$ from $\pi_k^\star$ for $k = 0, 1, \ldots, K - 1$. Its total cost is*

$$\mathcal{C}_{curriculum} := \sum_{k=0}^{K-1} \mathcal{C}\big( \pi_{k+1}^\star \mid \pi_k^\star \big).$$

*By equation 3 and the stepwise bound equation 1, for each $k$ we have*

$$m \, \Theta(C^\star) \ \le \ \mathcal{C}\big( \pi_{k+1}^\star \mid \pi_k^\star \big) \ \le \ M \, \Theta(C^\star),$$

*hence there exist constants $c_1, c_2 > 0$ such that*

$$c_1 \, C^\star \ \le \ \mathcal{C}\big( \pi_{k+1}^\star \mid \pi_k^\star \big) \ \le \ c_2 \, C^\star \quad \text{for all } k.$$

*Summing over $k$ gives*

$$c_1 \, K \, C^\star \ \le \ \mathcal{C}_{curriculum} \ \le \ c_2 \, K \, C^\star,$$

*i.e., $\mathcal{C}_{curriculum} = \Theta(K \, C^\star)$.*

*Combining the two parts shows that the curriculum transforms the direct exponential scaling $\Theta\big( (C^\star)^K \big)$ in task depth into a polynomial (linear) scaling $\Theta(K \, C^\star)$, as claimed.* $\qquad\square$

**Theorem 5** (Curriculum Learning with Spanner Sampling)**.** *Consider the theoretical settings in Foster et al. (2025) (realizable linear softmax parameterization), consider a curriculum of $K$ tasks with increasing difficulty, starting from a base model $\pi_{ref} := \pi_0^\star$. Suppose the sequence of optimal policies $\{\pi_k^\star\}_{k=1}^K$ satisfies assumptions in Eq.(1), (2), namely*

$$\mathcal{C}_{cov}(\pi_k^\star | \pi_{k-1}^\star) = \Theta(C^\star), \quad \forall k \in [K],$$
$$\mathcal{C}_{cov}(\pi_{k_2}^\star | \pi_{k_1}^\star) = \Theta((C^\star)^{k_2 - k_1}).$$

*For any $\epsilon > 0$ and $\delta \in (0, 1)$, by applying SpannerSampling in Foster et al. (2025) **sequentially** through the curriculum with appropriate choices of $T_{prompt}$, $T_{span}^k$, and $T_{exp}^k$ for each task, the curriculum learning algorithm learns a policy $\hat\pi$ with:*

$$\mathbb{E}_{\hat\pi_k \sim unif(\hat\pi^{(T_{exp}^{k-1})}, \ldots, \hat\pi^{(T_{exp}^k)})} \left[ J_\beta(\pi_k^\star) - J_\beta(\hat\pi_k) \right] \le \epsilon, \ k \in [K]$$

*with probability at least $1 - \delta$, and achieves the following computational efficiency bound:*

$$T_{comp}^{curriculum}(\epsilon, \delta) = \tilde{O}\left( K \cdot C^\star \cdot \frac{R_{\max}^2}{\beta^2} \Big[ \frac{R_{\max}^2}{\beta} \cdot \frac{d^2 \log^2(\delta^{-1})}{\min\{\epsilon, \beta\}} \Big] \right).$$

*Moreover, compared to direct estimation from $\pi_{ref}$ to $\pi_K^\star$, which is*

$$T_{comp}^{direct}(\epsilon, \delta) = \tilde{O}\left( (C^\star)^K \cdot \frac{R_{\max}^2}{\beta^2} \Big[ \frac{R_{\max}^2}{\beta} \cdot \frac{d^2 \log^2(\delta^{-1})}{\min\{\epsilon, \beta\}} \Big]. \right)$$

*Therefore, calling Spanner Sampling in a curriculum manner achieves exponential improvement in computational complexity:*

$$\frac{T_{comp}^{direct}}{T_{comp}^{curriculum}} = \Omega\left( \frac{(C^\star)^{K-1}}{K} \right)$$

*where the direct estimation requires $T_{comp}^{direct} = \tilde{O}\left( \mathcal{C}_{cov}(\pi^\star) \cdot \frac{R_{\max}^2}{\beta^2} \right) \cdot (C^\star)^K.$*

**Proof.** *We work under the realizable linear–softmax parameterization and oracle model of Foster et al. (2025). Let the KL–regularized objective be $J_\beta(\cdot)$ and let $R_{\max}$, $d$, and $\beta$ denote the reward bound, feature dimension, and temperature, respectively. For any pair of policies $(\pi, \pi')$, write the coverage coefficient as $C_{cov}(\pi \mid \pi') := \left| \frac{\pi}{\pi'} \right|_\infty$.*

***Step 1 (Per–task guarantee from Foster et al. (2025)).*** *By Theorem 3.1 of Foster et al. (2025), SpannerSampling, when run to target $\pi_k^\star$ starting from a reference $\pi_{k-1}^\star$ in the linear–softmax model, returns a policy $\hat{\pi}$ such that*

$$\mathbb{E}_{\hat{\pi} \sim unif(\hat{\pi}^{(1)}, \ldots, \hat{\pi}^{(T_{\exp})})}[J_\beta(\pi_k^\star) - J_\beta(\hat{\pi})] \leq \epsilon$$

*with probability at least $1 - \delta'$, using a number of oracle computations*

$$T_{comp}^{(k)}(\epsilon, \delta') = \tilde{O}\Big(C_{cov}(\pi_k^\star \mid \pi_{k-1}^\star) \cdot \underbrace{\frac{R_{\max}^2}{\beta^2}\Big[\frac{R_{\max}^2}{\beta} \cdot \frac{d^2 \log^2((\delta')^{-1})}{\min\{\epsilon, \beta\}}\Big]^2}_{=: \, \Gamma(\epsilon, \delta')}\Big),$$

*where $\tilde{O}(\cdot)$ hides polylogarithmic factors not depending on the coverage coefficient.[3] We emphasize that data efficiency $T_{data}$ is independent of coverage, while computational efficiency scales with coverage; SpannerSampling matches the linear-in-coverage lower bound.*

***Step 2 (Scheduling across a curriculum).*** *Apply Step 1 sequentially for tasks $k = 1, 2, \ldots, K$, using fresh budgets $(T_{prompt}, T_{span}^k, T_{exp}^k)$ at each stage, and set the per–stage failure probability to $\delta' = \delta/K$. The uniform–over–iterates evaluation in the statement mirrors the "return a uniformly random iterate" prescription in Foster et al. (2025), so the stagewise accuracy guarantee transfers verbatim. A union bound then yields that, simultaneously for all $k \in [K]$,*

$$\mathbb{E}_{\hat{\pi}_k \sim unif(\hat{\pi}^{(T_{\exp}^{k-1})}, \ldots, \hat{\pi}^{(T_{\exp}^k)})}[J_\beta(\pi_k^\star) - J_\beta(\hat{\pi}_k)] \leq \epsilon$$

*with probability at least $1 - \delta$. The total compute is the sum of stagewise costs:*

$$T_{comp}^{curriculum}(\epsilon, \delta) = \sum_{k=1}^K T_{comp}^{(k)}(\epsilon, \delta/K) = \tilde{O}\Big(\Gamma(\epsilon, \delta/K) \cdot \sum_{k=1}^K C_{cov}(\pi_k^\star \mid \pi_{k-1}^\star)\Big).$$

***Step 3 (Using the coverage structure of the curriculum).*** *By the curriculum assumptions (1)–(2), we have $C_{cov}(\pi_k^\star \mid \pi_{k-1}^\star) = \Theta(C^\star)$ for each $k$, hence*

$$T_{comp}^{curriculum}(\epsilon, \delta) = \tilde{O}\big(K \, C^\star \cdot \Gamma(\epsilon, \delta/K)\big).$$

*On the other hand,* direct *estimation of $\pi_K^\star$ from $\pi_{ref} = \pi_0^\star$ is controlled by the single coverage factor $C_{cov}(\pi_K^\star \mid \pi_0^\star) = \Theta((C^\star)^K)$ (by 2), and Theorem 3.1 gives*

$$T_{comp}^{direct}(\epsilon, \delta) = \tilde{O}\big((C^\star)^K \cdot \Gamma(\epsilon, \delta)\big).$$

***Step 4 (Comparing the two strategies).*** *Combining the above displays yields*

$$\frac{T_{comp}^{direct}(\epsilon, \delta)}{T_{comp}^{curriculum}(\epsilon, \delta)} = \Omega\Big(\frac{(C^\star)^{K-1}}{K}\Big),$$

*i.e., an exponential-to-polynomial improvement as a function of $K$, as claimed. This recovers the abstract form of Theorem. 1 with the concrete instantiation of the per–stage computational complexity provided by Foster et al. (2025).* $\qquad\square$

---

[3]We only need that $T_{comp}$ is (i) polynomial in $(d, \beta^{-1}, \epsilon^{-1}, \log(1/\delta'))$ and (ii) *linear* (up to logs) in $C_{cov}$, as established by Theorem 3.1 together with the matching lower bound in Foster et al. (2025).

**Remark 6** (Exponential–Polynomial Gaps Under Relaxed Conditions). *The exponential-to-polynomial separation in Theorem 5 remains valid under relaxed coverage conditions, analogous to the discussion after Theorem 1. Specifically, suppose that: (i) for some $k \in [K]$, the per-step coverage is only polynomial rather than constant, i.e.,*

$$C_{cov}(\pi_k^\star \mid \pi_{k-1}^\star) = (C^\star)^{p_k}, \quad 1 \le p_k \ll K,$$

*so that the per-task computational complexity grows as $\tilde{O}((C^\star)^{p_k} \cdot \Gamma(\epsilon, \delta))$; (ii) the direct coverage factor accumulates less than $(C^\star)^K$, relaxing to*

$$C_{cov}(\pi_K^\star \mid \pi_0^\star) = \Theta\big((C^\star)^{bK-c}\big), \quad 1 \le b \ll K, \ c \ll K.$$

*Then, under Spanner Sampling, the curriculum complexity satisfies*

$$T_{comp}^{curriculum} = \tilde{O}\Big( K \cdot \max_{k \in [K]} (C^\star)^{p_k} \cdot \Gamma(\epsilon, \delta) \Big) \ll \tilde{\Theta}\big((C^\star)^{bK-c} \cdot \Gamma(\epsilon, \delta)\big) = T_{comp}^{direct}.$$

*Thus, even under relaxed conditions, the curriculum strategy with Spanner Sampling retains an exponential advantage in computational complexity compared to direct estimation.*

**Remark 7** (Relation to E2H (CRL) theory and limits for post-training). *Parashar et al. (2025) analyze curriculum RL for LLMs under an Approximate Policy Iteration (API) lens and derive a stagewise sample complexity bound (their Thm. 3.2). In particular, letting $\epsilon_k$ be per–stage accuracy targets and $L_k$ distribution–sensitivity constants, they obtain $M_{CRL} = \sum_{k=1}^{K} \tilde{O}\big( \log^3(1/\epsilon_k)\, \epsilon_k^{-2}\, L_k^2 \cdots \big)$ and compare it to direct learning via a "curriculum efficiency factor" (CEF). Under geometric schedules $\epsilon_k = \epsilon_K\, e^{K-k}$ and $L_k = L_K/l^{K-k}$, they show*

$$M_{CRL} < M_{Direct} \iff \frac{(e\, l)^{2(1-K)} - 1}{1 - (e\, l)^2} < m - 1,$$

*where $m > 1$ encodes the relative hardness of direct learning (their Eq. (2)). Conceptually, this captures that a well–designed curriculum can turn a multiplicative blow-up across stages into a controlled (geometric) sum. While Parashar et al. (2025) do motivate curricula by a* distribution *gap between the pretraining distribution $d_0$ and task distribution $d_K$, their API analysis does not instantiate a density-/likelihood-ratio–type* coverage *term nor a dependence on the base model beyond the abstract factor $m$. Hence their theory, though consonant with ours at a high level (curricula beat direct learning), does not yet provide a coverage–based explanation of* post-training *that connects "good foundations" (a strong base policy) to concrete computational savings; Theorem 5 fills precisely this gap by making the dependence on coverage explicit.*

**Remark 8** (Relation to UFT complexity measure). *Liu et al. (2025b) propose a complexity notion tailored to unified finetuning: to achieve a 50% pass@1 success rate, the algorithm must explore at least a certain number of nodes in the search space $S_H$. Under this definition, they show that without a curriculum, the exploration complexity can grow exponentially with task depth, whereas curriculum strategies help avoid such exponential bottlenecks. Conceptually, this aligns with our message in Theorem 3, namely that curricula convert exponential costs into polynomial ones.*

*However, their measure does not explicitly account for (i) the number and structure of subtasks, (ii) the relative difficulty of these subtasks, or (iii) the role of a* coverage *coefficient that quantifies how well a base model supports subsequent targets. Moreover, the UFT framework does not yield a sample–complexity characterization directly, but instead works with node–exploration counts in $S_H$. In contrast, Theorem 3 provides a coverage–based explanation of post-training, connecting subtask decomposition and curriculum schedules to concrete sample–complexity bounds. This makes our result more directly interpretable in terms of the efficiency of real post-training pipelines.*

**Remark 9** (Relation to curriculum via sparse→mixed training in ReLU SGD). *Abbe et al. (2023c) study a* model-specific *curriculum for learning $k$-parities with a two-layer ReLU network trained by (noisy) gradient descent. Their curriculum is defined by an* ordering of input distributions*: the network is first trained on* sparse *inputs, then on the full* mixed *distribution $D_{\mathrm{mix}} = \rho D_\mu + (1 - \rho)D_u$, rather than by changing the task or exploiting a foundation model. Under bounded learning rates and a small fraction of sparse inputs, they prove a* separation in the number of training steps*: curriculum noisy-GD/SGD (sparse-first) learns $k$-parities in $\tilde{O}(d)$ (or $\tilde{O}(d)/\epsilon^2$ under accuracy $\epsilon$), whereas training on randomly ordered (mixed) samples requires at least polynomially more steps (e.g., $\tilde{\Omega}(d^{1+\delta})$ or $\tilde{\Omega}(d^2)$ in stated regimes).*

*Conceptually, this differs from Theorem 3. Our result is a* post-training *analysis in the realizable linear–softmax setting, where a* coverage coefficient $C_{cov}(\pi_k^\star \mid \pi_{k-1}^\star)$ *makes explicit how a good base policy (pretraining/foundation) reduces the computational complexity (e.g. Theorem. 5) from $\tilde{O}((C^\star)^K)$ (direct) to $\tilde{O}(KC^\star)$ (curriculum). In contrast, (Abbe et al., 2023c) do not model pretraining or any coverage/likelihood-ratio notion measuring proximity to the target policy; their curriculum is instead an* input-scheduling *device inside noisy-GD for a specific ReLU architecture. Thus, while both works support the high-level message that curricula can overcome training bottlenecks, (Abbe et al., 2023c) provide a step-complexity separation for a fixed neural model trained from scratch, whereas Theorem 5 offers a* coverage-based *explanation of post-training improvements tied to the quality (e.g. reasoning CoT power per Theorem. 3) of the base model.*

## F  Auxiliary Lemmas

**Classical probability tools for fixed-confidence identification.** We collect standard concentration and information-theoretic lemmas used in Step 6 of Prop. 1.

**Lemma 1** (Hoeffding/Chernoff inequality for Bernoulli means (Wainwright, 2019))**.** *Let $X_1, \ldots, X_m$ be i.i.d. Bernoulli(p), and $\hat{p}_m := \frac{1}{m} \sum_{r=1}^{m} X_r$. Then for any $\epsilon \in (0,1)$,*

$$\mathbb{P}\big(|\hat{p}_m - p| \geq \epsilon\big) \leq 2 \exp\big(-2m\epsilon^2\big).$$

*Equivalently, to ensure $\mathbb{P}(|\hat{p}_m - p| \geq \epsilon) \leq \delta$ it suffices that $m \geq \frac{1}{2\epsilon^2} \log \frac{2}{\delta}$.*

**Proof.***See Hoeffding (1963), Chernoff (1952), or Exercise 2.9.(a) in Wainwright (2019).* $\square$

**Lemma 2** (Two-arm fixed-confidence identification lower bound)**.** *Consider two Bernoulli arms with means $p_\star$ and $p$, and gap $\Delta := p_\star - p > 0$. Any (possibly adaptive) $\delta$-correct procedure that outputs the better arm with probability at least $1 - \delta$ must satisfy*

$$\mathbb{E}[T] \geq c \frac{\log(1/\delta)}{\text{KL}(\text{Bern}(p_\star) \,\|\, \text{Bern}(p))} \geq c' \, \Delta^{-2} \log(1/\delta),$$

*for universal constants $c, c' > 0$, where the second inequality uses Pinsker's bound $\text{KL}(\text{Bern}(p_\star) \,\|\, \text{Bern}(p)) \leq 2\,(p_\star - p)^2$.*

**Proof.***See Thm. 5 in Mannor & Tsitsiklis (2004) and Thm. 1 in Kaufmann et al. (2016).* $\square$

**Corollary 2** (Per-depth best-arm identification complexity)**.** *Fix depth $\ell$ with $|\mathcal{I}_\ell| = \Theta(d)$ Bernoulli arms and assume a unique best arm with acceptance-gap $\Delta > 0$ to all others. Any $\delta$-correct identification at depth $\ell$ requires at least*

$$\Omega\big(\Delta^{-2} \log(d/\delta)\big)$$

*oracle observations in expectation.*

**Proof.***This follows by applying Lemma 2 to each suboptimal-vs-best pair and distributing the confidence via union bound (or, sufficiency with uniform sampling via Lemma 1).* $\square$

## G  Details and Proofs of Autoregressive Reasoning Tree

**Definition 3** (Full Version of 2-States Conditioned Autoregressive Reasoning Tree (2S-ART))**.** *Let $[K] := \{1, \ldots, K\}$ be the dictionary and let $\text{EOS} := K{+}1$. Fix a depth $L \in \mathbb{N}$. A 2S-ART is specified by maps*

$$\{\Phi_l\}_{l=1}^{L}, \qquad \{\mathcal{I}_l\}_{l=1}^{L+1},$$

*and induces a function class $\mathcal{F}_{\text{2S-ART}}$ of autoregressive tasks $f : [K]^d \to [K{+}1]^{\leq L+1}$. Given an input $\mathbf{x} = (x_1, \ldots, x_d) \sim \text{Unif}([K]^d)$, define the output chain-of-thought (CoT) by the following iterative rule.*

***State and selectors.*** *For step $l = 1, 2, \ldots$, let*

$$\mathbf{CoT}_{l-1} := (x_1, \ldots, x_d, \text{EOS}, z_1, \ldots, z_{l-1}) \in [K{+}1]^{d+l}, \quad z_0 := \text{EOS}.$$

*The legal index selector $\mathcal{I}_l : [K+1]^{d+l-1} \to 2^{[d+l]}$ returns a set $\mathcal{I}_l(\mathbf{CoT}_{l-1}) \subseteq [d+l]$ with $|\mathcal{I}_l(\mathbf{CoT}_{l-1})| = \Theta(d)$ for all $l \leq L$, and the terminal selector satisfies $\mathcal{I}_{L+1}(\cdot) = \{d+1\}$. At step $l$, choose an index $i_l \in \mathcal{I}_l(\mathbf{CoT}_{l-1})$ and let $v_l := (\mathbf{CoT}_{l-1})_{i_l} \in [K+1]$.*

***Two-state update.*** *Each update is computed from the previous reasoning state and the chosen clue:*

$$\Phi_l : [K+1] \times [K+1] \to [K+1], \qquad z_l := \Phi_l(z_{l-1}, v_l), \quad l = 1, 2, \ldots$$

*with the EOS-absorbing property*

$$\Phi_l(z, \text{EOS}) = \text{EOS} \quad \text{for all } z \in [K+1], \ l = 1, \ldots, L.$$

*Hence, once $v_l = \text{EOS}$, all subsequent states remain EOS, denoting the end of the output.*

***Output.*** *Let $\ell := \min\{l \leq L : z_l = \text{EOS}\}$ (with the convention $\ell = L$ if no such $l$ exists). The task $f$ outputs the EOS-terminated CoT*

$$f(\mathbf{x}) = (z_1, \ldots, z_\ell, \text{EOS}).$$

*Because $|\mathcal{I}_l(\mathbf{CoT}_{l-1})| = \Theta(d)$ for $l \leq L$, each step selects from $\Theta(d)$ possible clues, giving a tree with branching $\Theta(d)$.*

***Subtask family.*** *For a target task $f_{S_\star} \in \mathcal{F}_{\text{2S-ART}}$ with realized index path $S_\star = (i_1, \ldots, i_{k^\star}, d+1)$, define prefix paths $S_\star^m := (i_1, \ldots, i_m, d+1)$ for $m = 1, \ldots, k^\star$. The associated subtask family is*

$$\mathcal{F}_{S_\star} := \{f_{S_\star^1}, \ldots, f_{S_\star^{k^\star}}\} \subset \mathcal{F}_{\text{2S-ART}},$$

*where each $f_{S_\star^m}$ is induced by the same $\{\Phi_l\}, \{\mathcal{I}_l\}$ but terminates immediately after choosing the $m$-th index (the next choice is forced to $d+1$).*

## G.1 PROOF OF REPRESENTATION THEOREM

**Proof.** *Proof of Corollary 1. At each depth $t = 1, \ldots, l$, the probability of selecting the unique legal child consistent with $S_\star^t$ is $\Theta(d^{-1})$ by the 2S-ART definition (near-uniform over $|\mathcal{I}_t| = \Theta(d)$ legal indices). After producing $z_l$, the next step must legally select EOS to terminate, which occurs with probability $\Theta(d^{-1})$. By the chain rule of conditional probabilities along the unique legal path, the total probability is the product of $(l+1)$ factors $\Theta(d^{-1})$, i.e., $\Theta(d^{-(l+1)})$.* □

**Proof.** *Proof of Theorem 2. Under the above assumption on $\text{FFN}_l$, it remains to realize the PART index-sampling policy via attention. It suffices to choose parameters satisfying*

$$\mathbf{K}^\top \mathbf{Q} = \begin{pmatrix} \mathbf{0}_{d_X \times d_X} & \mathbf{0}_{d_X \times (d_E - d_X)} \\ \mathbf{0}_{(d_E - d_X) \times d_X} & \mathbf{W} \end{pmatrix}, \qquad \mathbf{V} = (\mathbf{I}_{d_X} \quad \mathbf{0}), \tag{11}$$

*so that $\mathbf{V}\,\mathbf{E}[x_m] = p_m$ (position only) for any token $x_m$, and attention logits depend only on positional encodings: $\mathbf{E}[u]^\top \mathbf{K}^\top \mathbf{Q}\,\mathbf{E}[z_{l-1}] = \mathbf{p}_k^\top \mathbf{W}\,\mathbf{p}_{d+1+l}, \ \mathbf{p}_{d+1+l} := \mathbf{p}_{i_l}$ if $u$ is at position $k$. Moreover, for each $l$ choose constants $c_l > 0$ and impose a legality mask that sets logits of all $k \notin \mathcal{I}_l(z_{<l})$ to $-\infty$, while ensuring*

$$\mathbf{p}_k^\top \mathbf{W}\,\mathbf{p}_{d+1+l} = c_l \quad \text{for all } k \in \mathcal{I}_l(z_{<l}). \tag{12}$$

*Because $\mathbf{p}_i \perp \mathbf{U}$ and the block structure of $\mathbf{K}^\top \mathbf{Q}$ removes content–content and content–position interactions, the attention score from any memory token at position $k$ to the query $\mathbf{E}[z_{l-1}]$ depends only on positions: $s(k, l) = \mathbf{p}_k^\top \mathbf{W}\,\mathbf{p}_{d+1+l}$. By construction and the legality mask, $s(k, l) = c_l$ for $k \in \mathcal{I}_l(z_{<l})$ and $s(k, l) = -\infty$ otherwise (causal masking already suppresses all positions beyond $d+l$). Hence the softmax distribution over legal keys is exactly uniform.*

*With $\mathbf{V} = [\mathbf{I}_{d_X} \ \mathbf{0}]$, values contribute only content: for any legal position $k$, $\mathbf{V}\,\mathbf{E}[v_l(k)] = \boldsymbol{\mu}^{v_l(k)}$. Uniform attention over the legal set yields*

$$\hat{\mathbf{z}}_l = \frac{1}{|\mathcal{I}_l(z_{<l})|} \sum_{k \in \mathcal{I}_l(z_{<l})} \boldsymbol{\mu}^{v_l(k)}.$$

*Choose $\mathbf{U}$ with orthonormal columns. Then for any $w$ in the legal set, $\hat{\mathbf{z}}_l^\top \boldsymbol{\mu}^{v_l(w)} = |\mathcal{I}_l(z_{<l})|^{-1}$ and for any token outside the legal set the inner product is $0$. Therefore $\text{softmax}(\hat{\mathbf{z}}_l^\top \mathbf{U}/\beta)$ places*

*equal mass on legal content tokens and zero elsewhere, matching the PART policy $\Pr(i_l \mid z_{<l}) = |\mathcal{I}_l(z_{<l})|^{-1}$. By assumption, for each depth $l$ there exists $\mathrm{FFN}_l$ such that $\mathrm{FFN}_l \left( \boldsymbol{\mu}^{v_l(i_l)}, \mathbf{E}[z_{l-1}]_{:d_X} \right)$ implements $\Phi_l(z_{l-1}, v_l(i_l))$ on embeddings, yielding $\mathbf{E}[z_l] = [\boldsymbol{\mu}^{z_l}, \mathbf{p}_{d+1+l}]^\top$. At $l=L$, set the legality mask to permit only $k = d+1$ (EOS), which forces termination as in Definition 1. The construction exactly replicates PART.* $\qquad\square$

## G.2 Proof of Finetuning Algorithms

**Lemma 3.** *Fix a step $l$. Define masked attention logits and weights:*

$$s_l(j) := \mathbf{p}_j^\top \mathbf{W}\, \mathbf{p}_{c_l} + m_l(j), \quad \alpha_l(j) := \frac{\exp(s_l(j))}{\sum_{q \in \mathcal{I}_l} \exp(s_l(q))}, \quad j \in \mathcal{I}_l,$$

*where $m_l$ encodes causal/legal masking and is independent of $\mathbf{W}$. Define the attention-weighted token vector and vocabulary logits:*

$$\hat{\mathbf{p}}_{l+1}^{\mathrm{att}} := \sum_{j \in \mathcal{I}_l} \alpha_l(j)\, \mathbf{p}_j, \qquad \ell_r := \frac{\langle \hat{\mathbf{p}}_{l+1}^{\mathrm{att}}, \mathbf{p}_r \rangle}{\beta}, \ \ r \in \{1, ..., d+1\}.$$

*Let $p_{\mathrm{vocab}}(r) := \exp(\ell_r)/\sum_q \exp(\ell_q)$, and define the index policy as the probability of sampling the token that matches the $i_l$-th input value:*

$$\pi_{\mathbf{W}}(i_l \mid \mathbf{x}, \hat{\mathbf{p}}^{z_{1:l}}) := p_{\mathrm{vocab}}\big(u = \boldsymbol{\mu}_{x_{i_l}} \mid \hat{\mathbf{p}}_{l+1}^{\mathrm{att}}\big).$$

*Then the score admits the positional outer-product decomposition*

$$\nabla_{\mathbf{W}} \log \pi_{\mathbf{W}}(i_l \mid \cdot) = \sum_{k \in \mathcal{I}_l} \alpha_l(k)\, \eta_l(k)\, \mathbf{p}_k \mathbf{p}_{c_l}^\top, \quad \eta_l(k) := \frac{\left\langle \mathbf{p}_k - \hat{\mathbf{p}}_{l+1}^{\mathrm{att}},\, \mathbf{p}_{i_l} - \sum_r p_{\mathrm{vocab}}(r) \mathbf{p}_r \right\rangle}{\beta}.$$

*Moreover, under Eq. (12) (positional orthogonality; $\mathbf{W}$ acts only on positional blocks; the mask $m_l$ is $\mathbf{W}$-independent) and a finetuning regime that only reweights legal transitions within these blocks, the family $\{\mathbf{p}_k \mathbf{p}_{c_l}^\top\}$ forms an orthogonal basis for the reachable score directions. Consequently, $\nabla_{\mathbf{W}} \log \pi_{\mathbf{W}}$ decomposes uniquely onto orthogonal tensor blocks $\mathrm{span}\{\mathbf{p}_k\} \otimes \mathrm{span}\{\mathbf{p}_{c_l}\}$ (Orthogonal Block Isolation).*

**Proof.** *We expand every ingredient with step-by-step derivations.*

*(1) Vocabulary softmax – definition and gradient. Fix $l$, and define the normalized vocabulary probabilities:*

$$p_{\mathrm{vocab}}(r) = \frac{\exp(\ell_r)}{\sum_q \exp(\ell_q)}, \qquad \ell_r = \frac{\langle \hat{\mathbf{p}}_{l+1}^{\mathrm{att}}, \mathbf{p}_r \rangle}{\beta},$$

*with classes $r \in \{0, 1, \mathrm{EOS}\}$. The policy is*

$$\pi_{\mathbf{W}}(i_l \mid \cdot) = p_{\mathrm{vocab}}(x_{i_l}) := p_{\mathrm{vocab}}\big(u = \boldsymbol{\mu}_{x_{i_l}} \mid \hat{\mathbf{p}}_{l+1}^{\mathrm{att}}\big).$$

*We differentiate $\log p_{\mathrm{vocab}}(x_{i_l})$ w.r.t. $\hat{\mathbf{p}}_{l+1}^{\mathrm{att}}$:*

$$\nabla_{\hat{\mathbf{p}}_{l+1}^{\mathrm{att}}} \log p_{\mathrm{vocab}}(x_{i_l}) \overset{(1.1)}{=} \nabla_{\hat{\mathbf{p}}_{l+1}^{\mathrm{att}}} \left( \ell_{x_{i_l}} - \log \sum_r e^{\ell_r} \right) \tag{13}$$

$$\overset{(1.2)}{=} \frac{\mathbf{p}_{i_l}}{\beta} - \sum_r \frac{e^{\ell_r}}{\sum_q e^{\ell_q}} \frac{\mathbf{p}_r}{\beta} \tag{14}$$

$$\overset{(1.3)}{=} \frac{\mathbf{p}_{i_l} - \sum_r p_{\mathrm{vocab}}(r)\, \mathbf{p}_r}{\beta} \ :=: \ \mathbf{g}_l^{\mathrm{vocab}}. \tag{15}$$

*Step (1.1) expands the log-softmax; (1.2) uses $\partial \ell_r / \partial \hat{\mathbf{p}}_{l+1}^{\mathrm{att}} = \mathbf{p}_r / \beta$; (1.3) recognizes $p_{\mathrm{vocab}}$.*

*(2) Attention softmax – Jacobian and gradient w.r.t. $\mathbf{W}$. The masked logits and weights are*

$$s_l(k) = \mathbf{p}_k^\top \mathbf{W}\, \mathbf{p}_{c_l} + m_l(k), \qquad \alpha_l(k) = \frac{e^{s_l(k)}}{\sum_{q \in \mathcal{I}_l} e^{s_l(q)}}.$$

The softmax Jacobian is $\partial\alpha_l(j)/\partial s_l(k) = \alpha_l(j)(\delta_{jk} - \alpha_l(k))$. Since $m_l$ is $\mathbf{W}$-independent,

$$\nabla_{\mathbf{W}} s_l(k) = \nabla_{\mathbf{W}}\left(\mathbf{p}_k^\top \mathbf{W}\, \mathbf{p}_{c_l}\right) = \mathbf{p}_k \mathbf{p}_{c_l}^\top.$$

Thus the gradient of the attention-weighted token vector is:

$$\nabla_{\mathbf{W}} \hat{\mathbf{p}}_{l+1}^{\mathrm{att}} = \nabla_{\mathbf{W}}\Big( \sum_{j \in \mathcal{I}_l} \alpha_l(j)\mathbf{p}_j \Big) \tag{16}$$

$$\stackrel{(2.1)}{=} \sum_j \mathbf{p}_j \sum_k \frac{\partial\alpha_l(j)}{\partial s_l(k)}\, \nabla_{\mathbf{W}} s_l(k) \tag{17}$$

$$\stackrel{(2.2)}{=} \sum_j \mathbf{p}_j \sum_k \alpha_l(j)\big(\delta_{jk} - \alpha_l(k)\big)\, \mathbf{p}_k \mathbf{p}_{c_l}^\top \tag{18}$$

$$\stackrel{(2.3)}{=} \sum_k \alpha_l(k)\Big(\mathbf{p}_{x_k} - \sum_j \alpha_l(j)\mathbf{p}_j\Big)\, \mathbf{p}_k \mathbf{p}_{c_l}^\top \tag{19}$$

$$\stackrel{(2.4)}{=} \sum_k \alpha_l(k)\big(\mathbf{p}_k - \hat{\mathbf{p}}_{l+1}^{\mathrm{att}}\big)\, \mathbf{p}_k \mathbf{p}_{c_l}^\top. \tag{20}$$

Step (2.1) is the chain rule; (2.2) uses the softmax Jacobian and $\nabla_{\mathbf{W}} s_l$; (2.3) collects terms; (2.4) recognizes $\hat{\mathbf{p}}_{l+1}^{\mathrm{att}}$.

(3) Chain rule for the policy score. Combining (1)–(2),

$$\nabla_{\mathbf{W}} \log \pi_{\mathbf{W}}(i_l \mid \cdot) \stackrel{(3.1)}{=} \left(\nabla_{\hat{\mathbf{p}}_{l+1}^{\mathrm{att}}} \log p_{\mathrm{vocab}}(x_{i_l})\right)^\top \cdot \nabla_{\mathbf{W}} \hat{\mathbf{p}}_{l+1}^{\mathrm{att}} \tag{21}$$

$$\stackrel{(3.2)}{=} \sum_{k \in \mathcal{I}_l} \alpha_l(k) \underbrace{\left\langle \mathbf{p}_k - \hat{\mathbf{p}}_{l+1}^{\mathrm{att}},\, \mathbf{g}_l^{\mathrm{vocab}} \right\rangle}_{:=\, \eta_l(k)} \mathbf{p}_k \mathbf{p}_{c_l}^\top. \tag{22}$$

This yields the claimed positional outer-product decomposition with weights $\alpha_l(k)\eta_l(k)$, where $\mathbf{g}_l^{\mathrm{vocab}} = (\mathbf{p}_{i_l} - \sum_r p_{\mathrm{vocab}}(r)\mathbf{p}_r)/\beta$.

(4) Interchange and smoothness conditions. The above steps use: (i) $m_l$ is $\mathbf{W}$-independent, so $\nabla_{\mathbf{W}} s_l$ exists and is continuous; (ii) softmax is $C^\infty$, thus $\alpha_l$ and $p_{\mathrm{vocab}}$ are smooth in $\mathbf{W}$; (iii) boundedness of token embeddings $\mathbf{p}_\cdot$, vocabulary $\mathbf{U}$, and temperature $\beta > 0$ ensures an $L^1$ dominator for Leibniz interchange when taking expectations over trajectories (used later in policy-gradient proofs).

(5) Orthogonal Block Isolation (OBI) and the role of Eq. (12). Under Eq. (12): positional embeddings $\{\mathbf{p}_j\}$ are mutually orthogonal; $\mathbf{K}^\top \mathbf{Q}$ (hence $\mathbf{W}$) acts only on positional blocks; $\mathbf{V}$ projects out positional-channel–orthogonal components when forming logits with $\mathbf{U}$. Finetuning reweights only the legal transitions while preserving the block structure (the mask $m_l$ is fixed). Therefore, the reachable score directions lie in the span of $\{\mathbf{p}_k \mathbf{p}_{c_l}^\top\}$, and for any $(k, l) \neq (k', l')$,

$$\left\langle \mathbf{p}_k \mathbf{p}_{c_l}^\top,\, \mathbf{p}_{k'} \mathbf{p}_{c_{l'}}^\top \right\rangle = (\mathbf{p}_k^\top \mathbf{p}_{k'})(\mathbf{p}_{c_l}^\top \mathbf{p}_{c_{l'}}) = 0.$$

Hence cross-terms vanish and the decomposition in (3.2) is unique on the block-diagonal positional subspace $\mathrm{span}\{\mathbf{p}_k\} \otimes \mathrm{span}\{\mathbf{p}_{c_l}\}$. This establishes OBI with Eq. (12) as sufficient conditions. In practice, these are met at initialization (masking, orthogonality, block action), and finetuning that only reweights legal transitions preserves the required block-isolation. $\qquad\square$

**Lemma 4** (From per-step logit margins to overall 0-1 loss). Fix a target subtask family $\mathcal{F}_{S_\star} = \{f_{S_\star^1}, \ldots, f_{S_\star^{k^\star}}\}$ and consider decoding $k^\star + 1$ steps (the last for EOS). For each step $l \in \{1, \ldots, k^\star + 1\}$, let the legal index set be $\mathcal{I}_l$ with cardinality $d_l := |\mathcal{I}_l| \geq 1$, and denote the correct index by $i_l$ (with $i_{k^\star+1} := d+1$ for EOS). Define masked attention logits $s_l(j) := \mathbf{p}_j^\top \mathbf{W}\, \mathbf{p}_{c_l} + m_l(j)$ and attention weights $\alpha_l(j) = \mathrm{softmax}_j(s_l(j))$. Let

$$\Delta_l := s_l(i_l) - \max_{j \in \mathcal{I}_l \setminus \{i_l\}} s_l(j) \geq 0$$

be the per-step attention logit gap on the correct child. Assume pairwise-orthogonal token embeddings, bounded norms, and temperature $\beta > 0$ so that

$$\left|\eta_l(k)\right| \leq \frac{4}{\beta}, \qquad \text{and} \qquad p_{\mathrm{vocab}}(r) = \mathrm{softmax}_r\left(\langle \hat{\mathbf{p}}_{l+1}^{\mathrm{att}}, \mathbf{p}_r \rangle/\beta\right)$$

*as in Lemma 3. Let $M := K + 1$ denote the vocabulary size. For any error budget $(\varepsilon_1, \ldots, \varepsilon_{k^\star+1}) \in (0,1)^{k^\star+1}$, define the per-step required attention probability and corresponding margin threshold by*

$$\alpha_l^{\mathrm{req}} := \tfrac{1}{2} + \tfrac{\beta}{2} \log\Big(\tfrac{M(1-\varepsilon_l)}{\varepsilon_l}\Big), \qquad \Gamma_l := \log\Big(\tfrac{d_l-1}{(\alpha_l^{\mathrm{req}})^{-1}-1}\Big),$$

*whenever $\alpha_l^{\mathrm{req}} < 1$ (otherwise the requirement is infeasible).*

*If the per-step logit gaps satisfy $\Delta_l \geq \Gamma_l$ for all $l \in [k^{star}+1]$, then the per-step selection probability of the correct token obeys*

$$\pi_l := p_{\mathrm{vocab}}\big(u = \mathbf{p}_{i_l} \mid \hat{\mathbf{p}}_{l+1}^{\mathrm{att}}\big) \geq 1 - \varepsilon_l, \quad l = 1, \ldots, k^\star + 1.$$

*Consequently, letting the 0-1 loss be $L := 1 - R^{k^\star}(\mathrm{TF}(\cdot; \mathbf{W}))$ (so that $\mathbb{E}[L] = 1 - \mathbb{E}[R^{k^\star}]$), we have*

$$\mathbb{E}_{\mathbf{x} \sim \mathrm{Unif}([K]^d)}\big[L\big] \leq \sum_{l=1}^{k^\star+1} \varepsilon_l.$$

*In particular, choosing any $(\varepsilon_l)$ with $\sum_l \varepsilon_l \leq \varepsilon$ implies $\mathbb{E}[L] \leq \varepsilon$.*

**Proof.** *Step 1 (attention probability from logit gap). For $\alpha_l(\cdot) = \mathrm{softmax}(s_l(\cdot))$ and gap $\Delta_l$, the correct-child attention weight obeys the standard softmax bound*

$$\alpha_l(i_l) = \frac{1}{1 + \sum_{j \neq i_l} \exp\big(-(s_l(i_l) - s_l(j))\big)} \geq \frac{1}{1 + (d_l - 1)\,e^{-\Delta_l}}.$$

*Thus if $\Delta_l \geq \Gamma_l$ with $\Gamma_l$ defined in the statement, then*

$$\alpha_l(i_l) \geq \alpha_l^{\mathrm{req}}.$$

*Step 2 (vocabulary probability from attention concentration). With orthogonal embeddings, $\langle \hat{\mathbf{p}}_{l+1}^{\mathrm{att}}, \mathbf{p}_{i_l} \rangle = \alpha_l(i_l)$ and for any competitor token $r \neq x_{i_l}$, $\langle \hat{\mathbf{p}}_{l+1}^{\mathrm{att}}, \mathbf{p}_r \rangle \leq 1 - \alpha_l(i_l)$ (competitors include tokens not present in the mixture, whose inner products are $0 \leq 1 - \alpha_l$). Hence the vocabulary-logit gap between the correct token and any competitor is at least*

$$\gamma_l^{\mathrm{vocab}} \geq \alpha_l(i_l) - (1 - \alpha_l(i_l)) = 2\alpha_l(i_l) - 1.$$

*The softmax lower bound then gives*

$$\pi_l = \frac{e^{\alpha_l(i_l)/\beta}}{\sum_r e^{\langle \hat{\mathbf{p}}_{l+1}^{\mathrm{att}}, \mathbf{p}_r \rangle / \beta}} \geq \frac{1}{1 + \sum_{r \neq x_{i_l}} e^{-\gamma_l^{\mathrm{vocab}}/\beta}} \geq \frac{1}{1 + M\,e^{-(2\alpha_l(i_l)-1)/\beta}}.$$

*Therefore $\alpha_l(i_l) \geq \alpha_l^{\mathrm{req}}$ implies*

$$\pi_l \geq \frac{1}{1 + M\,e^{-(2\alpha_l^{\mathrm{req}}-1)/\beta}} = 1 - \varepsilon_l.$$

*Step 3 (from per-step success to sequence success). The sequence fails only if at least one step fails, hence by the union bound*

$$\mathbb{P}(\textit{sequence fails}) \leq \sum_{l=1}^{k^\star+1} (1 - \pi_l) \leq \sum_{l=1}^{k^\star+1} \varepsilon_l.$$

*Taking expectation over $\mathbf{x} \sim \mathrm{Unif}([K]^d)$ does not increase the bound, yielding the displayed inequality for the 0-1 loss $L = 1 - R^{k^\star}$.* $\qquad \square$

**Lemma 5** (Policy Gradients for REINFORCE). *Let the input $\mathbf{x}$ be fixed and the decoding length be $k^\star \geq 2$. Denote the generated token sequence by $\hat{\mathbf{p}}^{z_1:k^\star} = (\hat{\mathbf{p}}^{z_1}, \ldots, \hat{\mathbf{p}}^{z_{k^\star}})$. At step $l$ ($1 \leq l \leq k^\star + 1$), an attention policy first samples a secret index's token $i_l \sim \pi_{\mathbf{W}^{k^\star}}(\cdot \mid \mathbf{x}, \hat{\mathbf{p}}^{z_1:l}) := \hat{p}_{\mathbf{W}}\big(u = \boldsymbol{\mu}_{x_{i_l}} \mid \mathbf{x}, \hat{\mathbf{p}}^{z_1:l}\big)$. The next token is then deterministically produced by the deterministic Feedforward $\mathrm{FFN}_m(\cdot)$ at $m$-th reasoning procedure, so that the only source of randomness is the*

sampling of tokens $\boldsymbol{\mu}_{x_{i_l}}$ corresponding to secret index sequences $i_{1:k^\star+1}$. We therefore define the (random) trajectory as $\tau := (i_{1:k^\star+1})$ and the induced measure

$$p_{\mathbf{W}^{k^\star}}(\tau \mid \mathbf{x}) = \prod_{l=1}^{k^\star+1} \pi_{\mathbf{W}^{k^\star}}\big(i_l \mid \mathbf{x}, \hat{\mathbf{p}}^{z_{1:l}}\big),$$

where the tokens $\hat{\mathbf{p}}^{z_{1:k^\star}}$ are deterministic functions of $\tau$ via $g_{\mathbf{W}^{k^\star}}$. Let the terminal reward be $R^{k^\star}(\mathrm{TF}(\cdot; \mathbf{W}))$, which does not explicitly depend on $\mathbf{W}^{k^\star}$. Define the population loss

$$\mathcal{J}_{\mathrm{REINFORCE}}^{k^\star}(\mathbf{W}^{k^\star}) := \mathbb{E}_{\mathbf{x}\sim\mathcal{P}_x, \tau\sim p_{\mathbf{W}^{k^\star}}(\cdot|\mathbf{x})}\Big[R^{k^\star}\big(\boldsymbol{\mu}^{z_{k^\star}}(\tau_{-2})\big)\Big]. \tag{23}$$

Assume the following regularity conditions hold: (i) for all $l$, $\pi_{\mathbf{W}^{k^\star}}(i_l \mid \cdot) > 0$ is Fréchet differentiable in $\mathbf{W}^{k^\star}$ with log-smoothness; (ii) $R^{k^\star}$ is bounded and integrable; and (iii) differentiation can be interchanged with integration (or expectation) under dominated convergence / parameterized measure continuity. Denote $\mathbb{E}_\tau[\cdot] = \mathbb{E}_{\mathbf{x}\sim\mathcal{P}_x, \tau\sim p_{\mathbf{W}^{k^\star}}(\cdot|\mathbf{x})}[\cdot]$, the policy gradients are:

$$\nabla_{\mathbf{W}^{k^\star}} \mathcal{J}_{\mathrm{REINFORCE}}^{k^\star}(\mathbf{W}^{k^\star}) = \mathbb{E}_\tau\left[ R^{k^\star}\big(\boldsymbol{\mu}^{z_{k^\star}}(\tau_{-2})\big) \cdot \sum_{l=1}^{k^\star+1} \nabla_{\mathbf{W}^{k^\star}} \log \pi_{\mathbf{W}^{k^\star}}\big(i_l \mid \mathbf{x}, \hat{\mathbf{p}}^{z_{1:l}}\big) \right]. \tag{24}$$

The formula reflects that token generation is deterministic via $\phi$ while the non-Markovian dependency arises from the index policy $\pi$ depending on the full history. Therefore, by Lemma 3, we have

$$\nabla_{\mathbf{W}^{k^\star}} \mathcal{J}_{\mathrm{REINFORCE}}^{k^\star}(\mathbf{W}^{k^\star}) = \mathbb{E}_\tau\left[ R^{k^\star}\big(\boldsymbol{\mu}^{z_{k^\star}}(\tau_{-2})\big) \sum_{l=1}^{k^\star+1} \sum_{k\in\mathcal{I}_l} \alpha_l^{k^\star}(k)\, \eta_l^{k^\star}(k)\, \mathbf{p}_k \mathbf{p}_{c_l}^\top \right], \tag{25}$$

where

- $\alpha_l^{k^\star}(k) := \mathrm{softmax}_k(\mathbf{p}_k^\top \mathbf{W}^{k^\star} \mathbf{p}_{c_l} + m_l(k))$,

- $\hat{\mathbf{p}}_{l+1}^{\mathrm{att},k^\star} := \sum_{j\in\mathcal{I}_l} \alpha_l^{k^\star}(j)\mathbf{p}_j$,

- $\eta_l^{k^\star}(k) := \langle \mathbf{p}_k - \hat{\mathbf{p}}_{l+1}^{\mathrm{att},k^\star}, \mathbf{p}_{i_l} - \sum_r p_{\mathrm{vocab}}^{k^\star}(r)\mathbf{p}_r\rangle/\beta$,

- $p_{\mathrm{vocab}}^{k^\star}(\cdot) := \dfrac{\exp(\frac{\langle\hat{\mathbf{p}}_{l+1}^{\mathrm{att}},\mathbf{p}_r\rangle}{\beta})}{\sum_q \exp(\frac{\langle\hat{\mathbf{p}}_{l+1}^{\mathrm{att}},\mathbf{p}_q\rangle}{\beta})}$.

**Proof.** Write $\tau = (i_{1:k^\star+1})$ and abbreviate $\mathbb{E}_\tau$ for $\mathbb{E}_{\tau\sim p_{\mathbf{W}^{k^\star}}(\cdot|\mathbf{x})}$. The tokens $\hat{\mathbf{p}}^{z_{1:k^\star}}(\tau)$ are deterministic given $\tau$ via $g_{\mathbf{W}^{k^\star}}$.

We derive the gradient in a step-numbered manner:

$$\nabla_{\mathbf{W}^{k^\star}} \mathcal{J}_{\mathrm{REINFORCE}}^{k^\star} \overset{(1)}{=} \nabla_{\mathbf{W}^{k^\star}} \int R^{k^\star}\big(\boldsymbol{\mu}^{z_{k^\star}}(\tau_{-2})\big) p_{\mathbf{W}^{k^\star}}(\tau \mid \mathbf{x})\, d\tau \tag{26}$$

$$\overset{(2)}{=} \int R^{k^\star}\big(\boldsymbol{\mu}^{z_{k^\star}}(\tau_{-2})\big) \nabla_{\mathbf{W}^{k^\star}} p_{\mathbf{W}^{k^\star}}(\tau \mid \mathbf{x})\, d\tau \tag{27}$$

$$\overset{(3)}{=} \int R^{k^\star}\big(\boldsymbol{\mu}^{z_{k^\star}}(\tau_{-2})\big) p_{\mathbf{W}^{k^\star}}(\tau \mid \mathbf{x}) \nabla_{\mathbf{W}^{k^\star}} \log p_{\mathbf{W}^{k^\star}}(\tau \mid \mathbf{x})\, d\tau \tag{28}$$

$$\overset{(4)}{=} \mathbb{E}_\tau\left[ R^{k^\star}\big(\boldsymbol{\mu}^{z_{k^\star}}(\tau_{-2})\big) \sum_{l=1}^{k^\star+1} \nabla_{\mathbf{W}^{k^\star}} \log \pi_{\mathbf{W}^{k^\star}}\big(i_l \mid \mathbf{x}, \hat{\mathbf{p}}^{z_{1:l}}\big) \right], \tag{29}$$

which equals equation 24.

Step (1) is the definition of $\mathcal{J}_{\mathrm{REINFORCE}}^{k^\star}$; Step (2) uses the Leibniz interchange $\nabla_\theta \int f(\tau, \theta)d\tau = \int \nabla_\theta f(\tau, \theta)d\tau$ under the following conditions: (i) $R^{k^\star} \circ g_{\mathbf{W}^{k^\star}}$ is bounded; (ii) $p_{\mathbf{W}^{k^\star}}(\tau \mid \mathbf{x})$ is

*differentiable in $\mathbf{W}^{k^\star}$; (iii) there exists an integrable dominator $h(\tau)$ with $\|R^{k^\star}\nabla p\| \le h$ (dominated convergence / parameterized measure continuity). These hold because $R^{k^\star} \in \{0,1\}$ and $\pi$ is softmax-based with log-smoothness. Step (3) is the score-function identity $\nabla p = p\nabla\log p$. Step (4) applies the chain rule to $\log p_{\mathbf{W}^{k^\star}}(\tau\mid\mathbf{x}) = \sum_l \log\pi_{\mathbf{W}^{k^\star}}(i_l\mid\cdot)$, which holds regardless of parameter sharing across time. The resulting decomposition does not require Markovity of the state, since the history enters through the conditioning of $\pi$.*

*Finally, we note that deterministic $\phi$ does not affect the score terms and only enters through $R^{k^\star}\big(\boldsymbol{\mu}^{z_{k^\star}}(\tau_{-2})\big)$, preserving all interchanges above. Fubini's theorem applies whenever $\mathbb{E}_\tau\big[\,|R^{k^\star}\sum_l\nabla\log\pi_l|\,\big] < \infty$, which holds by boundedness of $R^{k^\star}$ and square-integrability (log-smoothness) of the score.*

*By Lemma 3, for each $l$ we have the score decomposition*

$$\nabla_{\mathbf{W}^{k^\star}}\log\pi_{\mathbf{W}^{k^\star}}(i_l\mid\cdot) = \sum_{k\in\mathcal{I}_l}\alpha_l^{k^\star}(k)\eta_l^{k^\star}(k)\mathbf{p}_k\mathbf{p}_{c_l}^\top.$$

*Substituting this into the REINFORCE identity*

$$\nabla\mathcal{J}_{\mathrm{REINFORCE}} = \mathbb{E}\big[R\sum_l\nabla\log\pi\big]$$

*and using linearity of expectation, yields the final expanded formula. Orthogonal Block Isolation guarantees each term lies in a unique positional tensor block $p_k p_{c_l}^T$, ensuring no cross-block interference in these expansions.* $\qquad\square$

**Lemma 6** (One-step REINFORCE update of per-step attention logit gap)**.** *Fix a step $l \in \{1,\dots,k^\star+1\}$. Define*

$$s_l(j;\mathbf{W}) := \mathbf{p}_j^\top\mathbf{W}\,\mathbf{p}_{c_l} + m_l(j), \qquad \Delta_l(\mathbf{W}) := s_l(i_l;\mathbf{W}) - \max_{j\in\mathcal{I}_l\setminus\{i_l\}}s_l(j;\mathbf{W}).$$

*Consider the one-step REINFORCE update that maximizes the reward (step size $\eta > 0$)*

$$\mathbf{W}^{(t+1)} = \mathbf{W}^{(t)} + \eta\,\nabla_{\mathbf{W}}\mathcal{J}_{\mathrm{REINFORCE}}^{k^\star}(\mathbf{W}^{(t)}),$$

*with all gradients evaluated at $\mathbf{W}^{(t)}$. Let $\mathbb{E}_\tau^{(t)}[\cdot]$ denote the trajectory expectation under the policy at $\mathbf{W}^{(t)}$ in Lemma 5, and reuse the notation of Lemma 3*

$$\alpha_l^{(t)}(j) = \mathrm{softmax}_j\big(\mathbf{p}_j^\top\mathbf{W}^{(t)}\mathbf{p}_{c_l} + m_l(j)\big), \quad \eta_l^{(t)}(j) = \frac{\langle\mathbf{p}_j - \hat{\mathbf{p}}_{l+1}^{\mathrm{att},(t)},\ \mathbf{p}_{i_l} - \sum_r p_{\mathrm{vocab}}^{(t)}(r)\mathbf{p}_r\rangle}{\beta}.$$

*Introduce the path-success and spurious-success decomposition. Let $\mathcal{T}_{\mathrm{true}}$ be the set of trajectories that select the correct indices $i_{1:k^\star}$ (and then EOS); define the true-path success probability*

$$p_{\mathrm{path}} := \mathbb{E}_\mathbf{x}\,\mathbb{P}_{\tau\mid\mathbf{x}}\big\{\tau\in\mathcal{T}_{\mathrm{true}}\big\} = \mathbb{E}_\mathbf{x}\Big[\prod_{l=1}^{k^\star+1}\pi_{\mathbf{W}^{k^\star}}\big(i_l\mid\mathbf{x},\hat{\mathbf{p}}^{z_{1:l}}\big)\Big].$$

*Let the spurious (reward-hacking) success rate be*

$$\rho_{\mathrm{spur}} := \mathbb{E}_\mathbf{x}\Big[\mathbb{P}_{\tau\mid\mathbf{x}}\big\{R^{k^\star}(\hat{\mathbf{p}}^{z_{k^\star}}(\tau)) = 1\mid\tau\notin\mathcal{T}_{\mathrm{true}}\big\}\Big],$$

*which depends on the task structure encoded by $\{\Phi_l\}, \{\mathcal{I}_l\}$. Then, for any policy,*

$$p_{\mathrm{succ}} = p_{\mathrm{path}} + (1 - p_{\mathrm{path}})\,\rho_{\mathrm{spur}}. \tag{30}$$

*In particular, at near-uniform initialization with $d_l := |\mathcal{I}_l|$ and $\pi(i_l\mid\cdot)\approx 1/d_l$ on legal children,*

$$p_{\mathrm{path}} \le \prod_{l=1}^{k^\star+1}\frac{1}{d_l}, \qquad p_{\mathrm{succ}} \le \rho_{\mathrm{spur}} + (1 - \rho_{\mathrm{spur}})\prod_{l=1}^{k^\star+1}\frac{1}{d_l}. \tag{31}$$

*For the parity case study with $\mathbf{x}\sim\mathrm{Unif}\{0,1\}^d$ and XOR kernels $\{\Phi_l\}$, any wrong index set yields the correct parity with probability $1/2$; hence*

$$\rho_{\mathrm{spur}}^{\mathrm{parity}} = \tfrac{1}{2}, \qquad p_{\mathrm{succ}} \le \tfrac{1}{2} + \tfrac{1}{2}\prod_{l=1}^{k^\star+1}\frac{1}{d_l}. \tag{32}$$

*Then:*

*1) (exact logit update for affine logits) For any $j \in \mathcal{I}_l$, we have*

$$s_l\big(j; \mathbf{W}^{(t+1)}\big) = s_l\big(j; \mathbf{W}^{(t)}\big) + \eta\, \mathbb{E}_\tau^{(t)}\big[\, R^{k^\star} \alpha_l^{(t)}(j)\, \eta_l^{(t)}(j)\,\big]. \tag{33}$$

*2) (subgradient inequality for the max) Let the active competitor set be $\mathcal{A}_l^{\max} := \arg\max_{j \neq i_l} s_l\big(j; \mathbf{W}^{(t-1)}\big)$. Then*

$$\Delta_l\big(\mathbf{W}^{(t+1)}\big) \geq \Delta_l\big(\mathbf{W}^{(t)}\big) + \eta\Big(\mathbb{E}_\tau^{(t)}[R^{k^\star} \alpha_l^{(t)}(i_l)\, \eta_l^{(t)}(i_l)] - \sup_{j \neq i_l} \mathbb{E}_\tau^{(t)}[R^{k^\star} \alpha_l^{(t)}(j)\, \eta_l^{(t)}(j)]\Big) + \mathcal{O}(\eta^2). \tag{34}$$

*If, in a neighborhood of $\mathbf{W}^{(t-1)}$, the active maximizer is unique and remains unchanged (there exists $j_l^{\max} \in \mathcal{A}_l^{\max}$ that stays the maximizer in that neighborhood), then the first-order exact form is*

$$\Delta_l\big(\mathbf{W}^{(t+1)}\big) = \Delta_l\big(\mathbf{W}^{(t)}\big) + \eta\Big(\mathbb{E}_\tau^{(t)}[R^{k^\star} \alpha_l^{(t)}(i_l)\, \eta_l^{(t)}(i_l)] - \mathbb{E}_\tau^{(t)}[R^{k^\star} \alpha_l^{(t)}(j_l^{\max})\, \eta_l^{(t)}(j_l^{\max})]\Big) + \mathcal{O}(\eta^2). \tag{35}$$

*3) (update of a smooth lower bound) Define the competitors' log-sum-exp lower bound*

$$\phi_l(\mathbf{W}) := s_l(i_l; \mathbf{W}) - \log \sum_{j \neq i_l} \exp\big(s_l(j; \mathbf{W})\big) \leq \Delta_l(\mathbf{W}).$$

*Let $\tilde{\alpha}_l(j) := \frac{\exp(s_l(j))}{\sum_{q \neq i_l} \exp(s_l(q))}$ be the softmax normalized only over competitors, evaluated at $\mathbf{W}^{(t-1)}$. Then*

$$\phi_l\big(\mathbf{W}^{(t+1)}\big) = \phi_l\big(\mathbf{W}^{(t)}\big) + \eta\Big(\mathbb{E}_\tau^{(t)}[R^{k^\star} \alpha_l^{(t)}(i_l)\, \eta_l^{(t)}(i_l)] - \sum_{j \neq i_l} \tilde{\alpha}_l^{(t)}(j)\, \mathbb{E}_\tau^{(t)}[R^{k^\star} \alpha_l^{(t)}(j)\, \eta_l^{(t)}(j)]\Big) + \mathcal{O}(\eta^2). \tag{36}$$

*Moreover, to recompute intermediate quantities after one update, the following first-order relations (Fréchet differentials at $\mathbf{W}^{(t-1)}$) hold:*

$$\delta s_l^{(t)}(k) = s_l\big(k; \mathbf{W}^{(t+1)}\big) - s_l\big(k; \mathbf{W}^{(t)}\big) = \eta\, \mathbb{E}_\tau^{(t)}[R^{k^\star} \alpha_l^{(t)}(k)\, \eta_l^{(t)}(k)], \tag{37}$$

$$\delta \alpha_l^{(t)}(k) = \alpha_l^{(t)}(k)\Big(\delta s_l(k) - \sum_{q \in \mathcal{I}_l} \alpha_l^{(t)}(q)\, \delta s_l(q)\Big) + \mathcal{O}(\eta^2), \tag{38}$$

$$\delta \hat{\mathbf{p}}_{l+1}^{\mathrm{att},(t)} = \sum_{k \in \mathcal{I}_l} \delta \alpha_l^{(t)}(k)\, \mathbf{p}_k + \mathcal{O}(\eta^2), \tag{39}$$

$$\delta p_{\mathrm{vocab}}^{(t)}(r) = p_{\mathrm{vocab}}^{(t)}(r) \frac{\big\langle \delta \hat{\mathbf{p}}_{l+1}^{\mathrm{att},(t)},\ \mathbf{p}_r - \sum_q p_{\mathrm{vocab}}^{(t)}(q)\mathbf{p}_q \big\rangle}{\beta} + \mathcal{O}(\eta^2), \tag{40}$$

$$\delta \eta_l^{(t)}(k) = \tfrac{1}{\beta}\Big(-\big\langle \delta \hat{\mathbf{p}}_{l+1}^{\mathrm{att},(t)},\ \mathbf{p}_{i_l} - \sum_r p_{\mathrm{vocab}}^{(t)}(r)\mathbf{p}_r \big\rangle - \big\langle \mathbf{p}_k - \hat{\mathbf{p}}_{l+1}^{\mathrm{att},(t)},\ \sum_r \delta p_{\mathrm{vocab}}^{(t)}(r)\, \mathbf{p}_r \big\rangle\Big) + \mathcal{O}(\eta^2). \tag{41}$$

*In practice, with $n$ trajectories prompted by $\mathbf{x}^s \sim \mathrm{Unif}([K]^d), \forall s \in [n]$, Monte Carlo estimators can replace the expectations in equation 33–equation 36, e.g.,*

$$\widehat{\delta s}_l^{(t)}(k) = \eta \frac{1}{n} \sum_{s=1}^n R^{k^\star}(\mathrm{TF}(\mathbf{x}^s; \mathbf{W}^{(t)}))\, \alpha_l^{s,(t)}(k)\, \eta_l^{s,(t)}(k).$$

**Proof.** *(1) By Lemma 5 and Lemma 3, at $\mathbf{W}^{(t-1)}$,*

$$\nabla_{\mathbf{W}} \mathcal{J}_{\mathrm{REINFORCE}}^{k^\star} = \mathbb{E}_\tau\Big[R^{k^\star} \sum_{u=1}^{k^\star+1} \sum_{k \in \mathcal{I}_u} \alpha_u(k)\, \eta_u(k)\, \mathbf{p}_k \mathbf{p}_{c_u}^\top\Big].$$

*Using $\nabla_{\mathbf{W}} s_l(j) = \mathbf{p}_j \mathbf{p}_{c_l}^\top$ and the Frobenius inner product, a Taylor expansion up to second order along the explicit-Euler path $\mathbf{W}^{(t+1)} = \mathbf{W}^{(t)} + \eta\, \mathbf{G}_t$ (with fixed $\mathbf{G}_t := \nabla_{\mathbf{W}} \mathcal{J}_{\mathrm{REINFORCE}}^{k^\star}(\mathbf{W}^{(t)})$) gives*

$$s_l\big(j; \mathbf{W}^{(t+1)}\big) = s_l\big(j; \mathbf{W}^{(t)}\big) + \eta\big\langle \mathbf{p}_j \mathbf{p}_{c_l}^\top,\ \nabla_{\mathbf{W}} \mathcal{J}_{\mathrm{REINFORCE}}^{k^\star}\big\rangle + \mathcal{O}(\eta^2),$$

*and, more explicitly,*

$$s_l\big(j; \mathbf{W}^{(t)} + \eta\,\mathbf{G}_t\big) = s_l\big(j; \mathbf{W}^{(t)}\big) + \eta\,\big\langle \nabla_{\mathbf{W}} s_l(j),\ \mathbf{G}_t \big\rangle + \frac{\eta^2}{2}\,\big\langle \mathbf{G}_t,\ \nabla_{\mathbf{W}}^2 s_l(j)[\mathbf{G}_t] \big\rangle.$$

*Since $s_l(j; \mathbf{W}) = \langle \mathbf{p}_j \mathbf{p}_{c_l}^\top,\ \mathbf{W} \rangle + m_l(j)$ is affine in $\mathbf{W}$, its Hessian vanishes: $\nabla_{\mathbf{W}}^2 s_l(j) \equiv 0$. Therefore the second-order term above is identically zero under the explicit-Euler step (no dependence of the direction on $\eta$), and only the $\mathcal{O}(\eta^2)$ bookkeeping remains for uniformity with subsequent nonlinear propagations. and OBI guarantees $\langle \mathbf{p}_j \mathbf{p}_{c_l}^\top,\ \mathbf{p}_k \mathbf{p}_{c_u}^\top \rangle = 0$ unless $(k, u) = (j, l)$, which yields equation 33.*

*(2) Let $g_j := \eta\,\mathbb{E}_\tau[R^{k^\star}\,\alpha_l(j)\,\eta_l(j)]$ denote the first-order increment of $s_l(j)$. By Danskin's theorem / the subgradient inequality,*

$$\max_{j \neq i_l} \big(s_l(j) + g_j\big)\ \leq\ \max_{j \neq i_l} s_l(j)\ +\ \max_{j \neq i_l} g_j,$$

*and substituting the definitions gives the lower bound equation 34. If the active maximizer is unique and remains unchanged in a neighborhood, then the directional derivative of $\max_{j \neq i_l}$ is given by that $j_l^{\max}$, which yields equation 35.*

*(3) Let $\phi_l = s_l(i_l) - \log \sum_{j \neq i_l} e^{s_l(j)}$, so $\Delta_l \geq \phi_l$. Its directional derivative is*

$$\delta\phi_l = \delta s_l(i_l) - \sum_{j \neq i_l} \tilde{\alpha}_l(j)\,\delta s_l(j).$$

*Substituting equation 33 (with $t \to t+1$) and collecting $\mathcal{O}(\eta^2)$ terms gives equation 36.*

*Finally, we detail the second-order sources for each intermediate quantity and then summarize them as $\mathcal{O}(\eta^2)$ terms:*

*(i) Softmax attention $\alpha_l$ to second order. Let $\boldsymbol{s}_l \in \mathbb{R}^{d_l}$ collect $s_l(\cdot)$ and write $\boldsymbol{\alpha}_l = \mathrm{softmax}(\boldsymbol{s}_l)$. For a perturbation $\delta\boldsymbol{s}_l = \mathcal{O}(\eta)$, the second-order expansion of component $j$ is*

$$\delta\alpha_l(j) = \sum_k J_{jk}^\alpha\,\delta s_l(k) + \frac{1}{2}\sum_{k,m} H_{j,km}^\alpha\,\delta s_l(k)\,\delta s_l(m) + \mathcal{O}(\eta^3),$$

*with Jacobian $J_{jk}^\alpha = \alpha_l(j)(\delta_{jk} - \alpha_l(k))$ and Hessian*

$$H_{j,km}^\alpha = \frac{\partial^2 \alpha_l(j)}{\partial s_l(k)\,\partial s_l(m)} = \alpha_l(j)\Big((\delta_{jm} - \alpha_l(m))(\delta_{jk} - \alpha_l(k))\ -\ \alpha_l(k)(\delta_{km} - \alpha_l(m))\Big).$$

*Because $\delta s_l = \mathcal{O}(\eta)$ from equation 33, the quadratic term contributes $\mathcal{O}(\eta^2)$.*

*(ii) Attention-weighted token vector $\hat{\mathbf{p}}_{l+1}^{\mathrm{att}}$. Since $\hat{\mathbf{p}}_{l+1}^{\mathrm{att}} = \sum_j \alpha_l(j)\,\mathbf{p}_j$ is linear in $\boldsymbol{\alpha}_l$,*

$$\delta\hat{\mathbf{p}}_{l+1}^{\mathrm{att}} = \sum_j \delta\alpha_l(j)\,\mathbf{p}_j = \sum_{j,k} J_{jk}^\alpha\,\delta s_l(k)\,\mathbf{p}_j + \frac{1}{2}\sum_{j,k,m} H_{j,km}^\alpha\,\delta s_l(k)\,\delta s_l(m)\,\mathbf{p}_j + \mathcal{O}(\eta^3),$$

*so $\delta\hat{\mathbf{p}}_{l+1}^{\mathrm{att}} = \mathcal{O}(\eta) + \mathcal{O}(\eta^2)$.*

*(iii) Vocabulary softmax $p_{\mathrm{vocab}}$. Let $\ell_r = \langle \hat{\mathbf{p}}_{l+1}^{\mathrm{att}},\ \mathbf{p}_r \rangle / \beta$ and $\boldsymbol{p} = \mathrm{softmax}(\boldsymbol{\ell})$. Then*

$$\delta\ell_r = \frac{\langle \delta\hat{\mathbf{p}}_{l+1}^{\mathrm{att}},\ \mathbf{p}_r \rangle}{\beta}, \qquad \delta p(r) = \sum_m J_{rm}^p\,\delta\ell_m + \frac{1}{2}\sum_{m,n} H_{r,mn}^p\,\delta\ell_m\,\delta\ell_n + \mathcal{O}(\eta^3),$$

*where $J_{rm}^p = p(r)(\delta_{rm} - p(m))$ and*

$$H_{r,mn}^p = \frac{\partial^2 p(r)}{\partial\ell_m\,\partial\ell_n} = p(r)\Big((\delta_{rn} - p(n))(\delta_{rm} - p(m))\ -\ p(m)(\delta_{mn} - p(n))\Big).$$

*Since $\delta\hat{\mathbf{p}}_{l+1}^{\mathrm{att}} = \mathcal{O}(\eta) + \mathcal{O}(\eta^2)$, we have $\delta\ell = \mathcal{O}(\eta) + \mathcal{O}(\eta^2)$ and thus $\delta p = \mathcal{O}(\eta) + \mathcal{O}(\eta^2)$.*

*(iv) The scalar $\eta_l(k)$. Write*

$$\eta_l(k) = \frac{1}{\beta}\Big\langle \underbrace{\mathbf{p}_k - \hat{\mathbf{p}}_{l+1}^{\mathrm{att}}}_{:=\,\mathbf{u}},\ \underbrace{\mathbf{p}_{i_l} - \sum_r p(r)\,\mathbf{p}_r}_{:=\,\mathbf{v}} \Big\rangle.$$

*Perturbing* $(\mathbf{u}, \mathbf{v}) \mapsto (\mathbf{u} - \delta\hat{\mathbf{p}}_{l+1}^{\mathrm{att}}, \ \mathbf{v} - \sum_r \delta p(r)\, \mathbf{p}_r)$ *and expanding to second order yields*

$$\delta\eta_l(k) = \tfrac{1}{\beta}\Big( -\langle\delta\hat{\mathbf{p}}_{l+1}^{\mathrm{att}}, \mathbf{v}\rangle - \langle\mathbf{u}, \sum_r \delta p(r)\, \mathbf{p}_r\rangle \Big) + \tfrac{1}{\beta}\Big( -\tfrac{1}{2}\langle\delta^2\hat{\mathbf{p}}_{l+1}^{\mathrm{att}}, \mathbf{v}\rangle - \langle\delta\hat{\mathbf{p}}_{l+1}^{\mathrm{att}}, \sum_r \delta p(r)\, \mathbf{p}_r\rangle - \tfrac{1}{2}\langle\mathbf{u}, \sum_r \delta^2 p(r)\, \mathbf{p}_r\rangle \Big) + \mathcal{O}(\eta^3),$$

*where* $\delta^2\hat{\mathbf{p}}_{l+1}^{\mathrm{att}}$ *and* $\delta^2 p(r)$ *collect the quadratic terms shown in (ii) and (iii). Since* $\delta\hat{\mathbf{p}}_{l+1}^{\mathrm{att}} = \mathcal{O}(\eta)$ *and* $\delta p = \mathcal{O}(\eta)$ *at leading order, all bracketed second-line contributions are* $\mathcal{O}(\eta^2)$.

*Collecting these, we obtain the one-step relations stated in equation 37–equation 41, with every omitted higher-order contribution explicitly accounted for by the quadratic (Hessian) terms above and summarized as* $\mathcal{O}(\eta^2)$ *due to the small step size.* $\qquad\square$

**Lemma 7** (Stepwise expected gradients at iteration $t$ with spurious success). *Fix iteration $t$ and a step $l \in \{1, \ldots, k^\star + 1\}$. For any index block $j \in \mathcal{I}_l$, write the block-projected expected REIN-FORCE gradient at $\mathbf{W}^{(t)}$ as*

$$G_{l,j}^{(t)} := \Big\langle \nabla_{\mathbf{W}} \mathcal{J}_{\mathrm{REINFORCE}}^{k^\star}(\mathbf{W}^{(t)}), \ \mathbf{p}_j \mathbf{p}_{c_l}^\top \Big\rangle = \mathbb{E}^{(t)}\big[ R^{k^\star}\, \alpha_l^{(t)}(j)\, \eta_l^{(t)}(j) \big],$$

*where* $\mathbb{E}^{(t)}[\cdot]$ *denotes the expectation over* $\mathbf{x} \sim \mathcal{P}_x$ *and* $\tau \sim p_{\mathbf{W}^{(t)}}(\cdot \mid \mathbf{x})$, *and* $\alpha_l^{(t)}(\cdot), \eta_l^{(t)}(\cdot)$ *are computed at* $\mathbf{W}^{(t)}$ *(Lemma 3).*

*Let* $C_{<l}$ *be the event that the index prefix* $(i_1, \ldots, i_{l-1})$ *matches the target* $(i_1^\star, \ldots, i_{l-1}^\star)$, *and denote*

$$
\begin{aligned}
p_{<l}^{(t)} &:= \mathbb{P}^{(t)}(C_{<l} = 1), \\
S_{l,j,\mathrm{corr}}^{(t)} &:= \mathbb{E}^{(t)}\big[ \alpha_l^{(t)}(j)\eta_l^{(t)}(j)\, \mathbf{1}\{i_l = i_l^\star\} \,\big|\, C_{<l} = 1 \big], \\
S_{l,j,\mathrm{wrong}}^{(t)} &:= \mathbb{E}^{(t)}\big[ \alpha_l^{(t)}(j)\eta_l^{(t)}(j)\, \mathbf{1}\{i_l \neq i_l^\star\} \,\big|\, C_{<l} = 1 \big], \\
\bar{S}_{l,j}^{(t)} &:= \mathbb{E}^{(t)}\big[ \alpha_l^{(t)}(j)\eta_l^{(t)}(j) \,\big|\, C_{<l} = 0 \big].
\end{aligned}
\tag{42}
$$

*Then*

$$G_{l,j}^{(t)} = p_{<l}^{(t)}\Big( q_{l|\mathrm{corr}}^{(t)} S_{l,j,\mathrm{corr}}^{(t)} + q_{l|\mathrm{wrong}}^{(t)} S_{l,j,\mathrm{wrong}}^{(t)} \Big) + (1 - p_{<l}^{(t)})\, q_{<l|\mathrm{wrong}}^{(t)} \bar{S}_{l,j}^{(t)}, \tag{43}$$

*where for the given reward* $R^{k^\star}$ *we define the conditional success probabilities:*

$$
\begin{aligned}
q_{l|\mathrm{corr}}^{(t)} &:= \mathbb{P}^{(t)}\big( R^{k^\star} = 1 \,\big|\, C_{<l} = 1, \ i_l = i_l^\star \big), \\
q_{l|\mathrm{wrong}}^{(t)} &:= \mathbb{P}^{(t)}\big( R^{k^\star} = 1 \,\big|\, C_{<l} = 1, \ i_l \neq i_l^\star \big), \\
q_{<l|\mathrm{wrong}}^{(t)} &:= \mathbb{P}^{(t)}\big( R^{k^\star} = 1 \,\big|\, C_{<l} = 0 \big).
\end{aligned}
\tag{44}
$$

*(a) Final-answer reward* $R^{k^\star} = R^{f_{S\star}}$. *Let* $p_{\mathrm{tail}}^{(t)}(l+1)$ *denote the probability of completing the remaining true path from step $l+1$ onward under $\mathbf{W}^{(t)}$, and let $\rho_{\mathrm{spur},\geq l}^{(t)}$ be the spurious-success rate conditioned on having a wrong index at step $r \geq l$ (first deviation at $\geq l$), and $\rho_{\mathrm{spur},<l}^{(t)}$ for having deviated before step $l$. Then*

$$q_{l|\mathrm{corr}}^{(t)} = p_{\mathrm{tail}}^{(t)}(l+1) + \big(1 - p_{\mathrm{tail}}^{(t)}(l+1)\big) \rho_{\mathrm{spur},\geq l+1}^{(t)}, \tag{45}$$

$$q_{l|\mathrm{wrong}}^{(t)} = \rho_{\mathrm{spur},\geq l}^{(t)}, \qquad q_{<l|\mathrm{wrong}}^{(t)} = \rho_{\mathrm{spur},<l}^{(t)}. \tag{46}$$

*For the parity case with* $\mathbf{x} \sim \mathrm{Unif}\{0,1\}^d$ *and XOR kernels,* $\rho_{\mathrm{spur},\geq l}^{(t)} = \rho_{\mathrm{spur},<l}^{(t)} = 1/2$ *for all $l$.*

*(b) Subtask-family reward* $R^{k^\star} = R^{\mathcal{F}_{S\star}}$. *Let* $T^{(t)} \in \{1, \ldots, k^\star + 1\}$ *be the (random) termination step (the first step where EOS is sampled). For each depth $r \in [k^\star]$, define the event*

$$U_r := \big\{ T^{(t)} = r+1 \text{ and the pre-EOS token equals } \boldsymbol{\mu}^{f_{S^r}}(\mathbf{x}) \big\}.$$

*Then* $R^{\mathcal{F}_{S\star}} = \mathbf{1}\{\cup_{r=1}^{k^\star} U_r\}$, *and the stepwise expected gradient admits the depth-wise expansion*

$$G_{l,j}^{(t)} = \sum_{r=l}^{k^\star} \mathbb{E}^{(t)}\big[ \mathbf{1}\{U_r\}\, \alpha_l^{(t)}(j)\, \eta_l^{(t)}(j) \big]. \tag{47}$$

*Consequently, for every $l \leq k^\star$,*

$$G^{(t)}_{l,j}\big|_{R=R^{\mathcal{F}_{S_\star}}} = G^{(t)}_{l,j}\big|_{R=R^{f_{S_\star}}} + \sum_{r=l}^{k^\star-1} \mathbb{E}^{(t)}\big[\mathbf{1}\{U_r\}\,\alpha^{(t)}_l(j)\,\eta^{(t)}_l(j)\big] \geq G^{(t)}_{l,j}\big|_{R=R^{f_{S_\star}}},$$

*which shows the subtask-family reward strictly adds nonnegative depth-wise contributions on shared prefixes.*

**Explicit per-depth success probabilities for (b).** *For $s \in [k^\star]$, define the per-step true-path continuation probabilities and their products*

$$\pi^{(t)}_s := \mathbb{P}^{(t)}\big(i_s = i^\star_s \mid C_{<s} = 1\big), \qquad \Pi^{(t)}_{a:b} := \prod_{s=a}^{b} \pi^{(t)}_s \ (\Pi^{(t)}_{a:b} \equiv 1 \ if \ a > b).$$

*Define the path events*

$$A_r := \{C_{<r} = 1, \ i_r = i^\star_r\}, \qquad B_r := \{C_{<r} = 1, \ i_r \neq i^\star_r\}, \qquad D_r := \{C_{<r} = 0\}.$$

*For each $r \geq l$, define branch-specific one-step termination probabilities at depth $r+1$:*

$$\theta^{(t)}_{r+1,A} := \mathbb{P}^{(t)}(i_{r+1} = \text{EOS} \mid A_r), \quad \theta^{(t)}_{r+1,B} := \mathbb{P}^{(t)}(i_{r+1} = \text{EOS} \mid B_r), \quad \theta^{(t)}_{r+1,D} := \mathbb{P}^{(t)}(i_{r+1} = \text{EOS} \mid D_r),$$

*and subtask-match probabilities at depth $r$ under the three branches (task dependent through $\{\Phi_l\}$):*

$$\rho^{(t)}_{r|A} := \mathbb{P}^{(t)}(\text{subtask match at } r \mid A_r), \ \ \rho^{(t)}_{r|B} := \mathbb{P}^{(t)}(\text{subtask match at } r \mid B_r), \ \ \rho^{(t)}_{r|D} := \mathbb{P}^{(t)}(\text{subtask match at } r \mid D_r).$$

*Then the event-level probabilities that enter equation 47 decompose explicitly as*

$$\mathbb{P}^{(t)}\big(U_r \mid C_{<l} = 1, \ i_l = i^\star_l\big)$$

$$= \underbrace{\rho^{(t)}_{r|A}\,\Pi^{(t)}_{l+1:r}\,\theta^{(t)}_{r+1,A}}_{\textit{true-path to } r} + \underbrace{\rho^{(t)}_{r|B}\,\Pi^{(t)}_{l+1:r-1}(1 - \pi^{(t)}_r)\,\theta^{(t)}_{r+1,B}}_{\textit{first deviation at } r} + \underbrace{\rho^{(t)}_{r|D}\,(1 - \Pi^{(t)}_{l+1:r-1})\,\theta^{(t)}_{r+1,D}}_{\textit{deviation before } r},$$

$$\mathbb{P}^{(t)}\big(U_r \mid C_{<l} = 1, \ i_l \neq i^\star_l\big)$$

$$= \mathbf{1}\{r = l\}\,\rho^{(t)}_{r|B}\,\theta^{(t)}_{l+1,B} + \mathbf{1}\{r > l\}\,\rho^{(t)}_{r|D}\,\theta^{(t)}_{r+1,D},$$

$$\mathbb{P}^{(t)}\big(U_r \mid C_{<l} = 0\big)$$

$$= \rho^{(t)}_{r|D}\,\theta^{(t)}_{r+1,D}.$$

$$(48)$$

*Consequently, identifying with the conditional success probabilities in Eq. equation 43 for $R = R^{\mathcal{F}_{S_\star}}$, the per-step success factors equal*

$$q^{(t)}_{l|\text{corr}} = \sum_{r=l}^{k^\star} \mathbb{P}^{(t)}\big(U_r \mid C_{<l} = 1, \ i_l = i^\star_l\big),$$

$$q^{(t)}_{l|\text{wrong}} = \sum_{r=l}^{k^\star} \mathbb{P}^{(t)}\big(U_r \mid C_{<l} = 1, \ i_l \neq i^\star_l\big) = \rho^{(t)}_{l|B}\,\theta^{(t)}_{l+1,B} + \sum_{r=l+1}^{k^\star} \rho^{(t)}_{r|D}\,\theta^{(t)}_{r+1,D}, \qquad (49)$$

$$q^{(t)}_{<l|\text{wrong}} = \sum_{r=l}^{k^\star} \mathbb{P}^{(t)}\big(U_r \mid C_{<l} = 0\big) = \sum_{r=l}^{k^\star} \rho^{(t)}_{r|D}\,\theta^{(t)}_{r+1,D}.$$

*In the parity case, $\rho^{(t)}_{r|A} \equiv 1$ and $\rho^{(t)}_{r|B} = \rho^{(t)}_{r|D} \equiv 1/2$, yielding the explicit decomposition Eq. equation 50 and, under uniform 2S-ART, the closed form Eq. equation 53.*

**Parity specialization** *Under the XOR kernel for subtasks with $\mathbf{x} \sim \text{Unif}\{0,1\}^d$, the per-branch subtask-match probabilities satisfy $\rho^{(t)}_{r|A} \equiv 1$ and $\rho^{(t)}_{r|B} = \rho^{(t)}_{r|D} \equiv \frac{1}{2}$. Writing the one-step termination probability at $r+1$ under a condition $E \in \{A_r, B_r, D_r\}$ as*

$$\theta^{(t)}_{r+1}(E) := \mathbb{P}^{(t)}\big(i_{r+1} = \text{EOS} \mid E\big),$$

*the probability of the event $U_r$ decomposes as*

$$\mathbb{P}^{(t)}(U_r) = \underbrace{\mathbb{P}^{(t)}(A_r)\,\theta_{r+1}^{(t)}(A_r)}_{\text{true-path contribution}} + \underbrace{\tfrac{1}{2}\,\mathbb{P}^{(t)}(B_r)\,\theta_{r+1}^{(t)}(B_r) + \tfrac{1}{2}\,\mathbb{P}^{(t)}(D_r)\,\theta_{r+1}^{(t)}(D_r)}_{\text{spurious (reward-hacking) contribution}}. \qquad (50)$$

*Moreover, writing $p_{\text{path}}^{(t)}(r) := \mathbb{P}^{(t)}(A_r)$ and $p_{\text{dev},\geq r}^{(t)} := \mathbb{P}^{(t)}(B_r)$, $p_{\text{dev},<r}^{(t)} := \mathbb{P}^{(t)}(D_r)$, we have bounds*

$$\mathbb{P}^{(t)}(U_r) \geq p_{\text{path}}^{(t)}(r)\,\inf\theta_{r+1}^{(t)}(A_r), \quad \mathbb{P}^{(t)}(U_r) \leq p_{\text{path}}^{(t)}(r)\,\sup\theta_{r+1}^{(t)}(A_r) + \tfrac{1}{2}\big(p_{\text{dev},\geq r}^{(t)}\,\sup\theta_{r+1}^{(t)}(B_r) + p_{\text{dev},<r}^{(t)}\,\sup\theta_{r+1}^{(t)}(D_r)\big). \qquad (51)$$

*If, in addition, the EOS policy at step $r+1$ is conditionally independent of the path branch (same marginal $\bar{\theta}_{r+1}^{(t)}$),*

$$\mathbb{P}^{(t)}(U_r) = \bar{\theta}_{r+1}^{(t)}\Big(p_{\text{path}}^{(t)}(r) + \tfrac{1}{2}\big(p_{\text{dev},\geq r}^{(t)} + p_{\text{dev},<r}^{(t)}\big)\Big). \qquad (52)$$

*Under the base uniform 2S-ART for parity, namely for $s \in [k^\star]$ and $r \in [k^\star]$,*

$$\pi_s^{(t)} = \frac{1}{d-s+1}, \qquad \theta_{r+1,A}^{(t)} = \theta_{r+1,B}^{(t)} = \theta_{r+1,D}^{(t)} = \frac{1}{d-r+1},$$

*define*

$$P_{r-1} := \prod_{s=1}^{r-1} \frac{1}{d-s+1} = \frac{1}{d(d-1)\cdots(d-r+2)} \quad (\text{and } P_0 := 1).$$

*Then for every $r \in [k^\star]$,*

$$\mathbb{P}^{(t)}(U_r) = \frac{P_{r-1}}{(d-r+1)^2} + \frac{1}{2}\frac{P_{r-1}(d-r)}{(d-r+1)^2} + \frac{1}{2}\frac{1-P_{r-1}}{d-r+1} = \frac{P_{r-1}\,(d-r+2)}{2\,(d-r+1)^2} + \frac{1-P_{r-1}}{2\,(d-r+1)}. \qquad (53)$$

**Proof.** *Starting from Lemma 5 and Lemma 3, for any $l, j$ we have*

$$G_{l,j}^{(t)} = \mathbb{E}^{(t)}\big[R^{k^\star}\,\alpha_l^{(t)}(j)\,\eta_l^{(t)}(j)\big].$$

*Condition on $C_{<l}$ and $i_l$ (tower property):*

$$G_{l,j}^{(t)} = \mathbb{E}^{(t)}\Big[\mathbb{E}^{(t)}\big[R^{k^\star}\,\alpha_l^{(t)}(j)\,\eta_l^{(t)}(j)\,\big|\,C_{<l},\,i_l\big]\Big]$$

$$= p_{<l}^{(t)}\,\mathbb{E}^{(t)}\Big[\alpha_l^{(t)}(j)\,\eta_l^{(t)}(j)\,\mathbb{E}^{(t)}\big[R^{k^\star}\,\big|\,C_{<l}=1,\,i_l\big]\,\big|\,C_{<l}=1\Big]$$

$$+ (1-p_{<l}^{(t)})\,\mathbb{E}^{(t)}\Big[\alpha_l^{(t)}(j)\,\eta_l^{(t)}(j)\,\mathbb{E}^{(t)}\big[R^{k^\star}\,\big|\,C_{<l}=0\big]\,\big|\,C_{<l}=0\Big].$$

*This gives equation 43 once we identify $q_{l|\cdot}^{(t)}$ by whether $i_l = i_l^\star$.*

*For (a), $R^{k^\star} = R^{f_{S^\star}}$: on the true path through step $l$ with $i_l = i_l^\star$, success thereafter requires completing the remainder true path; otherwise, success can occur spuriously, which yields the formulas for $q_{l|\text{corr}}^{(t)}$, $q_{l|\text{wrong}}^{(t)}$ and $q_{<l|\text{wrong}}^{(t)}$.*

*For (b), $R^{k^\star} = R^{\mathcal{F}_{S^\star}}$: success occurs at the unique termination depth $r$ where the pre-EOS token equals the subtask token; hence $R^{\mathcal{F}_{S^\star}} = \sum_{r=1}^{k^\star}\mathbf{1}\{U_r\}$. Expanding $\mathbf{1}\{\cup_r U_r\}$ as $\sum_r \mathbf{1}\{U_r\}$, exchanging sum and expectation, and applying the same conditioning as above yields*

$$G_{l,j}^{(t)} = \sum_{r=l}^{k^\star} \mathbb{E}^{(t)}\big[\mathbf{1}\{U_r\}\,\alpha_l^{(t)}(j)\,\eta_l^{(t)}(j)\big],$$

*which implies $G_{l,j}^{(t)}\big|_{R=R^{\mathcal{F}_{S^\star}}} \geq G_{l,j}^{(t)}\big|_{R=R^{f_{S^\star}}}$ via the nonnegative extra depths $r \in \{l,\ldots,k^\star-1\}$. In the parity case, producing the correct subtask token at depth $r$ deterministically fixes $z_r$; any immediate-EOS strategy yields the subtask reward at $r$, while spurious subtask matches contribute via the corresponding conditional probabilities.* $\qquad\square$

**Lemma 8** (Finite-sample variance of REINFORCE objective and gradient (general post-training)). *Let $(\mathbf{x}^{(s)}, \tau^{(s)})_{s=1}^n$ be i.i.d., where $\mathbf{x}^{(s)} \sim \mathcal{P}_x$ and $\tau^{(s)} \sim p_{\mathbf{W}^{k^\star}}(\cdot \mid \mathbf{x}^{(s)})$. Define the Monte Carlo estimators*

$$\widehat{\mathcal{J}}_n^{k^\star}(\mathbf{W}^{k^\star}) := \frac{1}{n} \sum_{s=1}^n R^{k^\star}(\boldsymbol{\mu}^{z^{k^\star}}(\tau_{-2}^{(s)})),$$

$$\widehat{\mathbf{g}}_n^{k^\star}(\mathbf{W}^{k^\star}) := \frac{1}{n} \sum_{s=1}^n R^{k^\star}(\hat{\mathbf{p}}^{z^{k^\star}}(\tau_{-2}^{(s)})) \sum_{l=1}^{k^\star+1} \nabla_{\mathbf{W}^{k^\star}} \log \pi_{\mathbf{W}^{k^\star}}(i_l^{(s)} \mid \mathbf{x}^{(s)}, \hat{\mathbf{p}}^{z^{1:l}}).$$

*Assume $R^{k^\star} \in \{0, 1\}$, the regularity conditions of Lemma 5, and Orthogonal Block Isolation (Lemma 3). Let*

$$p_{\text{succ}}(\mathbf{W}^{k^\star}) := \mathbb{E}_{\mathbf{x} \sim \mathcal{P}_x} \Big[ \mathbb{P}_{\tau \sim p_{\mathbf{W}^{k^\star}}(\cdot \mid \mathbf{x})} \big\{ R^{k^\star}(\hat{\mathbf{p}}^{z^{k^\star}}(\tau_{-2})) = 1 \big\} \Big] = \mathbb{E}_{\mathbf{x}, \tau} \big[ R^{k^\star}(\hat{\mathbf{p}}^{z^{k^\star}}(\tau_{-2})) \big].$$

*Then:*

*1) (Objective) Unbiasedness and variance:*

$$\mathbb{E}\big[\widehat{\mathcal{J}}_n^{k^\star}\big] = \mathcal{J}_{\text{REINFORCE}}^{k^\star}, \qquad \text{Var}\big[\widehat{\mathcal{J}}_n^{k^\star}\big] = \frac{1}{n} p_{\text{succ}}(1 - p_{\text{succ}}) \leq \frac{1}{4n}. \tag{54}$$

*In particular, under near-uniform initialization and using equation 31, we have*

$$p_{\text{succ}} \leq \rho_{\text{spur}} + (1 - \rho_{\text{spur}}) \prod_{l=1}^{k^\star+1} \frac{1}{d_l} \qquad (\text{if } d_l = \Theta(d), \ p_{\text{succ}} \leq \rho_{\text{spur}} + (1 - \rho_{\text{spur}}) \Theta(d^{-(k^\star+1)})), \tag{55}$$

*so that*

$$\text{Var}\big[\widehat{\mathcal{J}}_n^{k^\star}\big] \leq \frac{1}{n} \Big( \rho_{\text{spur}} + (1 - \rho_{\text{spur}}) d^{-(k^\star+1)} \Big) \Big( 1 - \rho_{\text{spur}} - (1 - \rho_{\text{spur}}) d^{-(k^\star+1)} \Big) \leq \frac{1}{4n}. \tag{56}$$

*2) (Gradient) Unbiasedness and covariance. Let $\boldsymbol{\mu} := \nabla_{\mathbf{W}^{k^\star}} \mathcal{J}_{\text{REINFORCE}}^{k^\star}$ be the population gradient. Then*

$$\mathbb{E}\big[\widehat{\mathbf{g}}_n^{k^\star}\big] = \boldsymbol{\mu}, \qquad \text{Cov}\big[\widehat{\mathbf{g}}_n^{k^\star}\big] = \frac{1}{n} \Big( \underbrace{\mathbb{E}_{\mathbf{x}, \tau}\Big[ R^{k^\star} \sum_{l,t=1}^{k^\star+1} \nabla \log \pi_l \otimes \nabla \log \pi_t \Big] - \boldsymbol{\mu} \otimes \boldsymbol{\mu}}_{=: \boldsymbol{\Sigma}_{\text{pop}}} \Big), \tag{57}$$

*where $\pi_l$ abbreviates $\pi_{\mathbf{W}^{k^\star}}(i_l \mid \mathbf{x}, \hat{\mathbf{p}}^{z^{1:l}})$ and $\otimes$ is the outer product in parameter space.*

*Furthermore, using Lemma 3, write a block-orthogonal expansion*

$$\nabla \log \pi_l = \sum_{k \in \mathcal{I}_l} \alpha_l^{k^\star}(k) \eta_l^{k^\star}(k) \mathbf{B}_{l,k}, \qquad \mathbf{B}_{l,k} := \mathbf{p}_k \mathbf{p}_{c_l}^\top.$$

*Denote the block coefficient $C_{l,k} := R^{k^\star} \alpha_l^{k^\star}(k) \eta_l^{k^\star}(k)$. Then the scalar variance in each orthogonal block is*

$$\text{Var}\Big[ \big\langle \widehat{\mathbf{g}}_n^{k^\star}, \frac{\mathbf{B}_{l,k}}{\|\mathbf{B}_{l,k}\|_F} \big\rangle \Big] = \frac{1}{n} \Big( \mathbb{E}[C_{l,k}^2] - \mathbb{E}[C_{l,k}]^2 \Big), \tag{58}$$

*and the Frobenius-mean-square error (variance) satisfies*

$$\mathbb{E}\big\|\widehat{\mathbf{g}}_n^{k^\star} - \boldsymbol{\mu}\big\|_F^2 = \frac{1}{n} \Big( \underbrace{\mathbb{E}\big[ R^{k^\star} \sum_{l=1}^{k^\star+1} \sum_{k \in \mathcal{I}_l} (\alpha_l^{k^\star}(k))^2 (\eta_l^{k^\star}(k))^2 \|\mathbf{B}_{l,k}\|_F^2 \big]}_{=: \Xi} - \|\boldsymbol{\mu}\|_F^2 \Big). \tag{59}$$

*Assume bounded embeddings and temperature $\beta > 0$ so that $|\eta_l^{k^\star}(k)| \leq 4/\beta$, and define the concentration factor*

$$S_l := \mathbb{E}_{\mathbf{x}, \tau} \Big[ \sum_{k \in \mathcal{I}_l^{\text{legal}}} (\alpha_l^{k^\star}(k))^2 \Big] \in \Big[ \frac{1}{d_l}, 1 \Big].$$

*Let $C_{\text{pos}} := \max_{l,k} \|\mathbf{p}_k\|_2^2 \|\mathbf{p}_{c_l}\|_2^2 = 1$. Then*

$$\Xi \leq \mathbb{E}[R^{k^\star}] \sum_{l=1}^{k^\star+1} \left( \frac{16}{\beta^2} C_{\text{pos}} \right) \mathbb{E}\Big[ \sum_{k \in \mathcal{I}_l^{\text{legal}}} (\alpha_l^{k^\star}(k))^2 \Big] \tag{60}$$

$$= \frac{16}{\beta^2} p_{\text{succ}} \sum_{l=1}^{k^\star+1} S_l. \tag{61}$$

*Consequently,*

$$\mathbb{E}\big\|\widehat{\mathbf{g}}_n^{k^\star} - \boldsymbol{\mu}\big\|_F^2 \leq \frac{1}{n} \frac{16}{\beta^2} p_{\text{succ}} \sum_{l=1}^{k^\star+1} S_l. \tag{62}$$

*Two useful specializations:*

- *(Near-uniform init) $S_l = \Theta(d^{-1})$, it holds that $\mathbb{E}\big\|\widehat{\mathbf{g}}_n^{k^\star} - \boldsymbol{\mu}\big\|_F^2 \leq \frac{1}{n} \frac{16k^\star+1}{\beta^2 d} p_{\text{succ}}$.*

- *(Worst-case spiky attention) $S_l \leq 1$, yielding $\mathbb{E}\|\widehat{\mathbf{g}}_n^{k^\star} - \boldsymbol{\mu}\|_F^2 \leq \frac{1}{n} \frac{16}{\beta^2} p_{\text{succ}} (k^\star + 1)$, which holds for the entire post-training.*

*Accordingly, the signal-to-noise ratio satisfies the general lower bound*

$$\text{SNR} := \frac{\|\boldsymbol{\mu}\|_F^2}{\mathbb{E}\|\widehat{\mathbf{g}}_n^{k^\star} - \boldsymbol{\mu}\|_F^2} \geq \frac{n \|\boldsymbol{\mu}\|_F^2}{(16/\beta^2) p_{\text{succ}} \sum_l S_l},$$

*and reduces to the near-uniform initialization scaling when $S_l \approx 1/d_l$.*

**Proof.** *We proceed in parts.*

*(A) Objective: unbiasedness and variance. Define the single-sample random variable*

$$Y := R^{k^\star} \big( \boldsymbol{\mu}^{z_{k^\star}} (\tau_{-2}) \big) \in \{0, 1\}.$$

*By definition of $\mathcal{J}_{\text{REINFORCE}}^{k^\star}$ (Eq. (24)'s expectation target without the score factor),*

$$\mathbb{E}[Y] = p_{\text{succ}}(\mathbf{W}^{k^\star}) = \mathcal{J}_{\text{REINFORCE}}^{k^\star}.$$

*Hence $\widehat{\mathcal{J}}_n^{k^\star} = \frac{1}{n} \sum_{s=1}^n Y^{(s)}$ is unbiased. Since $(Y^{(s)})$ are i.i.d. Bernoulli($p_{\text{succ}}$),*

$$\text{Var}(\widehat{\mathcal{J}}_n^{k^\star}) = \frac{1}{n} \text{Var}(Y) = \frac{1}{n} p_{\text{succ}}(1 - p_{\text{succ}}) \leq \frac{1}{4n}.$$

*(B) Upper bound on $p_{\text{succ}}$ under 2S-ART near-uniformity. Suppose at each step $l$ there are $d_l \geq 1$ legal children and success requires selecting a unique correct legal child. Under 2S-ART, conditionally almost surely $\pi_l(\text{correct} \mid \cdot) \leq 1/d_l$. Therefore, for any fixed $(\mathbf{x}, \tau)$-measurable history,*

$$\mathbb{P}(Y = 1 \mid \text{history}) = \prod_{l=1}^{k^\star+1} \pi_l(\text{correct} \mid \cdot) \leq \prod_{l=1}^{k^\star+1} \frac{1}{d_l},$$

*and taking expectation over the history yields $p_{\text{succ}} \leq \prod_l d_l^{-1}$. If $d_l \equiv d$, then $p_{\text{succ}} \leq d^{-(k^\star+1)}$. Substituting into part (A) gives the displayed variance bound for $\widehat{\mathcal{J}}_n^{k^\star}$.*

*(C) Gradient: unbiasedness and covariance. Let the single-sample gradient random tensor be*

$$\mathbf{G} := R^{k^\star} \sum_{l=1}^{k^\star+1} \nabla_{\mathbf{W}^{k^\star}} \log \pi_{\mathbf{W}^{k^\star}} \big( i_l \mid \mathbf{x}, \hat{\mathbf{p}}^{z_{1:l}} \big).$$

*By Lemma 5, $\mathbb{E}[\mathbf{G}] = \nabla \mathcal{J}_{\text{REINFORCE}}^{k^\star} =: \boldsymbol{\mu}$, hence $\widehat{\mathbf{g}}_n^{k^\star} = \frac{1}{n} \sum_{s=1}^n \mathbf{G}^{(s)}$ is unbiased. With i.i.d. samples,*

$$\text{Cov}(\widehat{\mathbf{g}}_n^{k^\star}) = \frac{1}{n} \text{Cov}(\mathbf{G}) = \frac{1}{n} \Big( \mathbb{E}[\mathbf{G} \otimes \mathbf{G}] - \boldsymbol{\mu} \otimes \boldsymbol{\mu} \Big).$$

*Since $R^{k^\star} \in \{0, 1\}$, $R^{k^\star 2} = R^{k^\star}$, giving*

$$\mathbb{E}[\mathbf{G} \otimes \mathbf{G}] = \mathbb{E}\Big[R^{k^\star} \sum_{l,t} \nabla \log \pi_l \otimes \nabla \log \pi_t\Big] =: \mathbf{\Sigma}_{\text{pop}}.$$

*This proves the displayed covariance formula.*

*(D) Block-wise variance via OBI. By Lemma 3, for each step $l$,*

$$\nabla \log \pi_l = \sum_{k \in \mathcal{I}_l} \alpha_l^{k^\star}(k) \, \eta_l^{k^\star}(k) \, \mathbf{B}_{l,k}, \quad \mathbf{B}_{l,k} := \mathbf{p}_k \mathbf{p}_{c_l}^\top, \quad \langle \mathbf{B}_{l,k}, \mathbf{B}_{l',k'} \rangle_F = 0 \; ((l,k) \neq (l',k')).$$

*Hence*

$$\mathbf{G} = \sum_{l,k} C_{l,k} \, \mathbf{B}_{l,k}, \qquad C_{l,k} := R^{k^\star} \alpha_l^{k^\star}(k) \eta_l^{k^\star}(k).$$

*Projecting onto a unit-Frobenius block direction gives a scalar average of i.i.d. terms:*

$$\Big\langle \widehat{\mathbf{g}}_n^{k^\star}, \frac{\mathbf{B}_{l,k}}{\|\mathbf{B}_{l,k}\|_F} \Big\rangle = \frac{1}{n} \sum_{s=1}^n C_{l,k}^{(s)}.$$

*Thus $\mathrm{Var}(\langle \widehat{\mathbf{g}}_n^{k^\star}, \mathbf{B}_{l,k}/\|\mathbf{B}_{l,k}\|_F \rangle) = \frac{1}{n}(\mathbb{E}[C_{l,k}^2] - \mathbb{E}[C_{l,k}]^2)$.*

*(E) Frobenius MSE identity. Using orthogonality of blocks,*

$$\|\mathbf{G}\|_F^2 = R^{k^\star} \sum_{l,k} (\alpha_l^{k^\star}(k))^2 (\eta_l^{k^\star}(k))^2 \|\mathbf{B}_{l,k}\|_F^2.$$

*Since $\mathbb{E}\|\widehat{\mathbf{g}}_n^{k^\star} - \boldsymbol{\mu}\|_F^2 = \frac{1}{n}(\mathbb{E}\|\mathbf{G}\|_F^2 - \|\boldsymbol{\mu}\|_F^2)$ for i.i.d. averages, we obtain the displayed identity with*

$$\Xi := \mathbb{E}\Big[R^{k^\star} \sum_{l=1}^{k^\star+1} \sum_{k \in \mathcal{I}_l} (\alpha_l^{k^\star}(k))^2 (\eta_l^{k^\star}(k))^2 \|\mathbf{B}_{l,k}\|_F^2\Big].$$

*(F) Concrete upper bound for general post-training. Assume bounded embeddings and temperature $\beta > 0$ so that $|\eta_l^{k^\star}(k)| \leq 4/\beta$. Define $S_l := \mathbb{E}[\sum_{k \in \mathcal{I}_l^{\text{legal}}} (\alpha_l^{k^\star}(k))^2] \in [1/d_l, 1]$. With $C_{\text{pos}} := \max_{l,k} \|\mathbf{B}_{l,k}\|_F^2$,*

$$\big|\eta_l^{k^\star}(k)\big| = \frac{\big|\langle \mathbf{p}_k - \hat{\mathbf{p}}^{\text{att}}, \, \mathbf{p}_{i_l} - \sum_r p_{\text{vocab}}(r) \mathbf{p}_r \rangle\big|}{\beta} \leq \frac{\|\mathbf{p}_k - \hat{\mathbf{p}}^{\text{att}}\|_2 \, \|\mathbf{p}_{i_l} - \sum_r p_{\text{vocab}}(r) \mathbf{p}_r\|_2}{\beta} \leq \frac{4}{\beta},$$

*where we used triangle inequality and that each difference of two convex combinations of bounded, pairwise-orthogonal embeddings has norm at most 2. Denoting $C_{\text{pos}} := \max_{l,k} \|\mathbf{B}_{l,k}\|_F^2 = \max_{l,k} \|\mathbf{p}_k\|_2^2 \|\mathbf{p}_{c_l}\|_2^2$, we bound*

$$\Xi \leq \mathbb{E}[R^{k^\star}] \sum_{l=1}^{k^\star+1} \sum_{k \in \mathcal{I}_l^{\text{legal}}} \Big(\frac{4}{\beta}\Big)^2 C_{\text{pos}} (\alpha_l^{k^\star}(k))^2 = \frac{16}{\beta^2} C_{\text{pos}} \, p_{\text{succ}} \sum_{l=1}^{k^\star+1} S_l. \tag{63}$$

*Absorbing $c_\alpha^2$ into the constant (redefining the front coefficient) matches the displayed bound on $\Xi$. Finally, recall the exact identity*

$$\mathbb{E}\big\|\widehat{\mathbf{g}}_n^{k^\star} - \boldsymbol{\mu}\big\|_F^2 = \frac{1}{n}\Big(\Xi - \|\boldsymbol{\mu}\|_F^2\Big).$$

*By Jensen's inequality $\|\mathbb{E}[\mathbf{G}]\|_F^2 \leq \mathbb{E}\|\mathbf{G}\|_F^2$, we have $\|\boldsymbol{\mu}\|_F^2 \leq \Xi$, so $\Xi - \|\boldsymbol{\mu}\|_F^2 \leq \Xi$. Therefore, for upper bounds it is valid to drop the negative term and use*

$$\mathbb{E}\big\|\widehat{\mathbf{g}}_n^{k^\star} - \boldsymbol{\mu}\big\|_F^2 \leq \frac{1}{n} \Xi \leq \frac{1}{n} \frac{16 \, C_{\text{pos}}}{\beta^2} \, p_{\text{succ}} \sum_{l=1}^{k^\star+1} \frac{1}{d_l}.$$

*Remark on the SNR line. From the variance bound just proved,*

$$\mathbb{E}\big\|\widehat{\mathbf{g}}_n^{k^\star} - \boldsymbol{\mu}\big\|_F^2 \leq \frac{C}{n} \, p_{\text{succ}} \sum_{l=1}^{k^\star+1} \frac{1}{d_l}$$

*for a constant $C$ depending only on $(\beta, C_{\text{pos}})$ and the near-uniformity constant. Therefore*

$$\text{SNR} = \frac{\|\boldsymbol{\mu}\|_F^2}{\mathbb{E}\|\widehat{\mathbf{g}}_n^{k^\star} - \boldsymbol{\mu}\|_F^2} \geq \frac{n \|\boldsymbol{\mu}\|_F^2}{C \, p_{\text{succ}} \sum_l d_l^{-1}}.$$

*If, in addition, one has a non-degeneracy bound $\|\boldsymbol{\mu}\|_F^2 \leq C' \, p_{\text{succ}}$ with $C'$ independent of depth (e.g., when the coefficient mass $\sum_{l,k} \mathbb{E}[|\alpha_l \eta_l|]$ remains $\mathcal{O}(\sum_l d_l^{-1})$), this yields the crude scaling*

$$\text{SNR} = \mathcal{O}\Big(\frac{n \, p_{\text{succ}}}{\sum_l d_l^{-1}}\Big),$$

*which reduces to $\mathcal{O}(n \, p_{\text{succ}} \, d/(k^\star + 1))$ for $d_l \equiv d$. The main message is that under 2S-ART, $p_{\text{succ}}$ decays exponentially with depth, making SNR scale poorly with $k^\star$ unless $n$ grows accordingly.* $\quad\square$

**Theorem 10** (Per-step REINFORCE sample size for block dominance). *Fix step $\ell$ and confidence $\delta \in (0,1)$. Let $|\mathcal{I}_\ell| = \Theta(d)$ and let $i_l^\star \in \mathcal{I}_\ell$ denote the unique correct child. For any $j \in \mathcal{I}_\ell$, given $n$ i.i.d. pairs $(\mathbf{x}^{(s)}, \tau^{(s)})$, define the empirical block mean*

$$\hat{\mu}_{\ell,j} := \frac{1}{n} \sum_{s=1}^n R^{k^\star} \alpha_\ell^{(s)}(j) \eta_\ell^{(s)}(j), \qquad R^{k^\star} \in \big\{ \mathbf{R}_{\mathbf{x}}^{f_{S_\star}}(\cdot), \, \mathbf{R}_{\mathbf{x}}^{\mathcal{F}_{S_\star}}(\cdot) \big\}.$$

*Consider three scenario:*

(A) *No-curriculum ($R_{\mathbf{x}}^{f_{S_\star}}(\cdot)$ as oracle; Algorithm. 1).*

(B) *Depth-increasing Curriculum ($\mathbf{R}_{\mathbf{x}}^{\mathcal{F}_{S_\star}}(\cdot)$ as oracle; Algorithm. 2).*

(C) *Hint-decreasing Curriculum ($\mathbf{R}_{\mathbf{x}}^{\mathcal{F}_{S_\star}}(\cdot)$ as oracle; Algorithm. 3).*

*Then, for the target $f_{S_\star} \in \mathcal{F}_{\text{2S-ART}}$ with $|S_\star| = k^\star + 1$, $S_\star = \{i_1^\star, i_2^\star, ..., i_{k^\star}^\star, d+1\}$, the sample complexity $n_\ell(\delta)$ to ensure with probability at least $1 - \delta$, $\max_{j \in \mathcal{I}_\ell} |\hat{\mu}_{\ell,j} - \mathbb{E}[\hat{\mu}_{\ell,j}]| \leq \Theta(\mu_{\ell,i_l^\star} - \max_{j \neq i_l^\star} \mu_{\ell,j})$ in case A $-$ C is*

$$\text{(A)} \; n_\ell(\delta) \geq \tilde{\Omega}\Big(d^{2(k^\star + 2 - \ell) - 2} (1 - \rho_{\text{spur}}^{\text{sup},>\ell})^{-2}\Big),$$

$$\text{(B)} \; n_\ell(\delta) \leq \tilde{O}\Big(d^2 (1 - \rho_{\text{spur}}^{\text{sup},\ell})^{-2}\Big), \tag{64}$$

$$\text{(C)} \; n_\ell(\delta) \leq \tilde{O}\Big(d^2 (1 - \rho_{\text{spur}}^{\text{sup},\, k^\star + 1 - \ell})^{-2}\Big),$$

*where $\tilde{\Omega}, \tilde{O}$ hide polylogarithmic factors in $d$ and $1/\delta$ and absolute constants depending on $\beta$.*

**Proof.** *We quantify, at a fixed step $\ell$, how many i.i.d. inputs $\mathbf{x}^{(s)} \sim \text{Unif}([K]^d)$ suffice so that, with probability at least $1 - \delta$, a one-step REINFORCE update increases the correct block more than any competitor by at least a fixed margin $C > 0$.*

***Distributional conventions and the REINFORCE gradient.*** *For each i.i.d. draw $\mathbf{x}^{(s)} \sim \text{Unif}([K]^d)$, the base model $\text{TF}_{\text{base}}$ samples a trajectory $\tau^{(s)} \sim p_{\mathbf{W}}(\cdot \mid \mathbf{x}^{(s)})$ (independently across $s$), where $\mathbf{W}$ is the current parameter. We write*

$$\mathbb{E}[\cdot] := \mathbb{E}_{\mathbf{x} \sim \text{Unif}([K]^d)} \mathbb{E}_{\tau \sim p_{\mathbf{W}}(\cdot | \mathbf{x})}[\cdot]$$

*for the population expectation. The outcome reward is*

$$R^{k^\star} \in \big\{ \mathbf{R}_{\mathbf{x}}^{f_{S_\star}}(\cdot), \, \mathbf{R}_{\mathbf{x}}^{\mathcal{F}_{S_\star}}(\cdot) \big\}, \qquad R^{k^\star} \in \{0,1\}.$$

*By Lemma 5 and Lemma 3, the REINFORCE gradient admits the block decomposition*

$$\nabla_{\mathbf{W}} \mathcal{J}_{\text{REINFORCE}} = \mathbb{E}\Big[ R^{k^\star} \sum_{t=1}^{k^\star+1} \sum_{k \in \mathcal{I}_t} \alpha_t(k) \eta_t(k) \, \mathbf{p}_k \mathbf{p}_{c_t}^\top \Big].$$

*Projecting onto the step-$\ell$ block $\mathbf{B}_{\ell,j} := \mathbf{p}_j \mathbf{p}_{c_\ell}^\top$ gives*

$$\underbrace{\big\langle \nabla_{\mathbf{W}} \mathcal{J}_{\text{REINFORCE}}, \, \mathbf{B}_{\ell,j} \big\rangle}_{=: \, \mu_{\ell,j}} = \mathbb{E}\big[ R^{k^\star} \alpha_\ell(j) \eta_\ell(j) \big]. \tag{65}$$

**Setup and block-wise variables.** *At step $\ell$, fix $j \in \mathcal{I}_\ell$ and define the single-sample block variable for $(\mathbf{x}, \tau)$:*

$$X_{\ell,j} := R^{k^\star} \alpha_\ell(j) \eta_\ell(j), \qquad R^{k^\star} \in \{0, 1\},$$

*where $\alpha_\ell(j), \eta_\ell(j)$ are computed from $(\mathbf{x}, \tau)$ and the current $\mathbf{W}$. By equation 65,*

$$\mu_{\ell,j} := \mathbb{E}[X_{\ell,j}] = \left\langle \nabla_{\mathbf{W}} \mathcal{J}_{\mathrm{REINFORCE}}, \mathbf{B}_{\ell,j} \right\rangle.$$

*Given $n$ i.i.d. pairs $(\mathbf{x}^{(s)}, \tau^{(s)})$, define the empirical estimator and its range bound*

$$\hat{\mu}_{\ell,j} := \frac{1}{n} \sum_{s=1}^{n} X_{\ell,j}^{(s)}, \qquad |X_{\ell,j}| \leq B = \Theta(\tfrac{4}{d\beta}),$$

*where $|\eta_\ell(j)| \leq 4/\beta$ (Lemma 3) and $\alpha_\ell(j) = \Theta(d^{-1})$. Note $B$ is the Hoeffding range parameter; it is* independent *of the margin defined below.*

*Let $i_l^\star$ be the unique correct child at depth $\ell$. We say the empirical gradient exhibits* block dominance *with margin $C$ if*

$$\hat{\mu}_{\ell,i_l^\star} - \max_{j \in \mathcal{I}_\ell \setminus \{i_l^\star\}} \hat{\mu}_{\ell,j} \geq C.$$

*By Lemma 6, this implies the increase of $s_\ell(i_l^\star)$ exceeds all competitors by at least $\eta C$ (up to $\mathcal{O}(\eta^2)$), hence strictly enlarging the step-$\ell$ attention gap.*

**Concentration.** *Hoeffding's inequality for bounded variables (Lemma 1) yields for any fixed $j$ (expectation over the joint randomness of $(\mathbf{x}^{(s)}, \tau^{(s)})$):*

$$\Pr\left( |\hat{\mu}_{\ell,j} - \mu_{\ell,j}| \geq t \right) \leq 2 \exp\left( -\frac{2nt^2}{B^2} \right).$$

*A union bound over $|\mathcal{I}_\ell| = \Theta(d)$ blocks gives that, with probability $\geq 1 - \delta$,*

$$\max_{j \in \mathcal{I}_\ell} |\hat{\mu}_{\ell,j} - \mu_{\ell,j}| \leq t_n := \frac{B}{\sqrt{2n}} \sqrt{\log \frac{2d}{\delta}}. \tag{66}$$

*If the* population *margin*

$$\gamma_\ell := \mu_{\ell,i_l^\star} - \max_{j \neq i_l^\star} \mu_{\ell,j} > 0,$$

*then setting $t_n \leq \gamma_\ell/4$ guarantees dominance with margin $C = \gamma_\ell/2$.*

**Master relation and the choice of $C$.** *From equation 66, requiring empirical dominance with margin $C \leq \gamma_\ell/2$ is ensured by taking $t_n \leq \gamma_\ell/2$. Solving*

$$\frac{B}{\sqrt{2n}} \sqrt{\log \frac{2d}{\delta}} \leq \frac{\gamma_\ell}{2}$$

*for $n$ yields the explicit sample-size condition*

$$n \geq \frac{2B^2}{\gamma_\ell^2} \log \frac{2d}{\delta}.$$

*Throughout this proof we fix $C := \gamma_\ell/2$ so the target constant margin is explicit.*

**Population margin under three settings.** *Using Lemma 7 and copied-PART near-uniformity $(\alpha_\ell(i_l^\star) = \Theta(1/d))$, with expectations taken over $\mathbf{x} \sim \mathrm{Unif}([K]^d)$ and $\tau \sim p_{\mathbf{W}}(\cdot \mid \mathbf{x})$:*

*(A) **No curriculum; terminal oracle $\mathbf{R}_{\mathbf{x}}^{f_{S^\star}}(\cdot)$.** Conditioning on a correct prefix to $\ell$, completing the true suffix (including EOS) has probability $\Theta(d^{-(k^\star+1-\ell)})$, while any deviation may be accepted with probability at most $\rho_{\mathrm{spur}}^{\mathrm{sup},>\ell}$. Under copied-PART near-uniformity $\alpha_\ell(i_l^\star) = \Theta(1/d)$ and $4/\beta \geq \eta_l(i_l^\star) \geq O(1/\beta) \geq \eta_l(j), j \neq i_l^\star$, Lemma 7 implies*

$$q_{\ell|\mathrm{corr}} - q_{\ell|\mathrm{wrong}} = \Theta(p_{\mathrm{tail}}(\ell+1)\left(1 - \rho_{\mathrm{spur}}^{\mathrm{sup},>\ell}\right)), \qquad p_{\mathrm{tail}}(\ell+1) = \Theta(d^{-(k^\star+1-\ell)}),$$

*Consequently, in case (A) we have the matching scaling over the expected margin*

$$\gamma_\ell^{\mathsf{A}} = \Theta\left(\frac{d^{-(k^\star+2-\ell)}}{\beta}\left(1 - \rho_{\mathrm{spur}}^{\mathrm{sup},>\ell}\right)\right),$$

*up to absolute constants, which will be used to derive the* lower bound *on $n_\ell(\delta)$.*

(B) **Depth-increasing curriculum; family oracle $\mathbf{R}_{\mathbf{x}}^{\mathcal{F}_{S_\star}}(\cdot)$ with external truncation at depth $\ell$.** Similar to Thm. 12, $\alpha_{i_l^\star}^{(\ell)} \geq 1 - \eta_\ell = \Theta(d^{-1})$ and any wrong child has acceptance at most $\rho_{\text{spur}}^{\sup,\ell}$. Also, we see that $4/\beta \geq \eta_l(i_l^\star) \geq O(1/\beta) \geq \eta_l(j), j \neq i_l^\star$. Thus

$$\gamma_\ell^{\mathsf{B}} = \Theta\Big(\frac{1}{\beta\, d^2}\, (1 - \rho_{\text{spur}}^{\sup,\ell})\Big).$$

(C) **Hint-decreasing curriculum (reverse indexing);** identical to (B) with $\ell \mapsto k^\star+1-\ell$, such that $\{\gamma_\ell^{\mathsf{C}}\}_{\ell=1}^{k^\star+1} = \{\gamma_{k^\star+1-\ell}^{\mathsf{B}}\}_{\ell=1}^{k^\star+1}$.

*From equation 66, taking $t_n \leq \gamma_\ell/2$ ensures block dominance with margin $\gamma_\ell/2$. Solving for $n$ with range bound $B = 4/(\beta d)$ gives the master condition $n \geq (2B^2/\gamma_\ell^2) \log \frac{2d}{\delta}$. For case (A), substituting the upper bound on $\gamma_\ell$ above yields the displayed lower bound on $n_\ell$. For cases (B) and (C), substituting the lower bounds on $\gamma_\ell$ yields the displayed upper bounds on $n_\ell$. Here $B$ is purely the bounded-range constant of $X_{\ell,j}$ (Hoeffding parameter); $\gamma_\ell$ is the population margin and does not influence the range. Under the near-uniformity and spurious-success cap assumptions used above, summarizing (A) as a $\Theta(\cdot)$ scaling for $\gamma_\ell$ is appropriate; we keep one-sided bounds when only sufficiency is needed for (B)/(C).* $\qquad\square$

**Theorem 11** (One-pass and online per-step schedules achieve per-step margin thresholds)**.** *For $\forall \varepsilon > 0$, consider the target per-step margin thresholds $\{\Gamma_\ell\}_{\ell=1}^{k^\star+1}$ for Lemma 4, with $\varepsilon_l = \varepsilon/(k^\star + 1)$, namely*

$$\Gamma_l = \log\big(\frac{d-1}{(\frac{1}{2} + \frac{\beta}{2}\log(\frac{(K+1)((k^\star+1)-\varepsilon)}{\varepsilon}))^{-1} - 1}\big) \leq \Theta(\log(d))$$

*For each step $\ell$, let $n_\ell(\delta)$ be the per-step sample size from Theorem 10 that guarantees, with probability at least $1 - \delta$, $\max_{j \in \mathcal{I}_\ell} |\hat{\mu}_{\ell,j} - \mathbb{E}[\hat{\mu}_{\ell,j}]| \leq (\mu_{\ell,i_l^\star} - \max_{j \neq i_l^\star} \mu_{\ell,j})/4$, guaranteeing that the empirical block-dominance margin to be no less than the thresholds $\{\gamma_\ell^{\mathsf{A}}/2\}_{\ell=1}^{k^\star+1}, \{\gamma_\ell^{\mathsf{B}}/2\}_{\ell=1}^{k^\star+1}, \{\gamma_\ell^{\mathsf{C}}/2\}_{\ell=1}^{k^\star+1}$.*

*There exist learning-rate schedules under which, with probability at least $1 - \delta$, the post-update per-step logit gaps satisfy $\Delta_\ell \geq \Gamma_\ell$ for all $\ell \in \{1, \ldots, k^\star+1\}$, so that Lemma 4 applies. We state them for three settings:*

*(A) No curriculum (Algorithm. 1); terminal oracle $\mathbf{R}_{\mathbf{x}}^{f_{S_\star}}(\cdot)$. Draw a single batch of size*

$$n_0 \geq \max_{\ell \in [k^\star+1]} n_\ell\big(\delta/(k^\star+1)\big) = \tilde{\Omega}\big(d^{2k^\star+4}\,(1 - \rho_{\text{spur}}^{\sup,>1})^{-2}\big),$$

*where the last equality takes the worst step (the first decision), and the extra union over steps would not influence the lower bound. Consider perform a single gradient step (one-shot) with learning rate $\eta^{\mathsf{A}} = \Theta(\beta \log(d) d^{k^\star+1}(1 - \rho_{\text{spur}}^{\sup,>1})^{-1}) \geq \Theta(\Gamma_\ell/(\gamma_1^{\mathsf{A}}/2))$. Guarantee: with probability at least $1 - \delta$, after this one update we have $\Delta_\ell \geq \Gamma_\ell$ for all $\ell$, including the final EOS step.*

*(B) Depth-increasing curriculum (Algorithm. 2); family oracle $\mathbf{R}_{\mathbf{x}}^{\mathcal{F}_{S_\star}}(\cdot)$ (external truncation). Sampling and updates are online per step: for $\ell = 1, 2, \ldots, k^\star+1$ do (i) draw $n_\ell = \tilde{\Theta}\big(d^2(1 - \rho_{\text{spur}}^{\sup,\ell})^{-2}\big)$ fresh samples; (ii) take one gradient step with learning rate $\eta_\ell^{\mathsf{B}} = \Omega(\beta \log(d) d^2(1 - \rho_{\text{spur}}^{\sup,\ell})^{-1}) \geq \Theta(\Gamma_l/(\gamma_\ell^{\mathsf{B}}/2))$. Guarantee: with probability at least $1 - \delta$, after $k^\star+1$ updates, $\Delta_\ell \geq \Gamma_\ell$ holds for all steps. The total sample size is $\sum_{\ell=1}^{k^\star+1} n_\ell = \tilde{\Theta}\big((k^\star + 1)d^2(1 - \rho_{\text{spur}}^{\max})^{-2}\big)$.*

*(C) Hint-decreasing curriculum (Algorithm. 3); family oracle $\mathbf{R}_{\mathbf{x}}^{\mathcal{F}_{S_\star}}(\cdot)$. Sampling and updates are online per step in reverse: for $\ell = k^\star+1, \ldots, 1$ do (i) draw $n_\ell = \tilde{\Theta}\big(d^2(1 - \rho_{\text{spur}}^{\sup, k^\star+1-\ell})^{-2}\big)$ fresh samples; (ii) take one gradient step with learning rate $\eta_\ell^{\mathsf{C}} = \Omega(\beta \log(d) d^2(1 - \rho_{\text{spur}}^{\sup, k^\star+1-\ell})^{-1}) \geq \Theta(\Gamma_l/(\gamma_\ell^{\mathsf{C}}/2))$. Guarantee: with probability at least $1 - \delta$, after $k^\star+1$ updates, $\Delta_\ell \geq \Gamma_\ell$ holds for all steps. The total sample size is $\sum_{\ell=1}^{k^\star+1} n_\ell = \tilde{O}\big((k^\star + 1)d^2(1 - \rho_{\text{spur}}^{\max})^{-2}\big)$.*

**Proof.** *The proof is direct based on OBI (Lemma 3), as well as the convergence conditions in Lemma 6 for a target $\varepsilon > 0$.* $\qquad\square$

**Proof.** *Proof of Theorem. 3. The proof follows by collaborating Theorem. 11 and Lemma 6.* $\qquad\square$

---

**Algorithm 1** No-Curriculum REINFORCE Finetuning (A)

---

1: Inputs: input distribution $\mathbf{x} \sim \text{Unif}([K]^d)$, error budget $\varepsilon > 0$, target length $k^\star$, terminal oracle $\mathbf{R}_{\mathbf{x}^\star}^{f_{S_\star}}(\cdot)$, sample complexity $n_0 = \tilde{\Omega}\big(d^{2k^\star+4}(1 - \rho_{\text{spur}}^{\text{sup},>1})^{-2}\big)$, learning rate $\eta^{\mathsf{A}} = \Theta(\beta \log(d) d^{k^\star+1}(1 - \rho_{\text{spur}}^{\text{sup},>1})^{-1})$.

2: Initialize transformer parameters $\mathbf{W}^{(0)}$.

3: Draw a single batch of $n_0$ i.i.d. samples $(\mathbf{x}^{(s)}, \tau^{(s)})_{s=1}^{n_0}$ where $\mathbf{x}^{(s)} \sim \text{Unif}([K]^d)$ and $\tau^{(s)} \sim p_{\mathbf{W}^{(0)}}(\cdot \mid \mathbf{x}^{(s)})$.

4: Compute empirical REINFORCE gradient:

5: $\quad \widehat{\mathbf{g}} = \frac{1}{n_0} \sum_{s=1}^{n_0} \mathbf{R}_{\mathbf{x}^{(s)}}^{f_{S_\star}}(\hat{\mathbf{p}}^{z_{k^\star}}(\tau_{-2}^{(s)})) \sum_{\ell=1}^{k^\star+1} \nabla_{\mathbf{W}} \log \pi_{\mathbf{W}^{(0)}}(i_\ell^{(s)} \mid \mathbf{x}^{(s)}, \hat{\mathbf{p}}^{z_{1:\ell}})$.

6: Update parameters: $\mathbf{W}^{(1)} = \mathbf{W}^{(0)} + \eta^{\mathsf{A}} \cdot \widehat{\mathbf{g}}$.

7: Return $\mathbf{W}^{(1)}$.

---

**Algorithm 2** Depth-Increasing Curriculum REINFORCE Finetuning (B)

---

1: Inputs: input distribution $\mathbf{x} \sim \text{Unif}([K]^d)$, error budget $\varepsilon > 0$, target length $k^\star$, family oracle $\mathbf{R}_{\mathbf{x}}^{\mathcal{F}_{S_\star}}(\cdot)$, per-step sample complexity $n_\ell = \tilde{\Theta}\big(d^2(1 - \rho_{\text{spur}}^{\text{sup},\ell})^{-2}\big)$, learning rates $\eta_\ell^{\mathsf{B}} = \Omega(\beta \log(d) d^2 (1 - \rho_{\text{spur}}^{\text{sup},\ell})^{-1})$.

2: Initialize transformer parameters $\mathbf{W}^{(0)}$.

3: **for** $\ell = 1$ to $k^\star + 1$ **do**

4: $\quad$ Draw fresh samples $(\mathbf{x}^{(s)}, \tau^{(s)})_{s=1}^{n_\ell}$ where $\mathbf{x}^{(s)} \sim \text{Unif}([K]^d)$ and $\tau^{(s)} \sim p_{\mathbf{W}^{(\ell-1)}}(\cdot \mid \mathbf{x}^{(s)})$.

5: $\quad$ If $\ell < k^\star + 1$, for each sample $s$, truncate trajectory at length $\ell + 1$:

6: $\quad\quad$ If $\tau^{(s)}$ contains EOS at position $\leq \ell$, discard sample.

7: $\quad\quad$ Otherwise, append EOS to create truncated sequence $\tau_{\text{trunc}}^{(s)}$.

8: $\quad$ If $\ell = k^\star + 1$, $\tau_{\text{trunc}}^{(s)} = \tau^{(s)}$.

9: $\quad$ Compute empirical REINFORCE gradient for step $\ell$:

10: $\quad\quad \widehat{\mathbf{g}}_\ell = \frac{1}{n_\ell} \sum_{s=1}^{n_\ell} \mathbf{R}_{\mathbf{x}^{(s)}}^{\mathcal{F}_{S_\star}}(\tau_{\text{trunc}}^{(s)}) \nabla_{\mathbf{W}} \log \pi_{\mathbf{W}^{(\ell-1)}}(i_\ell^{(s)} \mid \mathbf{x}^{(s)}, \hat{\mathbf{p}}^{z_{1:\ell}})$.

11: $\quad$ Update parameters: $\mathbf{W}^{(\ell)} = \mathbf{W}^{(\ell-1)} + \eta_\ell^{\mathsf{B}} \cdot \widehat{\mathbf{g}}_\ell$.

12: **end for**

13: Return $\mathbf{W}^{(k^\star+1)}$.

---

**Algorithm 3** Hint-Decreasing Curriculum REINFORCE Finetuning (C)

---

1: Inputs: input distribution $\mathbf{x} \sim \text{Unif}([K]^d)$, error budget $\varepsilon > 0$, target length $k^\star$, family oracle $\mathbf{R}_{\mathbf{x}}^{\mathcal{F}_{S_\star}}(\cdot)$, per-step sample complexity $n_\ell = \tilde{\Theta}\big(d^2(1 - \rho_{\text{spur}}^{\text{sup}, k^\star+1-\ell})^{-2}\big)$, learning rates $\eta_\ell^{\mathsf{C}} = \Omega(\beta \log(d) d^2 (1 - \rho_{\text{spur}}^{\text{sup}, k^\star+1-\ell})^{-1})$.

2: Initialize transformer parameters $\mathbf{W}^{(0)}$.

3: **for** $\ell = 1$ down to $k^\star + 1$ **do**

4: $\quad$ Draw fresh samples $(\mathbf{x}^{(s)}, \tau^{(s)})_{s=1}^{n_\ell}$ where $\mathbf{x}^{(s)} \sim \text{Unif}([K]^d)$ and $\tau^{(s)} \sim p_{\mathbf{W}^{(k^\star+1-\ell)}}(\cdot \mid \mathbf{x}^{(s)})$.

5: $\quad$ For each sample $s$, provide hint prefix of length $k^\star + 1 - \ell$:

6: $\quad\quad$ Let $\mathbf{h}^{(s)} = (i_1^\star, i_2^\star, \ldots, i_{k^\star+1-\ell}^\star)$ be the correct prefix.

7: $\quad\quad$ Generate remaining tokens $\tau_{\text{hint}}^{(s)}$ from $\mathbf{W}^{(k^\star+1-\ell)}$ conditioned on $\mathbf{h}^{(s)}$.

8: $\quad$ Compute empirical REINFORCE gradient for step $\ell$:

9: $\quad\quad \widehat{\mathbf{g}}_\ell = \frac{1}{n_\ell} \sum_{s=1}^{n_\ell} \mathbf{R}_{\mathbf{x}^{(s)}}^{\mathcal{F}_{S_\star}}(\tau_{\text{hint}}^{(s)}) \nabla_{\mathbf{W}} \log \pi_{\mathbf{W}^{(k^\star+1-\ell)}}(i_\ell^{(s)} \mid \mathbf{x}^{(s)}, \hat{\mathbf{p}}^{z_{1:\ell}})$.

10: $\quad$ Update parameters: $\mathbf{W}^{(k^\star+2-\ell)} = \mathbf{W}^{(k^\star+1-\ell)} + \eta_\ell^{\mathsf{C}} \cdot \widehat{\mathbf{g}}_\ell$.

11: **end for**

12: Return $\mathbf{W}^{(k^\star+1)}$.

---

### G.3 PROOF OF TEST-TIME SCALING

**Proposition 1** (Formal Version of the First Item in Thm. 2). *Assume* $\text{TF}_{\text{base}}$ *copies PART probability behavior with a unique correct child per depth. At depth $\ell$, consider FXRS (Alg. 4) to force a chosen visible x-token and BAI (Alg. 5) that, for each $j \in \mathcal{I}_\ell$, repeats FXRS+oracle $m$ times and selects the empirical best arm.*

*Then any $\delta$-correct identification of the ground-truth path using only the terminal oracle $\mathbf{R}_{\mathbf{x}}^{f_{S^\star}}$ under token-only observability requires*

$$T_{data} \;\geq\; \tilde{\Omega}\big(d^{\,2k^\star+1}\,(1-\bar{\rho}_{spur})^{-2}\big),$$

$$T_{comp} \;\geq\; \tilde{\Omega}\big(d^{\,2k^\star+1}\,(1-\bar{\rho}_{spur})^{-2}\big),$$

*where $\bar{\rho}_{spur} := \sup_\ell \rho_{spur}^{sup,>\ell}$. Here $\rho_{spur}^{sup,>\ell} \in [0,1)$ denotes the* terminal-oracle, suffix-level spurious-acceptance *parameter at depth $\ell$; formally, with $\mathcal{H}_{\ell-1}$ to denote the event that the history up to depth $\ell-1$ is correct, $C_\ell(j) := \{$at depth $\ell$, the visible token is forced to $x_{i_\ell} = x_j\}$ for the FXRS forcing event, define the* true-suffix *event*

$$E_{true} := \bigcap_{t=\ell+1}^{k^\star+1} \{J_t = j_t^\star\},$$

*where $j_t^\star$ is the unique correct child at depth $t$ given the correct history (with $j_{k^\star+1}^\star = d+1$). Let $E_{spur}$ be the complement of $E_{true}$ while still yielding acceptance by $\mathbf{R}_{\mathbf{x}}^{f_{S^\star}}$ on the final pre-EOS token. We then defines*

$$\rho_{spur}^{sup,>\ell} := \sup_{j \in \mathcal{I}_\ell}\; \mathbb{E}_{\mathbf{x}}\Big[\Pr_{\text{TF}_{\text{base}}}\big(E_{spur} \mid \mathbf{x}, \mathcal{H}_{\ell-1}, C_\ell(j)\big)\Big],$$

*which upper-bounds, in expectation over $\mathbf{x}$, the probability that a* wrong suffix *(after depth $\ell$) is nonetheless accepted by $\mathbf{R}_{\mathbf{x}}^{f_{S^\star}}$ when the depth-$\ell$ visible token is forced to $x_{i_\ell} = x_j$.*

---

**Algorithm 4** Forced X-token Rejection Sampling (FXRS) under token-only observability

---

1: Inputs: depth $\ell$, committed history $\mathbf{CoT}_{\ell-1}$, legal set $\mathcal{I}_\ell$, candidate $j \in \mathcal{I}_\ell$, model $\text{TF}_{\text{base}}$, oracle $\mathbf{R}_{\mathbf{x}}^{f_{S^\star}}$, max trials $T_{\max} = \Theta(d\log(\frac{d}{\delta}))$.
2: **for** $t = 1$ to $T_{\max}$ **do**
3:     From context $\mathbf{CoT}_{\ell-1}$, sample one step with $\text{TF}_{\text{base}}$ to emit $\hat{\boldsymbol{\mu}}^{x_{i_\ell}}$.
4:     **if** $\hat{\boldsymbol{\mu}}^{x_{i_\ell}} = \boldsymbol{\mu}^{x_j}$ **then**
5:         Roll out the remaining suffix using $\text{TF}_{\text{base}}$ until termination (EOS occurs at some depth by the copied PART behavior).
6:         Return the full rollout sequence and **stop**.
7:     **end if**
8: **end for**
9: If no match after $T_{\max}$, return **fail** (increase $T_{\max}$ if needed).

---

---

**Algorithm 5** BAI across depths with FXRS and $\mathbf{R}_{\mathbf{x}}^{f_{S_\star}}$

---

1: Inputs: legal sets $\{\mathcal{I}_\ell\}_{\ell=1}^{k^\star}$, model $\mathrm{TF}_{\mathrm{base}}$, oracle $\mathbf{R}_{\mathbf{x}}^{f_{S_\star}}$, confidence $\delta$.
2: Initialize committed partial CoT $\mathcal{H} \leftarrow ( )$.
3: **for** $\ell = 1$ to $k^\star + 1$ **do**
4:    Set per-arm repetitions $m \leftarrow \Theta\big(d^{2(k^\star+1-\ell)} \log(d/\delta)\big)$.
5:    **for** each $j \in \mathcal{I}_\ell$ **do**
6:       Initialize counter $c_j \leftarrow 0$.
7:       **for** $r = 1$ to $m$ **do**
8:          Run FXRS (Alg. 4) with candidate $j$ to obtain a full rollout (or retry until success). If FXRS fails, repeat until success.
9:          Query $\mathbf{R}_{\mathbf{x}}^{f_{S_\star}}$ on the final pre-EOS token of the rollout; record one Bernoulli outcome $Y_{j,r} \in \{0,1\}$.
10:         Update $c_j \leftarrow c_j + Y_{j,r}$.
11:       **end for**
12:       Set empirical acceptance $\hat{p}_j \leftarrow c_j/m$.
13:    **end for**
14:    Let $\hat{j}_\ell \leftarrow \arg\max_{j \in \mathcal{I}_\ell} \hat{p}_j$.
15:    Commit depth-$\ell$: resample until $\hat{\boldsymbol{\mu}}^{x_{i_\ell}} = \boldsymbol{\mu}^{x_{\hat{j}_\ell}}$, then compute $\hat{\boldsymbol{\mu}}^{z_\ell} = \mathrm{FFN}_\ell(\hat{\boldsymbol{\mu}}^{x_{i_\ell}}, \mathbf{E}[z_{\ell-1}]_{:d_{\mathrm{X}}})$; update $\mathcal{H} \leftarrow (\mathcal{H}, \hat{\boldsymbol{\mu}}^{x_{\hat{j}_\ell}}, \hat{\boldsymbol{\mu}}^{z_\ell})$.
16: **end for**
17: Return the committed path encoded by $\mathcal{H}$.

---

**Proof.** *Proof of Prop. 1.*

***Step 0 (probability spaces and notation).*** *For any event $A$, we write*

$$\Pr[A] := \mathbb{E}_{\mathbf{x} \sim \mathrm{Unif}([K]^d)}\Big[ \Pr_{\mathrm{TF}_{\mathrm{base}}} (A \mid \mathbf{x})\Big].$$

*That is, all probabilities $\Pr_{\mathrm{TF}_{\mathrm{base}}}(\cdot \mid \mathbf{x})$ are w.r.t. the internal sampling of $\mathrm{TF}_{\mathrm{base}}$, conditional on the fixed input $\mathbf{x}$. Unconditional probabilities/expectations $\Pr[\cdot]$ and $\mathbb{E}_{\mathbf{x}}[\cdot]$ are taken over $\mathbf{x} \sim \mathrm{Unif}([K]^d)$. We use $\mathcal{H}_{\ell-1}$ to denote the event that the history up to depth $\ell-1$ is correct, and we write*

$$C_\ell(j) := \{\text{at depth } \ell, \text{ the visible token is forced to } x_{i_\ell} = x_j\}$$

*for the FXRS forcing event.*

***Step 1 (conditioning and notation).*** *Fix $\mathbf{x}$ and condition on $\mathcal{H}_{\ell-1}$. Let $j_\star$ be the unique correct child at depth $\ell$. For $t \in \{\ell+1, \ldots, k^\star+1\}$, let $J_t$ denote the random child index selected by $\mathrm{TF}_{\mathrm{base}}$ at depth $t$ given the preceding context; conditionally on $\mathcal{H}_{\ell-1}$ and on $C_\ell(j)$, we have $J_t \sim \mathrm{Unif}(\mathcal{I}_t)$ with $|\mathcal{I}_t| = \Theta(d)$, independent across $t$; at $t=k^\star+1$, the correct choice is $d+1$ (EOS).*

***Step 2 (events and acceptance).*** *Define the* true-suffix *event*

$$E_{true} := \bigcap_{t=\ell+1}^{k^\star+1} \{J_t = j_t^\star\},$$

*where $j_t^\star$ is the unique correct child at depth $t$ given the correct history (with $j_{k^\star+1}^\star = d+1$). Let $E_{spur}$ be the complement of $E_{true}$ while still yielding acceptance by $\mathbf{R}_{\mathbf{x}}^{f_{S_\star}}$ on the final pre-EOS token. Then, for fixed $j$ at depth $\ell$,*

$$P_\ell^{term}(j) := \mathbb{E}_{\mathbf{x}}\Big[ \Pr_{\mathrm{TF}_{\mathrm{base}}} (E_{true} \mid \mathbf{x}, \mathcal{H}_{\ell-1}, C_\ell(j)) + \Pr_{\mathrm{TF}_{\mathrm{base}}} (E_{spur} \mid \mathbf{x}, \mathcal{H}_{\ell-1}, C_\ell(j))\Big].$$

***Step 3 (true-suffix probability including EOS).*** *Under the copied PART uniform branching and uniqueness, for $j=j_\star$ we must select the unique correct child at each of the remaining depths, including the terminating EOS choice. Hence, for every $\mathbf{x}$,*

$$\Pr_{\mathrm{TF}_{\mathrm{base}}} (E_{true} \mid \mathbf{x}, \mathcal{H}_{\ell-1}, C_\ell(j_\star)) := \prod_{t=\ell+1}^{k^\star+1} \frac{1}{|\mathcal{I}_t|} := \Theta\big(d^{-(k^\star+1-\ell)}\big).$$

*Taking $\mathbb{E}_{\mathbf{x}}[\cdot]$ preserves the same order.*

***Step 4 (uniform spurious bound).*** *By definition,*

$$\mathbb{E}_{\mathbf{x}}\Big[\Pr_{\mathrm{TF}_{\mathrm{base}}}(E_{spur}\mid \mathbf{x},\mathcal{H}_{\ell-1},C_\ell(j))\Big]\leq \rho_{spur}^{sup,>\ell}<1$$

*for all $j\in\mathcal{I}_\ell$.*

***Step 5 (gap lower bound).*** *Therefore,*

$$P_\ell^{term}(j_\star)-P_\ell^{term}(j)=\mathbb{E}_{\mathbf{x}}\Big[\Pr_{\mathrm{TF}_{\mathrm{base}}}(E_{true}\mid \mathbf{x},\mathcal{H}_{\ell-1},C_\ell(j_\star))\Big]$$

$$+\mathbb{E}_{\mathbf{x}}\Big[\Pr_{\mathrm{TF}_{\mathrm{base}}}(E_{spur}\mid \mathbf{x},\mathcal{H}_{\ell-1},C_\ell(j_\star))-\Pr_{\mathrm{TF}_{\mathrm{base}}}(E_{spur}\mid \mathbf{x},\mathcal{H}_{\ell-1},C_\ell(j))\Big]$$

$$\geq c\,d^{-(k^\star+1-\ell)}-\rho_{spur}^{sup,>\ell},$$

*for some absolute constant $c>0$. Rearranging constants yields the stated $\Theta(\cdot)\cdot(1-\rho_{spur}^{sup,>\ell})$ form.*

***Step 6 (oracle query complexity per depth).*** *Let the one-vs-best acceptance-gap at depth $\ell$ be $\Delta := P_\ell^{term}(j_\star)-\max_{j\neq j_\star}P_\ell^{term}(j) = \Theta(d^{-(k^\star+1-\ell)})(1-\rho_{spur}^{sup,>\ell})$. Viewing each candidate $j\in\mathcal{I}_\ell$ as a Bernoulli arm with mean $P_\ell^{term}(j)$, any $\delta$-correct identification among $|\mathcal{I}_\ell|=\Theta(d)$ arms requires (in expectation)*

$$\Omega(\Delta^{-2}\log(d/\delta))$$

*oracle observations by Lemma 2 and the confidence allocation via union bound (sufficiency for uniform sampling also follows from Lemma 1). Substituting the explicit $\Delta$ gives $\Omega(d^{2(k^\star+1-\ell)}\log(d/\delta))$.*

***Step 7 (overall $T_{data}$).*** *Summing over depths $\ell=1,\ldots,k^\star+1$ gives $T_{data}\geq\tilde\Omega(d^{2k^\star+1}(1-\bar\rho_{spur})^{-2})$.*

***Step 8 (overall $T_{comp}$).*** *In the best case (minimal token emissions) for FXRS, once the forced visible token matches (success on first try), one can immediately append EOS. This uses a constant number of model emissions (exactly 2). Therefore the model-sampling cost is at minimal a constant multiple of the oracle-query cost at that depth (for parity task, the chance is $1/2$, therefore the number to reach is $\Theta(\log(1/\delta))$ for allowable probability $1-o(1/\delta)$), and aggregating over depths yields a $T_{comp}$ lower bound matching the order of $T_{data}$ (up to polylog and spurious factors). Also, the best case for the resampling at Line 15 of Alg. 5 is $k^\star+1$ token emissions, which is smaller than $\tilde\Omega(d^{2k^\star+1}(1-\bar\rho_{spur})^{-2})$, yielding the final $T_{comp}\geq(d^{2k^\star+1}(1-\bar\rho_{spur})^{-2})$.* □

**Theorem 12** (Formal Version of the Second Item in Thm. 2 (complexities in $T_{data}$ and $T_{comp}$))**.** *Assume $\mathrm{TF}_{\mathrm{base}}$ copies PART probability behavior with a unique correct child per depth. Consider the layer-wise procedure in Alg. 6 that, at each depth $\ell$, uses FXRS (Alg. 4) to force a candidate visible x-token for every $j\in\mathcal{I}_\ell$, queries the family oracle $\mathbf{R}_{\mathbf{x}}^{\mathcal{F}_{S_\star}}$ under external truncation, and runs BAI (Alg. 5) to pick the best arm; then commits the chosen $(x_{i_\ell},z_\ell)$ and proceeds to the next depth.*

*Then there exists a choice of per-depth repetitions $m_\ell=\Theta(d^2(1-\rho_{spur}^{sup,\ell})^{-2}\log(d/\delta))$ such that identifying $S_\star$ with probability at least $1-\delta$ is achievable with*

$$T_{data}\leq\tilde{O}((k^\star+1)d^2(1-\rho_{spur}^{\max})^{-2}),\qquad T_{comp}\leq\tilde{O}((k^\star+1)d^3(1-\rho_{spur}^{\max})^{-2}),$$

*where $\rho_{spur}^{\max}:=\max_{\ell\in[k^\star+1]}\rho_{spur}^{sup,\ell}$. Here $\rho_{spur}^{sup,\ell}\in[0,1)$ denotes the family-oracle, per-depth spurious-acceptance parameter under external truncation at depth $\ell$; formally,*

$$\rho_{spur}^{sup,\ell}:=\sup_{\substack{j\in\mathcal{I}_\ell\\j\neq j_\star}}\mathbb{E}_{\mathbf{x}}\Big[\Pr_{\mathrm{TF}_{\mathrm{base}}}(\mathbf{R}_{\mathbf{x}}^{\mathcal{F}_{S_\star}}(\text{accept at depth }\ell)=1\mid\mathbf{x},\mathcal{H}_{\ell-1},C_\ell(j))\Big].$$

*Detailed derivations are given in the proof.*

**Proof.** ***Step 0 (notation and conditioning).*** *We reuse $\mathcal{H}_{\ell-1}$ for the event that the history up to depth $\ell-1$ is correct, and $C_\ell(j)$ for the inline-forcing event at depth $\ell$ that sets the visible token to $x_{i_\ell}=x_j$.*

---

**Algorithm 6** Layer-wise Truncated Accept-Reject (LTAR) with $\mathbf{R}_{\mathbf{x}}^{\mathcal{F}_{S_\star}}$ (BAI with inline forcing and truncation)

---

1: Inputs: legal sets $\{\mathcal{I}_\ell\}_{\ell=1}^{k^\star}$, confidence level $\delta$, per-depth budgets $m_\ell = \Theta\big(d^2(1 - \rho_{\text{spur}}^{\text{sup},\ell})^{-2} \log(\frac{d}{\delta})\big)$, max trials $T_{\max} = \Theta(d\log(\frac{d}{\delta}))$.
2: Initialize committed partial CoT $\mathcal{H} \leftarrow (\ )$.
3: **for** $\ell = 1$ to $k^\star + 1$ **do**
4:     For each $j \in \mathcal{I}_\ell$, estimate acceptance by $m_\ell$ repetitions with inline forcing:
5:        Initialize $c_j \leftarrow 0$.
6:     **for** $r = 1$ to $m_\ell$ **do**
7:        **for** $t = 1$ to $T_{\max}$ **do**
8:           From context $\mathcal{H}$, sample one step with $\text{TF}_{\text{base}}$ to emit $\hat{\boldsymbol{\mu}}^{x_{i_\ell}}$.
9:           **if** $\hat{\boldsymbol{\mu}}^{x_{i_\ell}} = \boldsymbol{\mu}^{x_j}$ **then**
10:              Compute $\hat{\boldsymbol{\mu}}^{z_\ell} \leftarrow \text{FFN}_\ell\big(\hat{\boldsymbol{\mu}}^{x_{i_\ell}}, \ \mathbf{E}[z_{\ell-1}]_{:d_{\mathbf{x}}}\big)$.
11:              Externally append EOS if $l < k^\star + 1$; query $\mathbf{R}_{\mathbf{x}}^{\mathcal{F}_{S_\star}}$ at depth $\ell$; record Bernoulli $Y_{j,r} \in \{0,1\}$.
12:              Update $c_j \leftarrow c_j + Y_{j,r}$ and **break** the retry loop.
13:           **end if**
14:        **end for**
15:        If no match within $T_{\max}$, optionally increase $T_{\max}$ and repeat this repetition.
16:     **end for**
17:     Set empirical acceptance $\hat{p}_j \leftarrow c_j/m_\ell$ for all $j \in \mathcal{I}_\ell$ and pick $\hat{j}_\ell = \arg\max_j \hat{p}_j$.
18:     Commit depth-$\ell$: resample until $\hat{\boldsymbol{\mu}}^{x_{i_\ell}} = \boldsymbol{\mu}^{x_{\hat{j}_\ell}}$, then compute $\hat{\boldsymbol{\mu}}^{z_\ell} = \text{FFN}_\ell(\hat{\boldsymbol{\mu}}^{x_{i_\ell}}, \mathbf{E}[z_{\ell-1}]_{:d_{\mathbf{x}}})$; update $\mathcal{H} \leftarrow (\mathcal{H}, \hat{\boldsymbol{\mu}}^{x_{\hat{j}_\ell}}, \hat{\boldsymbol{\mu}}^{z_\ell})$.
19: **end for**
20: Return the committed path encoded by $\mathcal{H}$.

---

*Given $\mathcal{H}_{\ell-1}$ and $C_\ell(j)$, for $t \geq \ell$ let $J_t$ denote the random child index sampled by $\text{TF}_{\text{base}}$ at depth $t$; under PART-like uniform branching, $J_t \sim \text{Unif}(\mathcal{I}_t)$ with $|\mathcal{I}_t| = \Theta(d)$, independent across $t$, and the correct EOS index is $d+1$ at $t = k^\star + 1$.*

*Define the depth-$\ell$ acceptance event under the family oracle with external truncation as*

$$A_\ell(j) := \{\mathbf{R}_{\mathbf{x}}^{\mathcal{F}_{S_\star}}(\textit{FXRS-forced } j \textit{ at depth } \ell, \textit{ truncated at } z_\ell) = 1\}.$$

*The (marginal) acceptance probability for candidate $j$ is*

$$\alpha_j^{(\ell)} := \mathbb{E}_{\mathbf{x}}\Big[\Pr_{\text{TF}_{\text{base}}}\big(A_\ell(j) \mid \mathbf{x}, \mathcal{H}_{\ell-1}, C_\ell(j)\big)\Big].$$

*By definition of the per-depth spurious parameter (family-oracle, external truncation)*

$$\rho_{\text{spur}}^{\text{sup},\ell} := \sup_{\substack{j \in \mathcal{I}_\ell \\ j \neq j_\star}} \mathbb{E}_{\mathbf{x}}\Big[\Pr_{\text{TF}_{\text{base}}}\big(A_\ell(j) \mid \mathbf{x}, \mathcal{H}_{\ell-1}, C_\ell(j)\big)\Big] < 1,$$

*we have $\alpha_j^{(\ell)} \leq \rho_{\text{spur}}^{\text{sup},\ell}$ for all $j \neq j_\star$. For the unique correct child $j_\star$, we assume*

$$\alpha_{j_\star}^{(\ell)} \geq 1 - \eta_\ell, \qquad \eta_\ell \in [0,1), \tag{67}$$

*which captures any oracle or model non-idealities.*

***Step 1 (acceptance gap and BAI sample complexity).*** *Under PART-like uniform branching, a single inline-forcing repetition succeeds in emitting the visible token $x_{i_\ell} = x_j$ within expected $\Theta(d)$ retries. Conditioned on success (we then deterministically compute $z_\ell$ via $\text{FFN}_\ell$ and externally append EOS before querying), the depth-$\ell$ acceptance probabilities satisfy*

$$\alpha_{j_\star}^{(\ell)} = \Theta\big(d^{-1}\big), \qquad \alpha_j^{(\ell)} \leq c\,d^{-1}\rho_{\text{spur}}^{\text{sup},\ell} \quad (j \neq j_\star),$$

*for some absolute constant $c > 0$, since the correct child is selected with probability $\Theta(d^{-1})$ and wrong children are upper-bounded by the spurious-acceptance parameter. Hence the one-vs-best gap obeys*

$$\Delta_\ell := \alpha_{j_\star}^{(\ell)} - \max_{j \neq j_\star} \alpha_j^{(\ell)} = \Theta(d^{-1}) \cdot \big(1 - \rho_{\text{spur}}^{\text{sup},\ell}\big).$$

*Running BAI at depth $\ell$ with*

$$m_\ell = \Theta\big(\Delta_\ell^{-2} \log(d/\delta)\big)$$

*oracle trials* per arm *identifies $j_\star$ with probability at least $1 - \delta$ by Chernoff bounds and a union bound over the $d$ arms.*

**Step 2 (per-depth reward-oracle query complexity).** *Each successful inline-forcing repetition issues one query to $\mathbf{R}_\mathbf{x}^{\mathcal{F}_{S_\star}}$. With $|\mathcal{I}_\ell|=\Theta(d)$ arms and $m_\ell$ repetitions per arm,*

$$\tilde{O}\big(m_\ell\big) = \tilde{O}\big(\Delta_\ell^{-2}\big) = \tilde{O}\Big(d^2, (1 - \rho_{\mathrm{spur}}^{\mathrm{sup},\ell})^{-2}\Big),$$

*since $\Delta_\ell^{-2} = \Theta\big(d^2 \, (1 - \rho_{\mathrm{spur}}^{\mathrm{sup},\ell})^{-2}\big)$.*

**Step 3 (aggregated $T_{data}$).** *Summing over depths $\ell=1,\ldots,k^\star$ and upper-bounding each per-depth spurious term by $\rho_{\mathrm{spur}}^{\mathrm{max}} := \max_{\ell \in [k^\star+1]} \rho_{\mathrm{spur}}^{\mathrm{sup},\ell}$,*

$$T_{data} \leq \tilde{O}\Big( \sum_{\ell=1}^{k^\star+1} d^2(1 - \rho_{\mathrm{spur}}^{\mathrm{sup},\ell})^{-2} \Big) \leq \tilde{O}\big((k^\star + 1)d^2(1 - \rho_{\mathrm{spur}}^{\mathrm{max}})^{-2}\big).$$

**Step 4 (per-depth model-sampling cost).** *Under LTAR external truncation, each repetition computes $\hat{\boldsymbol{\mu}}^{z_\ell}$ and appends EOS, costing $O(1)$ emissions per retry; with worst-case $O(d \log(d/\delta))$ retries, the per-repetition cost is $O(d \log(d/\delta))$. Hence, per-depth model-sampling cost is*

$$\tilde{O}\big(d \cdot m_\ell\big) = \tilde{O}\Big(d^3 \, (1 - \rho_{\mathrm{spur}}^{\mathrm{sup},\ell})^{-2}\Big).$$

**Step 5 (aggregated $T_{comp}$).** *Summing over depths and upper-bounding by $\rho_{\mathrm{spur}}^{\mathrm{max}}$,*

$$T_{comp} \leq \tilde{O}\Big( \sum_{\ell=1}^{k^\star+1} d^3 \, (1 - \rho_{\mathrm{spur}}^{\mathrm{sup},\ell})^{-2} \Big) \leq \tilde{O}\big((k^\star + 1)d^3 \, (1 - \rho_{\mathrm{spur}}^{\mathrm{max}})^{-2}\big).$$

**Additional commit cost across depths.** *After identifying $\hat{j}_\ell$ at each depth, LTAR performs a commit by resampling until $\hat{\boldsymbol{\mu}}^{x_{i_\ell}} = \boldsymbol{\mu}^{x_{\hat{j}_\ell}}$ and computing $\hat{\boldsymbol{\mu}}^{z_\ell}$, then (optionally) appending EOS. This contributes an extra $\tilde{O}(d)$ emissions per depth in the worst case (Bernstein-type retry bound), for a total $\tilde{O}((k^\star+1)d)$ over all depths. This term is strictly dominated by the $\tilde{O}((k^\star+1)d^3(1-\rho_{\mathrm{spur}}^{\mathrm{max}})^{-2})$ bound derived above, and hence is absorbed into $T_{comp}$.*

**Step 6 (success probability across depths).** *Allocate confidence across depths (e.g., replace $\delta$ by $\delta/(k^\star + 1)$ per depth) or absorb this into polylog factors; combining the steps above yields the stated bounds for $T_{data}$ and $T_{comp}$.* $\qquad\square$

**Remark 13** (Spurious-acceptance parameters and their roles)**.** *We use two families of spurious-acceptance parameters, tied to the oracle being queried and to how the rollout is conditioned:*

- *$\rho_{\mathrm{spur}}^{\mathrm{sup},>\ell} \in [0,1)$ (terminal-oracle, suffix-level): for queries to $\mathbf{R}_\mathbf{x}^{f_{S_\star}}$, after forcing a visible child $j$ at depth $\ell$ via FXRS, it upper-bounds the expected probability (over $\mathbf{x}$ and the internal sampling of $\mathrm{TF}_{\mathrm{base}}$) that a wrong suffix on depths $\ell+1,\ldots,k^\star+1$ is nonetheless accepted by the oracle. We also define a uniform bound $\bar{\rho}_{\mathrm{spur}} := \sup_\ell \rho_{\mathrm{spur}}^{\mathrm{sup},>\ell}$ for aggregating across depths.*
- *$\rho_{\mathrm{spur}}^{\mathrm{sup},\ell} \in [0,1)$ (family-oracle, per-depth): for queries to $\mathbf{R}_\mathbf{x}^{\mathcal{F}_{S_\star}}$ under external truncation at depth $\ell$ (so the queried token is $z_\ell$), it upper-bounds the expected acceptance probability of any wrong child $j \neq j_\star$ at depth $\ell$. We also use $\rho_{\mathrm{spur}}^{\mathrm{max}} := \max_{\ell \in [k^\star+1]} \rho_{\mathrm{spur}}^{\mathrm{sup},\ell}$.*

*These quantities are* not *interchangeable: $\rho_{\mathrm{spur}}^{\mathrm{sup},>\ell}$ refers to suffix-level acceptance under the terminal oracle $\mathbf{R}_\mathbf{x}^{f_{S_\star}}$, while $\rho_{\mathrm{spur}}^{\mathrm{sup},\ell}$ refers to per-depth acceptance under the family oracle $\mathbf{R}_\mathbf{x}^{\mathcal{F}_{S_\star}}$ with truncation. Both families depend on task class parameters (e.g., output alphabet size $K$) and encapsulate the severity of reward hacking. Our bounds carry these terms explicitly: lower bounds involve $(1 - \bar{\rho}_{\mathrm{spur}})^{-2}$; upper bounds involve $\sum_\ell d \, (1 - \rho_{\mathrm{spur}}^{\mathrm{sup},\ell})^{-2}$.*

# H    PROOFS OF PARITY PROBLEM

**Theorem 14** (Representation Theorem of Any Parity Function). *For any $k' \in [d/2]$, any index set $S = S^{k'} \subset [d]$, and the associated parity function $f_{S^{k'}} \in \mathcal{P}_{d,k'}$ defined in equation 7, there exists a transformer*

$$\text{TF}^{(k')}(\cdot; \mathbf{W}^\star)$$

*with parameters $\mathbf{W}^\star$ such that for every $\mathbf{x} \in \{0,1\}^d$, the autoregressive next-token prediction defined in equation 6 produces exactly the subtask decomposition sequence in equation 7, up to the concatenation with positional embeddings.*

*Concretely, let $S^{k'} = \{i_1, \ldots, i_{k'}\}$ and define $z_m$ recursively as in equation 7. Then*

$$\text{TF}^{(k')}(\mathbf{E}[x_1], \ldots, \mathbf{E}[x_d], \mathbf{E}[\text{EOS}]; \mathbf{W}^\star) = \mathbf{E}[z_1],$$

*and more generally for $m = 2, \ldots, k'$,*

$$\text{TF}^{(k')}(\mathbf{E}[x_1], \ldots, \mathbf{E}[x_d], \mathbf{E}[\text{EOS}], \mathbf{E}[z_1], \ldots, \mathbf{E}[z_{m-1}]; \mathbf{W}^\star) = \mathbf{E}[z_m],$$

*where each $\mathbf{E}[z_m] = [\boldsymbol{\mu}^{z_m}, \mathbf{p}_{d+m}]^\top$ is the concatenation of the token embedding $\boldsymbol{\mu}^{z_m}$ and its positional embedding $\mathbf{p}_{d+m}$. Finally, at step $k' + 1$ the model deterministically outputs the EOS embedding:*

$$\text{TF}^{(k')}(\mathbf{E}[x_1], \ldots, \mathbf{E}[x_d], \mathbf{E}[\text{EOS}], \mathbf{E}[z_1], \ldots, \mathbf{E}[z_{k'}]; \mathbf{W}^\star) = \mathbf{E}[\text{EOS}].$$

*Thus, $\text{TF}^{(k')}(\cdot; \mathbf{W}^\star)$ exactly realizes the chain-of-thought subtask decomposition of $f_{S^{k'}}$ via next-token prediction.*

**Theorem 15** (`PART` Behavior of Parity Class). *Consider a base model with Uniform Ordered Transition Probability at each node, where "legal" means both non-repeating and maintaining strictly increasing order ($i_1 < i_2 < i_3 < \cdots$), and where $k = o(d) < \lfloor d/2 \rfloor$ satisfying $d - k = \Theta(d)$.*

*Under this assumption, at each parent node, the model uniformly selects among all legal children nodes. Specifically:*

- *At the Root node: $d$ variables $\{x_1, x_2, \ldots, x_d\}$ are available, each with probability $1/d$*

- *At node $x_i$ in the tree: only variables $\{x_{i+1}, x_{i+2}, \ldots, x_d\}$ and EOS are legal, each with probability $1/(d - i + 1) = \Theta(d^{-1})$.*

- *EOS is always a legal choice at any node, ensuring termination*

*For a given $d$ and $k < \lfloor d/2 \rfloor$, the number of legal CoT sequences of length $k' + 1$ (including the EOS token) that compute parity functions $f_{S^{k'}} \in \mathcal{P}_{d,k'}$ is:*

$$|\mathcal{L}_{k'+1}| = \binom{d}{k'} \tag{68}$$

*This counts the number of ways to choose $k'$ distinct variables from $d$ available variables while maintaining strictly increasing order, followed by an EOS token.*

*Under the Uniform Ordered Transition Probability assumption, the probability of any specific legal CoT sequence of length $k' + 1$ is:*

$$P(\text{specific sequence of length } k' + 1) = O\left(\frac{(\log d)^{k'}}{d^{k'+1}}\right) \tag{69}$$

*When $k' = O\left(\frac{\log d}{\log \log d}\right)$, it holds that $P(\text{specific sequence of length } k' + 1) \leq O(d^{-k'})$*

*The probability of generating any CoT sequence of length $k' + 1$ is:*

$$P(\text{length } k' + 1) = O\left(\binom{d}{k'} \frac{(\log d)^{k'}}{d^{k'+1}}\right) \tag{70}$$

*This probability distribution over CoT lengths exhibits a systematic bias that depends on the relationship between $k'$ and $d$.*

**Proof.** *We analyze the structure of legal CoT sequences systematically:*

***Step 1: Legal Sequence Structure and Length Constraint.*** *A legal CoT sequence that computes a parity function $f_{S^{k'}}$ must have length $k' + 1$ and be structured as:*

$$Root \rightarrow x_{i_1} \rightarrow x_{i_2} \rightarrow \cdots \rightarrow x_{i_{k'}} \rightarrow EOS \tag{71}$$

*where $\{i_1, i_2, \ldots, i_{k'}\}$ are distinct indices from $[d]$ with strictly increasing order ($i_1 < i_2 < \cdots < i_{k'}$), satisfying our constraints for parity problems. Every legal CoT must terminate with EOS to indicate completion of the parity computation. The constraint $k < \lfloor d/2 \rfloor$ ensures that at the maximum length $k + 1$, the only legal next prediction is EOS, preventing unbounded tree growth.*

***Step 2: Counting Legal Sequences of Length*** $k' + 1$. *The number of legal sequences of length $k' + 1$ (including EOS) that compute parity functions $f_{S^{k'}}$ is:*

$$|\mathcal{L}_{k'+1}| = \binom{d}{k'} \tag{72}$$

*This counts:*

- *$\binom{d}{k'}$ ways to choose $k'$ distinct variables from $d$ available variables*
- *The ordering constraint $i_1 < i_2 < \cdots < i_{k'}$ eliminates the need for permutations*
- *Each sequence must end with EOS to be valid, making the total length $k' + 1$*

***Step 3: Transition Probability Calculation.*** *Under the Uniform Ordered Transition Probability assumption, the probability of any specific legal sequence is the product of transition probabilities at each step:*

*For a sequence $Root \rightarrow x_{i_1} \rightarrow x_{i_2} \rightarrow \cdots \rightarrow x_{i_{k'}} \rightarrow EOS$ ($1 \le i_1 < i_2 < \cdots < i_{k'} \le d$):*

- *$Root \rightarrow x_{i_1}$: probability $1/d$ (choosing from $d$ variables)*
- *$x_{i_1} \rightarrow x_{i_2}$: probability $d^{-(d-i_1+1)}$ (choosing from remaining $d - i_1 + 1$ variables)*
- *$x_{i_2} \rightarrow x_{i_3}$: probability $d^{-(d-i_2+1)}$ (choosing from remaining $d - i_2 + 1$ variables)*
- *$\vdots$*
- *$x_{i_{k'-1}} \rightarrow x_{i_{k'}}$: probability $d^{-(d-i_{k-1}+1)}$ (choosing from remaining $d - i_{k'} + 1$ variables)*
- *$x_{i_{k'}} \rightarrow EOS$: probability $d^{-(d-i_k+1)}$.*

*Therefore:*
$$P(\text{specific sequence}) := \mathbb{E}[F] = \mathbb{E}\left[\frac{1}{d(d - i_1 + 1) \cdots (d - i_{k'} + 1)}\right] \tag{73}$$

*For $1 \le k' \le d$, we here serve to show that*

$$\mathbb{E}[F] = O\left(\frac{(\log d)^{k'}}{d^{k'+1}}\right).$$

*By Lemma 9,*

$$\mathbb{E}[F] = \frac{e_{k'}\left(1, \frac{1}{2}, \ldots, \frac{1}{d}\right)}{d \binom{d}{k'}}.$$

*Recall Maclaurin's inequality (monotonicity of symmetric means): for any $a_1, \ldots, a_d \ge 0$, if*

$$S_k(a) := \frac{e_k(a_1, \ldots, a_d)}{\binom{d}{k}} \quad (k = 1, \ldots, d),$$

*then the sequence $\{S_k(a)^{1/k}\}_{k=1}^d$ is nonincreasing; in particular*

$$S_{k'}(a) \le \left(S_1(a)\right)^{k'} \quad \text{for all } k' \ge 1.$$

*Apply this with $a_t = 1/t$ ($t = 1, \ldots, d$). We get*

$$\frac{e_{k'}\left(1, \frac{1}{2}, \ldots, \frac{1}{d}\right)}{\binom{d}{k'}} \leq \left(\frac{e_1\left(1, \frac{1}{2}, \ldots, \frac{1}{d}\right)}{\binom{d}{1}}\right)^{k'} = \left(\frac{H_d}{d}\right)^{k'},$$

*where $H_d = \sum_{t=1}^{d} \frac{1}{t}$ is the $d$-th harmonic number. Hence*

$$\mathbb{E}[F] \leq \frac{1}{d}\left(\frac{H_d}{d}\right)^{k'}.$$

*Using the standard bound $H_d \leq 1 + \log d$ for all $d \geq 1$, we obtain*

$$\mathbb{E}[F] \leq \frac{1}{d}\left(\frac{1 + \log d}{d}\right)^{k'} = O\left(\frac{(\log d)^{k'}}{d^{k'+1}}\right),$$

*which proves the claim.*

*In particular, when $k' = O\left(\frac{\log d}{\log \log d}\right)$, the logarithmic factor in the bound above never exceeds a fixed polynomial in $d$. Indeed, observe that*

$$(\log d)^{k'} \leq (\log d)^{C \cdot \frac{\log d}{\log \log d}} = \exp\left(C \cdot \frac{\log d}{\log \log d} \cdot \log \log d\right) = d^C,$$

*for some absolute constant $C > 0$. Therefore*

$$\mathbb{E}[F] \leq \frac{(\log d)^{k'}}{d^{k'+1}} \leq \frac{d^C}{d^{k'+1}} = O(d^{-k'}),$$

*which shows that in the regime $k' = O\left(\frac{\log d}{\log \log d}\right)$ the expectation decays at least on the order of $d^{-k'}$.*

**Step 4: Probability of Length** $k' + 1$. *Since there are $\binom{d}{k'}$ different sequences of length $k' + 1$, and each has the same probability structure (though with different specific values), the total probability of generating any CoT sequence of length $k' + 1$ is:*

$$P(\text{length } k' + 1) = \sum_{\text{all sequences of length } k'+1} P(\text{specific sequence}) \tag{74}$$

*This can be expressed as:*

$$P(\text{length } k' + 1) = \Theta\left(\binom{d}{k'} \cdot d^{-k'}\right) \tag{75}$$

$\square$

**Lemma 9** (Expected reciprocal product over a random $k'$-subset). *Fix integers $d \geq 1$ and $0 \leq k' \leq d$. Choose indices*

$$1 \leq i_1 < i_2 < \cdots < i_{k'} \leq d$$

*uniformly at random among all $\binom{d}{k'}$ $k'$-subsets of $\{1, \ldots, d\}$, and define*

$$F(i_1, \ldots, i_{k'}) := \frac{1}{d \prod_{j=1}^{k'} (d - i_j + 1)}.$$

*Let the degree-$k'$ elementary symmetric polynomial be*

$$e_{k'}(y_1, \ldots, y_d) := \sum_{1 \leq t_1 < \cdots < t_{k'} \leq d} y_{t_1} \cdots y_{t_{k'}}, \qquad (e_0 \equiv 1),$$

*and write the (generalized) harmonic numbers*

$$H_d := \sum_{t=1}^{d} \frac{1}{t}, \qquad H_d^{(m)} := \sum_{t=1}^{d} \frac{1}{t^m} \quad (m \geq 2).$$

*Then the expectation of F admits the exact closed form*

$$\mathbb{E}[F] = \frac{e_{k'}\left(1, \frac{1}{2}, \ldots, \frac{1}{d}\right)}{d\binom{d}{k'}}$$

*and, for fixed $k'$ as $d \to \infty$, the asymptotic expansion*

$$\mathbb{E}[F] = \frac{1}{d\binom{d}{k'}}\left(\frac{(\log d)^{k'}}{k'!} + \frac{\gamma(\log d)^{k'-1}}{(k'-1)!} + O\big((\log d)^{k'-2}\big)\right) = \frac{(\log d)^{k'}}{d^{k'+1}}\left(1 + O\big(\tfrac{1}{\log d}\big)\right),$$

*where $\gamma$ is the Euler–Mascheroni constant. In particular,*

$$\mathbb{E}[F] \; \asymp \; \frac{(\log d)^{k'}}{d^{k'+1}}.$$

*Moreover, the first few exact instances are*

$$k' = 1: \quad \mathbb{E}[F] = \frac{H_d}{d\binom{d}{1}} = \frac{H_d}{d^2},$$

$$k' = 2: \quad \mathbb{E}[F] = \frac{H_d^2 - H_d^{(2)}}{2d\binom{d}{2}},$$

$$k' = 3: \quad \mathbb{E}[F] = \frac{H_d^3 - 3H_d H_d^{(2)} + 2H_d^{(3)}}{6d\binom{d}{3}}.$$

**Proof.** *Step 1: Reparametrization and identification with an elementary symmetric sum.* Define $t_j := d - i_j + 1$ for $j = 1, \ldots, k'$. Because $i_1 < \cdots < i_{k'}$, we have

$$1 \le t_{k'} < \cdots < t_1 \le d,$$

*so $\{t_1, \ldots, t_{k'}\}$ is exactly a $k'$-subset of $\{1, \ldots, d\}$ and, since the $(i_1, \ldots, i_{k'})$ are chosen uniformly, the unordered set $\{t_1, \ldots, t_{k'}\}$ is also uniformly distributed over all $\binom{d}{k'}$ $k'$-subsets. With this change of variables,*

$$\prod_{j=1}^{k'}(d - i_j + 1) = \prod_{j=1}^{k'} t_j, \qquad F(i_1, \ldots, i_{k'}) = \frac{1}{d} \cdot \frac{1}{\prod_{j=1}^{k'} t_j}.$$

*Taking expectation over all $\binom{d}{k'}$ subsets and recalling the definition of $e_{k'}$,*

$$\mathbb{E}[F] = \frac{1}{d\binom{d}{k'}} \sum_{1 \le t_1 < \cdots < t_{k'} \le d} \frac{1}{t_1 \cdots t_{k'}} = \frac{e_{k'}\left(1, \frac{1}{2}, \ldots, \frac{1}{d}\right)}{d\binom{d}{k'}}.$$

*This proves the exact formula.*

*Step 2: A generating-function expression for $e_{k'}\left(1, \frac{1}{2}, \ldots, \frac{1}{d}\right)$. Consider*

$$G_d(x) := \prod_{t=1}^{d}\left(1 + \frac{x}{t}\right) = \sum_{k=0}^{d} e_k\left(1, \frac{1}{2}, \ldots, \frac{1}{d}\right)x^k.$$

*Taking logarithms and expanding,*

$$\log G_d(x) = \sum_{t=1}^{d}\log\left(1 + \frac{x}{t}\right) = \sum_{m \ge 1}\frac{(-1)^{m+1}}{m}x^m \underbrace{\sum_{t=1}^{d}\frac{1}{t^m}}_{H_d^{(m)}} = \sum_{m \ge 1}\frac{(-1)^{m+1}}{m}H_d^{(m)}x^m.$$

*Hence*

$$G_d(x) = \exp\left(\sum_{m \ge 1}\frac{(-1)^{m+1}}{m}H_d^{(m)}x^m\right),$$

*so the coefficient*

$$e_{k'}\left(1, \tfrac{1}{2}, \ldots, \tfrac{1}{d}\right) = [x^{k'}]\, G_d(x)$$

*is a degree-$k'$ polynomial in the variables $H_d^{(1)}, \ldots, H_d^{(k')}$ with leading term*

$$\frac{(H_d)^{k'}}{k'!} \qquad (H_d = H_d^{(1)}),$$

*and without an $H_d^{k'-1}$ term (this is the standard consequence of Newton's identities / exponential formula; concrete examples for $k' = 2, 3$ are listed in the statement).*

***Step 3: Asymptotics of the ingredients and of the ratio.*** *We use the classical asymptotics*

$$H_d = \log d + \gamma + o(1), \qquad H_d^{(m)} \to \zeta(m) \ (m \geq 2), \qquad \binom{d}{k'} = \frac{d^{k'}}{k'!}\left(1 + O(\tfrac{1}{d})\right), \quad (d \to \infty, \ k' \text{ fixed}).$$

*Plugging these into the polynomial expression for $e_{k'}$ from Step 2 gives*

$$e_{k'}\left(1, \tfrac{1}{2}, \ldots, \tfrac{1}{d}\right) = \frac{(\log d + \gamma + o(1))^{k'}}{k'!} + O\big((\log d)^{k'-2}\big) = \frac{(\log d)^{k'}}{k'!} + \frac{\gamma(\log d)^{k'-1}}{(k'-1)!} + O\big((\log d)^{k'-2}\big).$$

*Therefore*

$$\frac{e_{k'}(1, 1/2, \ldots, 1/d)}{\binom{d}{k'}} = \frac{(\log d)^{k'}}{d^{k'}}\left(1 + O\Big(\frac{1}{\log d}\Big)\right),$$

*and multiplying by the prefactor $1/d$ yields*

$$\mathbb{E}[F] = \frac{(\log d)^{k'}}{d^{k'+1}}\left(1 + O\Big(\frac{1}{\log d}\Big)\right),$$

*which is exactly the asserted asymptotic expansion in the lemma.*

***Step 4 (Optional sanity check).*** *By Maclaurin's inequalities,*

$$\frac{e_{k'}(1, 1/2, \ldots, 1/d)}{\binom{d}{k'}} \leq \left(\frac{e_1}{\binom{d}{1}}\right)^{k'} = \left(\frac{H_d}{d}\right)^{k'} \sim \frac{(\log d)^{k'}}{d^{k'}},$$

*so $\mathbb{E}[F] \leq (\log d)^{k'}/d^{k'+1}$ up to lower-order factors, matching the leading term above.*

$\square$

**Proof.** *Representation for* PART *of parity. We fix the first $d_{\mathcal{X}}$ columns of $\mathbf{K}, \mathbf{Q}$ to zero so that the attention scores are determined by only the positional encodings. This ensures that the trans-former focuses on learning which positions contribute to the parity at each step. $\mathbf{K}, \mathbf{Q}$ are then reparametrized by a single matrix $\mathbf{W} \in \mathbb{R}^{(d_{\mathrm{E}} - d_{\mathcal{X}})^2}$; conversely, the value matrix is set to only preserve the $\mathbf{x}$ component, as follows.*

$$\mathbf{K}^\top \mathbf{Q} = \begin{pmatrix} \mathbf{0}_{d_{\mathcal{X}} \times d_{\mathcal{X}}} & \mathbf{0}_{d_{\mathcal{X}} \times (d_{\mathrm{E}} - d_{\mathcal{X}})} \\ \mathbf{0}_{(d_{\mathrm{E}} - d_{\mathcal{X}}) \times d_{\mathcal{X}}} & \mathbf{W} \end{pmatrix}, \quad \mathbf{V} = \begin{pmatrix} \mathbb{I}_{d_{\mathcal{X}} \times d_{\mathcal{X}}} & \mathbf{0}_{d_{\mathcal{X}} \times (d_{\mathrm{E}} - d_{\mathcal{X}})} \end{pmatrix}.$$

*This type of reparametrization is common in the literature to make dynamical analysis tractable (Kim & Suzuki, 2024; 2025). Then denote $x_{m+d+1} = z_m, m \in [k]$, we have $\hat{\mathbf{z}}_m = \sum_{j=1}^{m-1+d} \sigma_j(\mathbf{w}_m)\boldsymbol{\mu}^{x_j}$ where the softmax scores $\sigma_j(\mathbf{W}_m) = e^{w_{j,m}} / \sum_{\alpha=1}^{m-1+d} e^{w_{\alpha,m}}$.*

***$W$ Matrix for Uniform Ordered Transition Probability.*** *For $0 < m \leq k$, $i_m \neq d+1$*

$$\begin{aligned} \mathbf{p}_{d+1+m} &= \mathbf{p}_{i_m} \\ \mathbb{E}[x_{d+1+m}] &:= [\boldsymbol{\mu}^{z_m}, \mathbf{p}_{i_m}]^\top \\ (\mathbf{p}_{j \leq i_m})\mathbf{W}(\mathbf{p}_{i_m}) &= -\infty \\ (\mathbf{p}_{i_m < j \leq d+1})\mathbf{W}(\mathbf{p}_{i_m}) &= c_{i_m} \end{aligned} \tag{76}$$

*where $z_m = z_{m-1} \oplus x_{i_m}$.*

*For $\mathbb{E}[x_{d+1}] = [\boldsymbol{\mu}^{\text{EOS}}, \mathbf{p}_{d+1}]^\top$,*

$$(\mathbf{p}_{j<d+1})\mathbf{W}(\mathbf{p}_{d+1}) = c_{d+1} \tag{77}$$

*Then, denote $x_{m+d+1} = z_m, m \in [k]$, we have*

$$\boldsymbol{\mu}^{x_{i_m}} = \hat{\boldsymbol{\mu}}^{x_{i_m}} \sim \text{softmax}((\sum_{j<m-1+d} \frac{e^{(\mathbf{p}_j)\mathbf{W}(\mathbf{p}_{i_{m-1}})}}{\sum_{j<m-1+d} e^{(\mathbf{p}_j)\mathbf{W}(\mathbf{p}_{i_{m-1}})}} \boldsymbol{\mu}^{x_j})^\top \mathbf{U}/\beta). \tag{78}$$

*Then $\mathbb{E}[z_m] = \mathbb{E}[x_{m+d+1}] = [\phi(\hat{\boldsymbol{\mu}}^{x_{i_m}} + \boldsymbol{\mu}^{z_{m-1}}), \mathbf{p}_{i_m}]^\top$.* $\qquad\square$

