# OpenReview forum: "Provable Benefit of Curriculum in Transformer Tree-Reasoning Post-Training"
_ICLR.cc/2026/Conference — Submitted to ICLR 2026_

### Official Review · Reviewer_7hYf · 2025-10-20

**Soundness:** 1
**Presentation:** 1
**Contribution:** 1
**Rating:** 0
**Confidence:** 3

**Summary:**

Proposes a framework to prove that curriculum training is provably good for post-training chain-of-thought generation.

**Strengths:**

Unable to identify or validate the strengths of the paper due to the opaque presentation.

**Weaknesses:**

The paper is poorly written all through.  The introduction states the problem in mathematical terms using symbols that are undefined. The notation section re-introduces the symbols but refers to the introduction for their significance.    Some specific examples below.

The problem statement and the assumptions in the introduction (lines 57-79) launch into formulaic equations without any supporting definitions of the quantities involved.   This makes it impossible to decipher the problem being addressed.

Preliminaries and notation (lines 125)  "For each prompt $x$, policies are conditional probability measures $\pi(. | x)$ on the output space."

What does the subscript $k$ of $\pi_k$ mean in line 126?

Assuming the inputs $x$ and the outputs are embedding vectors of real numbers, $\pi$ is a probability density function.   In which case what does the derivative of one probability density function with respect to another (line 127) mean?  And the variable $o$ of line 127 is entirely undefined.

Lines 132-134:  The definition of pass-rate is circular in that it refers to the introduction section which in turn requires the notations to extract meaning.

Theorem 1 of line 135 is meaningless in light of the opaque notation and definitions.  The important notions of a task $k$, and a curriculum of tasks $K$ are undefined.
What is assumed to be "absolute continuous"  in the theorem, line 137?

lines 143-146 of Theorem 1:  Difficult to understand what this means.
"Further assume a complexity–mismatch alignment ...up to harmless logarithmic factors in a confidence parameter"

Figure 2 caption and elsewhere. "secret index" is used but never defined.

**Questions:**

None at this time

---

### Official Review · Reviewer_PGDS · 2025-10-31

**Soundness:** 3
**Presentation:** 3
**Contribution:** 3
**Rating:** 4
**Confidence:** 2

**Summary:**

This paper proposes a theoretical analysis framework to explain the benefit of curriculum learning during LLM post-training. The framework treats CoT generation as a State-Conditioned Autoregressive Reasoning Tree (2S-ART).  The paper show step-wise curriculum post-training can reduce exponential depth dependence into polynomial order. The exponential-to-polynomial from curriculum learning explains why curriculum learning empirically outperforms non-curriculum approaches.

**Strengths:**

1. The paper is well-motivated by addressing a key open question of principled post-training explanation.
2. The proposed 2S-ART framework abstracts the curriculum learning process into a mathematically analyzable tree structure, providing a way to measure the complexity of reasoning.
3. The paper theoretically indicates directions for future research, namely that curriculum learning can reduce the difficulty for models to learn complex tasks.

**Weaknesses:**

1. The paper relies on an exponential complexity assumption, but the source of this exponential growth is unclear: is it due to decoding search (e.g., CoT tree expansion) or the intrinsic difficulty of reasoning tasks? Moreover, there is no empirical evidence that solving harder reasoning problems actually requires exponentially more reasoning steps.
2. The paper provides no empirical results to support the theory — even a small-scale experiment demonstrating that curriculum learning achieves a task with exponentially fewer training steps would greatly strengthen the claim. Without such evidence, the work remains largely theoretical and its practical significance to real LLM training remains unclear.

**Questions:**

Can the authors provide even small-scale experiments showing that curriculum learning reduces training complexity as predicted by the theory?

---

> ### Author Response · Authors · 2025-11-21
>
> Thank you for recognizing the motivation, explanatory value, and potential impact of our framework. We appreciate your positive assessment and would like to address your concerns point-by-point below.
>
> ---
>
> ***W1.** The paper relies on an exponential complexity assumption, but the source of this exponential growth is unclear: is it due to decoding search (e.g., CoT tree expansion) or the intrinsic difficulty of reasoning tasks?*
>
> ---
>
> **A1. Thank you for raising this point. We clarify what is actually assumed in our theory and how it connects to both decoding search and intrinsic task difficulty.**
>
> 1. **Our condition in Theorem 1 is about the *existence* of a “$K$-th-root” curriculum, not that direct learning complexity itself grows exponentially with some parameters.**
>    In Theorem 1, we start from a fixed base policy $\pi_{\text{ref}}$, a hard target optimal policy $\pi^\star$, and their direct learning complexity $\mathcal{C}(\pi^\star \mid \pi_{\text{ref}}) \gg 1$ (e.g., $d^{\Theta(K)}$ for parity task). We then proved that a $K$-step curriculum
>    $\pi_0^\star := \pi_{\text{ref}}, \pi_1^\star, \dots, \pi_K^\star := \pi^\star$ is provably better than direct learning, if the step-wise learning complexities satisfies
>    $\mathcal{C}(\pi_k^\star \mid \pi_{k-1}^\star) =\Theta(\sqrt[K]{\mathcal{C}(\pi^\star \mid \pi_{\text{ref}})}):=\Theta(C^\star)>1$
>    (or the relaxed $(b/p_{\max})K$-th root per Eq.(4)). Thus, we do **not** assume exponential intrinsic difficulty; rather, we assume that the hard task can be factored into $K$ curriculum subtasks whose step-wise difficulties are moderatably large in the order of $K$-th roots of the direct difficulty.
> 2. **Scope of our theory: a sufficient condition highlighting potential, not a universal guarantee.**
>    Theorem 1 provides a *sufficient* condition under which curriculum post-training reduces an exponential-scale direct complexity to polynomial. This is standard theoretical practice: we isolate conditions under which the studied learning algorithms can achieve large gains. We do not claim that every real-world curriculum or every LLM post-training setup will satisfy the $K$-th-root condition; that depends on the actual task family and curriculum design.
> 3. **In our concrete case study, the $K$-th-root condition originates from both CoT tree expansion and intrinsic task difficulty.**
>    Section 2.1 analyzes a 2-layer transformer GD-learned task family, the 2-State Conditioned Autoregressive Reasoning Tree (2S-ART), modeling autoregressive tree-structured reasoning (Yue et al., 2025) and covering tasks such as parity and Countdown (Gandhi et al., 2024). For this family, we prove that their are two natural curricula that *do* satisfy the $K$-th-root condition and empirically exhibit the predicted exponential improvement.
>    Under Definition 2, the base model assigns comparable probability $\Theta(d^{-1})$ to each legal child, making the success probability of generating a depth-$K$ CoT equal to $\Theta(d^{-K})$. The existence of the $K$-th-root curricula arises from **(i)** CoT tree expansion and **(ii)** the intrinsic difficulty of the curriculum tasks. Two natural curricula then yield per-step success rate $\Theta(d^{-1})$, i.e., the $K$-th root of $\Theta(d^{-K})$:
>
>    * **Depth-increasing curricula** (Parashar et al., 2025): tasks increase from generating short prefixes to full depth-$K$ CoTs.
>    * **Hint-decreasing curricula** (Liu et al., 2025; Amani et al., 2025): tasks decrease the hint length from $K-1$ (predict only EOS) to $0$ (generate the full CoT).
>
>    In both cases, each step has success probability $\Theta(d^{-1})$, which in the end yields exactly the $K$-th-root curriculum complexity condition when considering sample complexity of finetuning (Theorem 3) or reward-oracle complexity in test-time scaling (Theorem 4).
>
> ---

---

> ### Author Response · Authors · 2025-11-21
>
> ***W2.** Moreover, there is no empirical evidence that solving harder reasoning problems actually requires exponentially more reasoning steps.*
>
> **A2.** Thank you for the comment. To clarify, we **never** claim that harder reasoning problems require *exponentially more reasoning steps*. This is not an assumption or conclusion made anywhere in our theory.
>
> ---
>
> ***W3.*** *The paper provides no empirical results to support the theory — even a small-scale experiment demonstrating that curriculum learning achieves a task with exponentially fewer training steps would greatly strengthen the claim. Without such evidence, the work remains largely theoretical and its practical significance to real LLM training remains unclear. Can the authors provide even small-scale experiments showing that curriculum learning reduces training complexity as predicted by the theory?*
>
> **A3.** Thank you for the suggestion. We first clarify that our theory does **not** claim that curriculum strategies achieve a task with *exponentially fewer training steps*. Our exponential-improvement results concern complexity—such as finetuning sample complexity or reward-oracle complexity in test-time scaling—not the number of SGD updates. For finetuning, we follow the standard one-step learning argument widely used in theoretical analyses of shallow networks and transformers (Cornacchia and Mossel, 2023; Kim and Suzuki, 2025; Kawata et al., 2025).
>
> In addition, just like Foster et al. (2025), who studied realizable linear MDPs to reveal base‑model coverage effects, and propose algorithms without simulations, our work follows this standard trend and serves as a pure theoretical work to highlight the **theoretical potential** of curriculum post-training, whose empirical benefits have been widely-observed in real LLMs (Parashar et al. (2025); Liu et al., 2025; Shi et al., 2025). As discussed in A1 for Reviewer GLjV, real LLMs exhibit rich, skewed behaviors beyond our simplified framework, so an exact match to the theoretical ratio $K\sqrt[K]{\mathcal{C}(\pi^\star \mid \pi_{\text{ref}})}/\mathcal{C}(\pi^\star \mid \pi_{\text{ref}})$ is not expected. This does not diminish the theoretical contribution—many fields (e.g., quantum algorithms) develop idealized frameworks precisely to articulate theoretical merit.
>
> ---
>
> **Summary**
>
> Thank you again for recognizing that our work is well-motivated and for acknowledging its theoretical contributions. We hope the above responses help clarify the points of misunderstanding. Should the reviewer have further feedback, we would be grateful to continue the productive exchange. Once again, thank you very much for your time and consideration.
>
> ---
>
> ### **References**
>
> Yue et al. Does Reinforcement Learning Really Incentivize Reasoning Capacity in LLMs Beyond the Base Model? NeurIPS 2025
>
> Parashar et al. (2025) Curriculum reinforcement learning from easy to hard tasks improves llm reasoning
>
> Tong et al. (2025). DART-Math: Difficulty-Aware Rejection Tuning for Mathematical Problem-Solving. NeurIPS 2024
>
> Gandhi et al. (2024). Cognitive Behaviors that Enable Self-Improving Reasoners, or, Four Habits of Highly Effective STaRs
>
> Liu et al. (2025)  UFT: Unifying supervised and reinforcement fine-tuning. NeurIPS 2025
>
> Amani et al. (2025) Lower Bounds for Chain-of-Thought Reasoning in Hard-Attention Transformers. ICML 2025
>
> Kim and Suzuki (2025). Transformers Provably Solve Parity Efficiently with Chain of Thought. ICLR 2025
>
> Kawata et al. (2025). From Shortcut to Induction Head: How Data Diversity Shapes Algorithm Selection in Transformers. NeurIPS 2025
>
> Foster et al (2025) Is a Good Foundation Necessary for Efficient Reinforcement Learning? The Computational Role of the Base Model in Exploration. COLT 2025
>
> Shi et al. (2025). Efficient reinforcement finetuning via adaptive curriculum learning.

---

### Official Review · Reviewer_GLjV · 2025-11-02

**Soundness:** 3
**Presentation:** 2
**Contribution:** 2
**Rating:** 6
**Confidence:** 2

**Summary:**

This paper gives a learning‑theoretic account of why curriculum‑style post‑training can help LLM reasoning. It models chain‑of‑thought generation as a two‑states conditioned autoregressive reasoning tree (2S‑ART) and posits a uniform‑branching “base” model that spreads probability over legal next steps. Curriculum stages are formalized as (i) depth‑increasing (longer CoTs) or (ii) hint‑decreasing (shorter prefixes). Under outcome‑only (binary) rewards, the analysis shows curriculum avoids an exponential bottleneck, achieving polynomial sample complexity, while analogous guarantees hold for test‑time scaling by reducing both reward‑oracle calls and sampling cost from exponential to polynomial. Canonical tasks (Parity, Countdown) and a representation theorem connect the 2S‑ART abstraction to transformers.

**Strengths:**

2S‑ART captures stepwise CoT reasoning with legal action sets and state updates, making “prefix curriculum” and “hint curriculum” precise. The paper also establishes exponential‑to‑polynomial separations for both RL fine‑tuning under outcome‑only rewards and for test‑time scaling via curriculum‑aware querying.

**Weaknesses:**

1. Strong base‑model assumptions. The uniform‑branching coverage (“base model assigns comparable mass across legal children”) is convenient for proofs but unrealistic for modern LLMs whose next‑token distributions are highly skewed and prompt‑dependent. The main theorems hinge on this coverage/complexity alignment.
2. Idealized task/trace structure. The framework assumes a single “correct” index path with a known legal‑set policy; many real problems have multiple near‑equivalent paths and context‑dependent constraints. The parity/countdown focus limits external validity.
3. Missing empirical calibration. The theory predicts polynomial scaling benefits; a small empirical study with real RLHF/RFT post‑training on diverse reasoning sets would strengthen the practical relevance.

**Questions:**

See weaknesses

---

> ### Author Response · Authors · 2025-11-21
>
> Thank you for your recognition for our work and its theoretical account of curriculum-style post-training. We're encouraged for your understanding and interest in our work, and would like to address your concerns as below.
>
> ---
> ***W1.** Strong base‑model assumptions. The uniform‑branching coverage (“base model assigns comparable mass across legal children”) is convenient for proofs but unrealistic for modern LLMs whose next‑token distributions are highly skewed and prompt‑dependent. The main theorems hinge on this coverage/complexity alignment.*
>
> ---
> **A1:** Thank you for this feedback. We first clarify that the condition you highlight appears only in Section 2.1, which is a 2S-ART **case study** illustrating how the conditions and conclusions of Theorem 1 naturally hold for a specific transformer tree-reasoning setup.
>
> 1. **Theorem 1 itself is more general and does *not* require this condition**.  In Theorem 1, the condition that neighboring policies in the curriculum $\pi_0,\pi_1,...,\pi_K=\pi^\star$ satisfy $||\pi_k / \pi_{k-1}||\_\infty = \Theta(\sqrt[K]{||\pi^\star / \pi_0||_\infty})$ for all $k\in[K]$ plays the same conceptual role as *uniform-branching coverage* in 2S-ART: both express that each curriculum stage has comparable, manageable difficulty. That is, Theorem 1 — and the additional case study in Appendix E (Theorem 5) — do **not** rely on any tree-specific uniform-branching assumption. Instead, they are stated in terms of a more general per-stage distribution-distance (coverage) condition, which can capture broader scenarios.
> 2. **Why the 2S-ART condition is natural in a theoretical setting.**  As in much theoretical work (e.g., Foster et al., 2025), our aim is to analyze a clean, stylized regime where sharp guarantees are possible, rather than to fully model all behaviors of modern LLMs. In the Section 2.1 2S-ART case study, *uniform-branching coverage* encodes that the base model has learned all subtasks of the same CoT depth to a comparable degree during next-token pretraining (Chen et al., 2025; Li et al., 2025), which is reasonable when these subtasks appear with similar frequency in a reasonably balanced data distribution.
>    We agree that real LLMs often have highly skewed token and task distributions, so this condition will not hold universally — especially when subtasks of equal CoT length appear at very different frequencies and are mastered to different extents. Extending our complexity guarantees to systematically handle non-uniform branching is an important and interesting direction for future work, and **we'd emphasize this more clearly in the revised version**.
>
> ---
>
> ***W2.** Idealized task/trace structure. The framework assumes a single “correct” index path with a known legal‑set policy; many real problems have multiple near‑equivalent paths and context‑dependent constraints. The parity/countdown focus limits external validity.*
>
> ---
> **A2:** Thank you for the insightful question. We would like to clarify that the “single correct index path with a known legal-set policy” appears only in Section 2.1, which is a **case study** illustrating why Theorem 1 naturally holds in a transformer-learned tree-reasoning setting. **Theorem 1 itself applies to a more general class of reasoning processes and does *not* require a unique index path.** For example, as an additional case study, Appendix E (Theorem 5) analyzes a realizable Markov Decision Process as the underlying reasoning process **without** the structural assumption you mentioned; curriculum benefits are proved there as well. Of course, this setting introduces its own modeling choices — for instance, it does not analyze transformer architectures, instead focusing on the associated linear–softmax policy class.
>
> To connect Theorem 1 to concrete computations and reasoning dynamics of 2-layer transformers (Yue et al., 2025), we introduce the 2S-ART model. Beyond chain-of-thought behavior for parity and Countdown as you noted, this framework is expressive enough to capture, per Line 223-226, Markov-chain (Kim et al., 2025), Induction-head–style associative recall (Nichani et al., 2025), and Causal-graph reasoning (Nichani et al., 2024).
>
> As with these prior theoretical analyses, our goal is to study a tractable yet expressive family of tasks, not to cover **every** realistic large-scale application. Parity, for instance, is a classical setup because of its non-linearly separable structure and long-standing value for understanding model capabilities. Our contribution is to provide a transformer-implementable tree-reasoning framework (2S-ART) under which the conditions of Theorem 1 arise naturally—thereby yielding exponential-vs-polynomial separations (e.g, the $d^{\Theta(K)}$ vs $\Theta(Kd^{2})$ complexity for parity).
>
> Looking ahead, an important direction for future work is to broaden the task/trace structure to capture more complex function classes learned by transformers beyond the 2S-ART framework, which we discussed in Section 3.

---

> ### Author Response · Authors · 2025-11-21
>
> ***W3.** Missing empirical calibration. The theory predicts polynomial scaling benefits; a small empirical study with real RLHF/RFT post‑training on diverse reasoning sets would strengthen the practical relevance.*
>
> **A3:** Thank you for the suggestion. We would first like to note that, similar to Foster et al. (2025)—who investigated realizable linear MDPs to reveal base-model coverage effects and proposed algorithms *without* conducting simulations—our work is intended as a **purely theoretical** study. Our goal is to highlight the *theoretical potential* of curriculum post-training, whose empirical benefits have already been widely observed in real LLMs (Parashar et al., 2025; Liu et al., 2025; Shi et al., 2025).
>
> Second, our exponential-to-polynomial separations are established to show what *could* be achieved **if** the sufficient $K$-th-root curriculum condition in Theorem 1 or Theorem 5, or the uniform-coverage condition in the transformer-based 2S-ART setting, are satisfied. These results are **not** claims that such conditions *necessarily* hold in real-world training pipelines; rather, they illustrate the theoretical advantages that such curricula can provide.
>
> ---
>
> **Summary**
>
> Thank you again for acknowledgement of our theoretical contribution. Should the reviewer have any further comments, we are happy to contrinue our productive exchange. Once again, we thank the reviewer's valuable time and feedback.
>
> ---
>
> ### **References**
>
> Foster et al (2025) Is a Good Foundation Necessary for Efficient Reinforcement Learning? The Computational Role of the Base Model in Exploration. COLT 2025
>
> Chen et al. (2025). The Coverage Principle: How Pre-Training Enables Post-Training
>
> Li et al. (2025) Generalist Reward Models: Found Inside Large Language Models
>
> Kim et al. (2025). Metastable Dynamics of Chain-of-Thought Reasoning: Provable Benefits of Search, RL, and Distillation. ICML 2025
>
> Nichani et al (2025). Understanding factual recall in transformers via associative memories. ICLR 2025
>
> Nichani et al (2024). How transformers learn causal structure with gradient descent. ICML 2024
>
> Parashar et al. (2025) Curriculum reinforcement learning from easy to hard tasks improves llm reasoning
>
> Liu et al. (2025)  UFT: Unifying supervised and reinforcement fine-tuning. NeurIPS 2025
>
> Shi et al. (2025). Efficient reinforcement finetuning via adaptive curriculum learning.

---

### Official Review · Reviewer_vyLJ · 2025-11-18

**Soundness:** 1
**Presentation:** 1
**Contribution:** 2
**Rating:** 2
**Confidence:** 3

**Summary:**

The paper proposes a theoretical framework to prove that curriculum post-training can avoid exponential sample-complexity bottlenecks in both RL fine-tuning and test-time scaling.

**Strengths:**

- The notion of 2S-ART provides a unifying abstraction that captures a broad class of reasoning behaviors.
- Theoretically characterizing when and why curriculum is more effective in post-training is an interesting question. This paper presents a general framework as well as analyzes a concrete synthetic task of parity.

**Weaknesses:**

- The writing quality hinders comprehension. Below are several points that substantially obstruct my understanding:
  - In Theorem 1, the notation $\mathcal C(\pi_{k'}^\star | \pi_{k}^\star )$ seems undefined. It is unclear what training algorithm is used, and why $\|\frac{\pi^\star }{\pi_{\mathrm{ref}}}\|_{\infty}$  is an appropriate proxy for difficulty.
  - For Theorem 3, I could not locate a full proof in the appendix. It would help to provide a clear pointer and a proof sketch in the main text.

- The proof of Theorem 1 becomes straightforward once the assumptions (1,2,3) are imposed. Thus, it is crucial to justify that these assumptions hold for the reasoning tasks studied, but the paper does not provide a clear validation. It is also unclear how the concrete examples of parity satisfy the assumptions and how they connect back to the general theorem.
- The theoretical novelty appears limited, and the results may not have substantial interest to the broader community.
  - For theorem 1, the core idea that $\mathcal C(\pi_{k'}^\star | \pi_{k}^\star )$  surves as a proxy of difficulty is studied in previous works [1].
  - The benefit of curriculum used in Theorem 3 looks similar to existing analyses of CoT training [2, 3]. It would be helpful if the authors could clarify on the difference in setup and proof techniques compared to these work.

**Questions:**

- Is the $\Omega(d^{ k^\star+1})$ in Theorem 3.1 derived via an statistical query-style argument? If so, how does this lower bound relate to the general complexity measure in Theorem 1?
- Why is $\|\frac{\pi^\star }{\pi_{\mathrm{ref}}}\|_{\infty}$ an appropriate proxy for difficulty? (If I understand correctly, [1] only justified it under linear softmax model parameterization.) It would be helpful if the authors can provide a formal statement establishing its relationship to statistical or computational complexity.

Please also refer to the weaknesses. I am open to adjusting my score based on the authors’ responses and the discussion with other reviewers.

---

> ### Author Response · Authors · 2025-11-21
>
> Thank you for your recognition for our theoretical framework as intersting and serving as a unifying various reasoning abstraction. We're encouraged for your interest in our work, and would like to address your concerns as below.
>
> ---
>
> ***W1 & Q2.** The writing quality hinders comprehension.*
>
> - *Why is $||\tfrac{\pi^\star}{\pi_{\text{ref}}}||_\infty$ an appropriate proxy for difficulty? (If I understand correctly, [1] only justified it under linear softmax model parameterization.)*
> - *In Theorem 1, the notation $\mathcal{C}(\pi_{k^{\prime}}^{\star}|\pi_{k}^\star)$ seems undefined. It is unclear what training algorithm is used.*
> - *It would be helpful if the authors can provide a formal statement establishing $||\tfrac{\pi^\star}{\pi_{\text{ref}}}||_\infty$'s relationship to statistical or computational complexity.*
> - *For Theorem 3, I could not locate a full proof in the appendix. It would help to provide a clear pointer and a proof sketch in the main text.*
>
> ---
>
> **A1:** Thank you for your feedback. We would like to clarify as below:
>
> - **Why $||\tfrac{\pi^\star}{\pi_{\text{ref}}}||_\infty$ serves as a proxy of difficulty?** As motivated in the introduction, one central question in curriculum theory is defining *difficulty*. In **pre-training** theory, difficulty are task-specific (e.g., variance of irrelevant features (Saglietti et. al, 2022), access to teacher checkpoints (Panigrahi et al., 2025), or sparsity of the parity Hamming weight (Abbe et al, 2023)). In **post-training** of LLM, however, empirical practice overwhelmingly uses *pass (success) rate* as the operative notion of difficulty (Parashar et al., 2025; Tong et al., 2025).
>
>   We observe that a principle formalization of notion is the **likelihood bound/coverage coefficient** $||\tfrac{\pi^\star}{\pi_{\text{ref}}}||_\infty$ (Brown et al., 2024; Snell et al., 2024; Wu et al., 2024). It quantifies how much the optimal policy concentrates probability mass where the base model places little mass; its inverse gives a **lower bound on the success rate**. Importantly, this formulation comes from the success-rate definition itself—**not** from assuming any neural parameterization.
> - **Role of Theorem 1 and the notation $\mathcal{C}(\pi_{k'}^\star \mid \pi_k^\star)$.**  Here, $\mathcal{C}(\pi_{k'}^\star \mid \pi_k^\star)$ denotes the learning complexity of reaching $\pi_{k'}^\star$ starting from $\pi_k^\star$. Per you noticed, Theorem 1 is intentionally stated at a **fully general** level: it does not commit to a particular learning algorithm, and therefore does *not* specify whether the complexity refers to sample complexity (finetuning) or reward-oracle complexity (test-time scaling). The theorem holds whenever Conditions Eq.(1)-(3) (or their relaxed versions above Eq.(4)) are satisfied.
> - **Relationship between $||\tfrac{\pi^\star}{\pi_{\text{ref}}}||\_\infty$ and $\mathcal{C}(\pi^{\star}|\pi_{\text{ref}}^\star)$.** In Theorem 1, this relationship is assumed through Condition Eq.(3); its concrete form depends on the specific learning algorithm. For example:
>
>   - **Rejection sampling** (Line 63–67): $\mathcal{C}(\pi^{\star}|\pi_{\text{ref}}^\star)=\Theta(||\tfrac{\pi^\star}{\pi_{\text{ref}}}||_\infty \log(\delta^{-1}))$ with $\delta$ as the confidence parameter for the $≥1-\delta$ confidence argument.
>   - **2S-ART model under the finetuning regime (Sec.2.1)**: $\mathcal{C}(\pi^{\star}|\pi_{\text{ref}}^\star)=\widetilde{\Theta}(||\tfrac{\pi^\star}{\pi_{\text{ref}}}||\_\infty^{2})$ and $\mathcal{C}(\pi_{k^{\prime}}^{\star}|\pi_{k}^\star)=\widetilde{\Theta}(||\tfrac{\pi_{k^{\prime}}^{\star}}{\pi_{k}^\star}||\_\infty^{2})$ with $\widetilde{\Theta}(\cdot)$ hides logarithmic factors. Here, we naturally have $||\tfrac{\pi_{k+1}^{\star}}{\pi_{k}^\star}||\_\infty = \Theta(d),||\tfrac{\pi_{k^{\prime}}^{\star}}{\pi_{k}^\star}||\_\infty = \Theta(d^{k'-k}),||\tfrac{\pi^\star}{\pi_{\text{ref}}}||\_\infty = \Theta(d^{k+1})$ in our setting, per discussed in Line 244-253. The complexity-difficulty relations are then explicitly discussed in Remark (Line 398-406) and Proof Outline (Line 427-429), where the inverse-square relationship is the classic information-theoretic limit.
> - **Proof Sketch of Theorem 3.** We respectfully emphasize that the proof sketch **is already included** the proof sketch in **Line 422-431**, and the appendix menu clearly indicates that the whole Appendix G.2 serves for Theorem 3. The core idea follows the standard one-step update framework (Cornacchia & Mossel, 2023; Kawata et al., 2025): the sample complexity is governed by how many samples are needed to separate the meaningful update direction from noisy directions (e.g., wrong secret index still yields ~1/2 success probability for parity task). In our setting the margin is in the order of $\Theta(d^{-(k+1)})$ vs $\Theta(d^{-1})$ for direct vs curriculum, respectively—yielding the separation stated in the theorem.

---

> ### Author Response · Authors · 2025-11-21
>
> ***W2.** The proof of Theorem 1 becomes straightforward once the assumptions (1,2,3) are imposed. Thus, it is crucial to justify that these assumptions hold for the reasoning tasks studied, but the paper does not provide a clear validation. It is also unclear how the concrete examples of parity satisfy the assumptions and how they connect back to the general theorem.*
>
> ---
> **A2:** Thank you for the question. We'd like to clarify the role of Theorem 1 and how the assumptions are validated in our setting.
>
>  Theorem 1 is intended to characterize a **sufficient condition** regime, under which curriculum post-training could shine out—namely Conditions Eq.(1-3) (or their relaxed versions above Eq. (4)). The purpose of the subsequent sections is precisely to **justify that these conditions naturally hold** in our autoregressive reasoning tree model under Definition 1-2, with parity reasoning appearing as a special case of this broader structure.
>
> In **Line 244-253**, we already explain why Conditions Eq.(1)-(2)are satisfied in this model. The remaining requirement—the relationship between $||\tfrac{\pi^\star}{\pi_{\text{ref}}}||\_\infty$ and $\mathcal{C}(\pi^{\star}|\pi_{\text{ref}}^\star)$ is established separately for:
>
> * **Finetuning (Theorem 3)**, where $\mathcal{C}$ is sample complexity, and
> * **Test-time scaling (Theorem 4)**, where $\mathcal{C}$ is reward-query and computational complexity.
>
> Both the theorem statements and their proof outlines make explicit the **squared dependence** relation ($\mathcal{C}(\pi^{\star}|\pi_{\text{ref}}^\star)=\widetilde{\Theta}(||\tfrac{\pi^\star}{\pi_{\text{ref}}}||\_\infty^{2})$ and $\mathcal{C}(\pi_{k^{\prime}}^{\star}|\pi_{k}^\star)=\widetilde{\Theta}(||\tfrac{\pi_{k^{\prime}}^{\star}}{\pi_{k}^\star}||\_\infty^{2})$), thereby completing the validation that Conditions Eq.(1)–(3) indeed hold in our reasoning-task setting.
>
> ---
>
> ***Q1.** Is the $\Omega(d^{k^{\star}+1})$ in Theorem 3.1 derived via an statistical query-style argument? If so, how does this lower bound relate to the general complexity measure in Theorem 1?*
>
> **A3:** Clearly **No**. We are not operating under the statistical query style where the theory statement should equip "$O(\varepsilon)$-approximate gradient oracle" argument. Instead, our analysis works directly with standard stochastic gradients, computed from data and CoT trajectories generated by the neural policy.
>
> Also, We respectfully note that our bound is $\widetilde{\Omega}(d^{2k^{\star}+2})$, **not** $\Omega(d^{k^{\star}+1})$. In Theorem 1, the general learning complexity $\mathcal{C}(\pi^\star \mid \pi_{\text{ref}}^\star)$ is intentionally left abstract, to be instantiated within a concrete learning setup. As clarified in **Lines 244–253**, under our autoregressive reasoning-tree model, Eq.(1)-(2) are satisfied such that $C^{\star}=d$ and $||\tfrac{\pi^\star}{\pi_{\text{ref}}}||\_\infty=\Theta(C^\star \cdot (C^\star)^{k^\star})=\Theta(d^{k^\star+1})$. Then Theorem 3 establishes the information-theoretic relationship $\mathcal{C}(\pi^{\star}|\pi_{\text{ref}}^\star)=\widetilde{\Theta}(||\tfrac{\pi^\star}{\pi_{\text{ref}}}||\_\infty^{2})$ as detailed in the Remark (Lines 398–406) and the Proof Outline (Lines 427–429).

---

> ### Author Response · Authors · 2025-11-21
>
> ***W3.*** *The theoretical novelty appears limited, and the results may not have substantial interest to the broader community.*
> - *For theorem 1, the core idea that $\mathcal{C}(\pi_{k^{\prime}}^{\star}|\pi_{k}^\star)$ surves as a proxy of difficulty is studied in previous works [1].*
> - *The benefit of curriculum used in Theorem 3 looks similar to existing analyses of CoT training [2, 3]. It would be helpful if the authors could clarify on the difference in setup and proof techniques compared to these work.*
>
> ---
> **A4:** We respecfally disagree with the reviewer's comments. Since the reviewer cites undefined references [1–3]", we respond based on the closest reasonable interpretation.
>
> * **Clarification on $\mathcal{C}(\pi_{k'}^\star \mid \pi_k^\star)$ vs. difficulty.**
>   First, $\mathcal{C}(\pi_{k'}^\star \mid \pi_k^\star)$ denotes the *learning complexity* required to reach $\pi_{k'}^\star$ from $\pi_k^\star$. It is **not** the proxy for difficulty. We suspect the reviewer intended to refer to $||\tfrac{\pi^\star}{\pi_{\text{ref}}}||\_\infty$.
>   While measuring distances between two policies using $||\tfrac{\pi^\star}{\pi_{\text{ref}}}||\_\infty$ is not new, to our knowledge we are the **first** to identify its *direct and natural connection* to **success/pass rate**, the operational difficulty measure used throughout LLM post-training (Parashar et al., 2025; Tong et al., 2025). This connection holds because $||\tfrac{\pi^\star}{\pi_{\text{ref}}}||\_\infty$ quantifies how heavily $\pi^\star$ concentrates probability on regions poorly covered by $\pi_{\text{ref}}$, and its inverse forms a lower bound on pass rate.
>   Although the reviewer does not specify "[1]", we suspect they refer to Foster et al. (2025). That work studies the relationship between $||\tfrac{\pi^\star}{\pi_{\text{ref}}}||\_\infty$ and reward oracle-query complexity plus computational complexity *under a linear-softmax parameterization* and their custom SpannerSampling algorithm, emphasizing the **necessity of coverage** (what we discuss in **A1**). They do **not** treat $||\tfrac{\pi^\star}{\pi_{\text{ref}}}||\_\infty$ as a difficulty notion for LLM post-training, nor do they connect this insight to curriculum strategies.
> * **Clarifying differences from works analogous to the reviewer’s "[2,3]".**
>   The reviewer does not specify "[2,3]", but the most plausible candidates are **Parashar et al. (2025)** and **Liu et al. (2025)**—both of which we already compare against in Appendix D, with further clarifications in Remarks 7 and 8 in Appendix E. The reviewer might also be referring to **Kim and Suzuki (2025)** or **Wen et al. (2025)**, which study CoT for parity in **pre-training**, not post-training.
>   Concretely:
>
>   * **Parashar et al. (2025)** analyze CoT through *Approximate Policy Iteration* with techniques designed for actor–critic updates (Chen & Maguluri, 2025). This RL-style setup is quite different from transformer tree-reasoning, and their exponentially decaying approximation error assumptions do not logically connet LLM post-training behavior.
>   * **Liu et al. (2025)** characterize curriculum benefit in terms of the **number of nodes explored in a search tree**, under a curriculum that **uniformly samples hint depth**. Their key mechanism is that every reasoning depth is explored with equal probability scale $\Theta(1/K)$, avoiding exponential branching. This is fundamentally different from our autoregressive transformer dynamics under **depth-increasing or hint-decreasing** curricula, which rely on the **stepwise moderate-difficulty condition** (A2), not uniform hint sampling.
>   * **Kim and Suzuki (2025)** and **Wen et al. (2025)** study **pre-training** for parity from scratch,focusing on comparing the training outcomes with and without CoT supervision. Their CoT supervision provides **per-step gradient signals** equally across depths (mirroring Liu et al.’s uniform-hint outcome), whereas our focus is **outcome reward-only step-wise curriculum post-training** for a broader class of function, which is structurally and analytically distinct. (See our A3 to Reviewer wQMa)
>
> ---

---

> ### Author Response · Authors · 2025-11-21
>
> **Summary**
>
> Thank you again for your recognition of our unifying framework. We hope our rebuttals could successfully resolved your misunderstandings. Should the reviewers have any further follow-up, we would be more than happy to continue our productive exchange. Once again, we thank the reviewers for their valuable time and comment.
>
> ---
>
> ### **References**
>
> Saglietti et. al (2022). An analytical theory of curriculum learning in teacher–student networks
>
> Parashar et al. (2025) Curriculum reinforcement learning from easy to hard tasks improves llm reasoning
>
> Abbe et al. (2023) Provable advantage of curriculum learning on parity targets with mixed inputs
>
> Tong et al. (2025) DART-Math: Difficulty-Aware Rejection Tuning for Mathematical Problem-Solving
>
> Brown et al (2024) Large language monkeys: Scaling inference compute with repeated sampling
>
> Snell et al (2024) Scaling LLM test-time compute optimally can be more effective than scaling model parameters
>
> Wu et al (2024) Making RL with preference-based feedback efficient via randomization
>
> Cornacchia & Mossel (2023) A mathematical model for curriculum learning
>
> Kawata et al. (2025) From Shortcut to Induction Head: How Data Diversity Shapes Algorithm Selection in Transformers
>
> Foster et al (2025) Is a Good Foundation Necessary for Efficient Reinforcement Learning? The Computational Role of the Base Model in Exploration
>
> Chen and Maguluri. (2025) An approximate policy iteration viewpoint of actor-critic algorithms
>
> Liu et al. (2025)  UFT: Unifying supervised and reinforcement fine-tuning
>
> Kim and Suzuki (2025) Transformers Provably Solve Parity Efficiently with Chain of Thought
>
> Wen et al. (2025) From Sparse Dependence to Sparse Attention: Unveiling How Chain-of-Thought Enhances Transformer Sample Efficiency

---

### Official Review · Reviewer_wQMa · 2025-11-20

**Soundness:** 1
**Presentation:** 1
**Contribution:** 2
**Rating:** 2
**Confidence:** 4

**Summary:**

This paper studies the optimization-based learnability of transformer in CoT reasoning tasks. The writing is difficult, with key notations missing or difficult to find, which obfuscated the main results and limits the value of the work.

The paper presents four main theorems.

Theorem 1 aims to show that curriculum improves divergence blow-up under certain bounded-divergence conditions, but the correctness is not checkable because the core definition of C(pi| pi’) is not given. (If I have to guess, it is something related to $||\pi_{k+1}/\pi_{k}||_{L^\infty}$ but then some of the assumed constants $C_{k,k'}$ might be in conflicts and need further clarification.)

Theorem 2 states that a random policy on the 2S-ART tasks defined in Definition 1 can be simulated by a transformer, if the FFN can simulate certain target operation. However, unless such policy can obtained by random initialization with high probability, the theorem neither give enough justification assuming such distribution as initial transformer policy, nor is it a suitable choice to show the hardness of the task in Theorem 3.


The results in Theorem 3 and 4 are more readable and relevant to the main message. The results appear legitimate and is as not difficult to understand as the previous ones. However the assumption of step-level oracle as curriculum is rather strong. In fact, it trivializes the problem and the result become very close to Kim & Suzuki, as both papers study learning CoT to solve parity tasks. The main difference is that Kim & Suzuki did not consider cases where irrelevant parity tokens exist. But the proof technique is largely the same, and the one-step GD analysis can be similarly derived.

**Strengths:**

Theorem 3 and 4 provide certain technical contribution, but needs a better way of presentation.

**Weaknesses:**

The writing of multiple sections are barely readable. Some definition of notations are missing or difficult to find, making the correctness of Theorem 1 not checkable. Even though the Theorem 3 and 4 can be of interest to theory folks, the paper still requires significant rewriting to be presentable. As for the value of Theorem 3 and 4, see above in the summary.

**Questions:**

None

---

> ### Author Response · Authors · 2025-11-21
>
> Thank you for your valuable time and feedback. We would like to address your concerns as below.
>
> ---
> ***W1.** Theorem 1 aims to show that curriculum improves divergence blow-up under certain bounded-divergence conditions, but the correctness is not checkable because the core definition of $\mathcal{C}(\pi_{k'}^\star \mid \pi_k^\star)$ is not given. (If I have to guess, it is something related to $||\pi_{k+1}/\pi_{k}||\_{L^\infty}$ but then some of the assumed constants $C_{k,k'}$ might be in conflicts and need further clarification.)*
>
> **A1:** Thank you for the comment. We clarify the misunderstandings concisely below.
>
> - **"Theorem 1 shows divergence blow-up." — This is not what we do.** Theorem 1 does *not* analyze divergence blow-up. It only states when curriculum reduces total learning complexity compared to direct training, assuming three simple conditions (Eqs. 1–3).
> - **Definition of $\mathcal{C}(\pi_{k'}^\star \mid \pi_k^\star)$.** It denotes the *learning complexity* of reaching $\pi_{k'}^\star$ starting from $\pi_k^\star$. Theorem 1 is deliberately **algorithm-agnostic**: $\mathcal{C}(\cdot)$ can be sample complexity (finetuning; Theorem 3) or reward-oracle complexity (test-time scaling; Theorem 4) in specific algorithm learning regime. In contrast, $C_{k'+1',k}:=||\tfrac{\pi_{k^{\prime}}^{\star}}{\pi_{k}^\star}||\_\infty$ measures *difficulty*, whose reciprocal bounds the success rate, the standard difficulty metric used in LLM post-training (Parashar et al., 2025; Tong et al., 2025). Theorem 1 is intended to characterize a **sufficient condition** regime, namely the Conditions Eqs.(1-3), stating that there exists a constant $C^{\star}>1$ such that
>
> $$
> (1) :C_{k+1,k}:=||\tfrac{\pi_{k+1}^{\star}}{\pi_{k}^\star}||\_\infty=\Theta(C^\star), \\ (2):C_{k'+1',k}:=||\tfrac{\pi_{k^{\prime}}^{\star}}{\pi_{k}^\star}||\_\infty=\Theta(\prod_{r=k}^{k^{\prime}-1} C_{r+1,r}), \\  (3):\mathcal{C}(\pi_{k'}^\star \mid \pi_k^\star)=\widetilde{\Theta}(C_{k^{\prime}, k}).
> $$
>
> **Just** relying on the three conditions, one immediately obtains that the direct-training cost
> $$
> \mathcal{C}\_{\mathrm{direct}}:=\mathcal{C}(\pi^{\star}\_{K}|\pi^{\star}\_{0})=\Theta\big(C\_{K,0}\big)=\Theta\big((C^{\star})^{K}\big)
> $$
> while curriculum step-wise post-training cost
> $$
> \mathcal{C}\_{\mathrm{curriculum}}:=\sum\_{k=0}^{K-1}\mathcal{C}(\pi^{\star}\_{k+1}\mid \pi^{\star}\_{k})=\Theta\big(\sum_{k=0}^{K-1} C\_{k+1,k}\big)=\Theta(KC^{\star})
> $$
>
> giving an exponential improvement.
>
> Their **relaxed versions of conditions** above Eq.(4)—namely (i) some steps admit polynomial exponents $\mathcal{C}(\pi^{\star}\_{k+1}| \pi^{\star}\_{k})=(C^{\star})^{p\_k}$ with $1\le p_k\ll K$; (ii) the direct complexity satisfies $\mathcal{C}\_{\mathrm{direct}}=\Theta((C^{\star})^{bK-c})$ for some $1\le b\ll K$ and $c\ll bK$, can generate similar curriculum benefit result: $\mathcal{C}\_{\mathrm{curriculum}}=O\big(K(C^{\star})^{p\_{\max}}\big)\ll\Theta \big((C^{\star})^{bK-c}\big)=\mathcal{C}\_{\mathrm{direct}}.$
>
> The **subsequent sections** of the paper are devoted to **showing that these conditions arise naturally** in our autoregressive reasoning-tree model (2S-ART), with parity reasoning being a special case of this broader structure.
>
> **Remark: Relationship between difficulty $||\tfrac{\pi^\star}{\pi_{\text{ref}}}||\_\infty$ and complexity $\mathcal{C}(\pi^{\star}|\pi_{\text{ref}}^\star)$.** In Theorem 1, this relationship is assumed via Condition Eq.(3); its *concrete* form depends on the specific learning algorithm. For example:
>
> - **Rejection sampling** (Line 63–67): $\mathcal{C}(\pi^{\star}|\pi_{\text{ref}}^\star)=\Theta(||\tfrac{\pi^\star}{\pi_{\text{ref}}}||\_\infty \log(\delta^{-1}))$ with $\delta$ as the confidence parameter.
> - **2S-ART model under the finetuning regime (Sec.2.1)**:In our 2S-ART setting Corollary 1 and Lines 244–253 show that, Eqs.(1-2) hold with $C^\star=d$:
>   $$
>   ||\tfrac{\pi_{k+1}^{\star}}{\pi_{k}^\star}||\_\infty = \Theta(d),||\tfrac{\pi_{k^{\prime}}^{\star}}{\pi_{k}^\star}||\_{\infty} = \Theta(d^{k'-k}),||\tfrac{\pi^\star}{\pi_{\text{ref}}}||\_\infty = \Theta(d^{k+1})
>   $$
>
>   Our Theorem 3 shows that
>
>   $$
>   \mathcal{C}(\pi^{\star}|\pi_{\text{ref}}^\star)=\widetilde{\Theta}(||\tfrac{\pi^\star}{\pi_{\text{ref}}}||\_\infty^{2}),\ \mathcal{C}(\pi_{k^{\prime}}^{\star}|\pi_{k}^\star)=\widetilde{\Theta}(||\tfrac{\pi_{k^{\prime}}^{\star}}{\pi_{k}^\star}||\_\infty^{2})
>   $$
>
>   with $\widetilde{\Theta}(\cdot)$ hides logarithmic factors. This complexity–difficulty relationships are explicitly discussed in the Remark (Lines 398–406) and the Proof Outline (Lines 427–429), where the inverse-square dependence is the classical information-theoretic limit.

---

> ### Author Response · Authors · 2025-11-21
>
> ***W2.** Theorem 2 states that a random policy on the 2S-ART tasks defined in Definition 1 can be simulated by a transformer, if the FFN can simulate certain target operation. However, unless such policy can obtained by random initialization with high probability, the theorem neither give enough justification assuming such distribution as initial transformer policy, nor is it a suitable choice to show the hardness of the task in Theorem 3.*
>
> ---
> **A2:** Thanks for your feedback. We clarify below. Given $\mathbf{x}=(x_1,...,x_d)$, $\mathrm{EOS}$ and their embeddings
> $$
> \mathbf{E}[x\_m]=[{\boldsymbol{\mu}^{x\_m}}^{\top},{\mathbf{p}\_m}^{\top}]^{\top}:=\mathbf{E}[z_{-m}],m\in[d],\mathbf{E}[z_0]:=\mathbf{E}[\mathrm{EOS}]=[{\boldsymbol{\mu}\_{\mathrm{EOS}}}^{\top},{\mathbf{p}\_{d+1}}^{\top}]^{\top}
> $$
> where $\boldsymbol{\mu}^{x_m},\boldsymbol{\mu}_{\mathrm{EOS}}$ are feature embeddings and $\mathbf{P}:=[\mathbf{p}\_1,...,\mathbf{p}\_{d+1}]$ are positional embeddings, our transformer architecture is
> $$
> \hat{\mathbf{p}}\_{l}=\sum\_{j=-d}^{l-2}\mathbf{V}(\mathbf{E}[z\_j]\operatorname{softmax}(\mathbf{E}[z\_j]^\top \mathbf{K}^\top \mathbf{Q}\mathbf{E}[z\_{l-1}])) \in \mathbb{R}^{d\_{\mathrm{X}}}
> $$
>
> $$
> \mathbf{p}\_{i\_{l}}\sim\operatorname{softmax}(\hat{\mathbf{p}}\_l^{\top}\mathbf{P}/\beta), \quad \text{The algorithm terminates with $\mathbf{E}[\mathrm{EOS}]$ if $\mathbf{p}\_{d+1}$ is sampled} $$
>
> $$\hat{\boldsymbol{\mu}}^{z\_{l}}=\operatorname{FFN}\_{l}({\boldsymbol{\mu}}^{x\_{i\_{l}}}, \hat{\boldsymbol{\mu}}^{z\_{l-1}}), \quad \mathbf{E}[z\_{l}]= [\hat{\boldsymbol{\mu}}^{z\_{l}}, \mathbf{p}\_{i\_{l}}]^\top \in \mathbb{R}^{d\_{\mathrm{E}}}.
> $$
>
> The transformer thus **first samples a positional embedding**, and then applies the FFN to perform the operation. Once the FFN can simulate the target operation (e.g., XOR for parity), the only remaining requirement is that the **attention mechanism uniformly selects the secret index** $i\_{l}$ from the legal set $\mathcal{I}\_l(z\_{<l})$ defined in Definition 1. To achieve this, we construct attention parameters:
> $$
> \mathbf{K}^{\top}\mathbf{Q}=\begin{pmatrix}
> \boldsymbol{0}\_{d_{\mathrm{X}}\times d\_{\mathrm{X}}} & \boldsymbol{0}\_{d\_{\mathrm{X}}\times (d\_{\mathrm{E}}-d\_{\mathrm{X}})} \\\\
> \boldsymbol{0}\_{(d\_{\mathrm{E}}-d\_{\mathrm{X}})\times d\_{\mathrm{X}}} & \mathbf{W}
> \end{pmatrix},\qquad
> \mathbf{V}= \begin{pmatrix} \boldsymbol{0} & \mathbf{I}_{(d\_{\mathrm{E}}-d\_{\mathrm{X}})} \end{pmatrix},
> $$
>
> so that for any token (x_m), $\mathbf{V}\mathbf{E}[x\_m] =\mathbf{p}\_m$ and the attention logits depend *only* on positional encodings: $\mathbf{E}[u]^{\top}\mathbf{K}^{\top}\mathbf{Q}\mathbf{E}[z\_{l-1}] = \mathbf{p}\_{k}^{\top}\mathbf{W}\mathbf{p}\_{d+1+l}$, $\mathbf{p}\_{d+1+l}:=\mathbf{p}\_{i\_{l}}$ if $u$ is at position $k$. For each $l$, we pick constants $c_l>0$ and impose a legality mask setting logits of all $k\notin\mathcal{I}\_l(z\_{<l})$ to $-\infty$, while enforcing $\mathbf{p}\_{k}^{\top}\mathbf{W}\mathbf{p}\_{d+1+l}=c_l,\forall k\in\mathcal{I}\_l(z_\{<l})$ (Eq.(12)). This ensures **uniform attention logits on all legal indices**, and therefore a **uniform sampling distribution**—precisely the class of base policies assumed in Definition 1.
>
> **Hardness of the task**. Below Corollary 1 (Lines 244–253), we explain how difficulty arises: when the base model exhibits the uniform-branching coverage behavior in Definition 1, deeper CoT requires more secret-index selections, and the success probability decays exponentially:$||\tfrac{\pi_{k+1}^{\star}}{\pi_{k}^\star}||\_\infty = \Theta(d),||\tfrac{\pi_{k^{\prime}}^{\star}}{\pi_{k}^\star}||\_\infty = \Theta(d^{k'-k}),||\tfrac{\pi^\star}{\pi_{\text{ref}}}||\_\infty = \Theta(d^{k+1})$. These match exactly the coverage-growth patterns in Eqs. (1)–(2), and are the quantities that determine the hardness in Theorem 3.

---

> ### Author Response · Authors · 2025-11-21
>
> ***W3.** The assumption of step-level oracle as curriculum is rather strong. In fact, it trivializes the problem and the result become very close to Kim & Suzuki...*
>
> **A3:** We respectfully disagree with the reviewer’s characterization and clarify the distinctions below.
>
> 1. **Assumption of step-level oracle.**
>    Our *hint-decreasing curricula* (Amani et al., 2025) and *depth-increasing curricula* (Parashar et al., 2025) are direct formalizations of widely used empirical practices, and our work aim to explain the potential benefit of such step-wise approach. Moreover, as discussed above Eq.(4), the step-wise difficulty assumption can be **further relaxed** to the form $\Theta(d^{p_l})$ with $p_{\max}=\max_{l\in[k]} p_l \ll k$. Such step-wise difficulty models are also standard in other curriculum-theoretic work, including Parashar et al. (2025) and Wang et al. (2025), where similar step-wise oracle assumptions are adopted. Liu et al. (2025), in contrast, do not assume a step-level oracle in theory; instead, they **uniformly sample hint depths**. Their analysis studied the **number of nodes explored in a search tree**, where each reasoning depth is visited with probability $\Theta(1/K)$. This avoids exponential budget and, from a gradient perspective, is ***closer*** in spirit to **Kim & Suzuki**, whose CoT supervision supplies **per-step gradient signals** equally across depths—mirroring Liu et al.’s uniform-hint mechanism. In contrast, our theory has a different lens: they complete that **2S-ART satisfies the sufficient conditions (Eqs. 1–3 and their relaxed versions)** required by Theorem 1. Particularly, Theorems 3–4 focus on establishing **complexity–difficulty alignment**: $\mathcal{C}(\pi^\star \mid \pi_{\text{ref}}^\star)=\widetilde{\Theta}(||\tfrac{\pi^\star}{\pi_{\text{ref}}}||\_\infty^{2}),\mathcal{C}(\pi_{k'}^\star \mid \pi_k^\star)=\widetilde{\Theta}(||\tfrac{\pi_{k'}^\star}{\pi_k^\star}||\_\infty^{2})$. Together with Corollary 1 (Lines 244–253) which shows that Eqs. (1–2) hold with $C^\star = d$: $||\tfrac{\pi_{k+1}^{\star}}{\pi_{k}^\star}||\_\infty = \Theta(d),||\tfrac{\pi_{k^{\prime}}^{\star}}{\pi_{k}^\star}||\_\infty = \Theta(d^{k'-k}),||\tfrac{\pi^\star}{\pi_{\text{ref}}}||\_\infty = \Theta(d^{k+1})$, this alignment directly leads to the **exponential curriculum advantage** predicted by Theorem 1.
> 2. **Conceptual difference.** Kim & Suzuki analyze **pretraining from scratch on a parity task**, comparing training dynamics ***with vs. without*** CoT supervision.In contrast, our work focus on showing that **step-wise curriculum post-training with suitable difficulty structure** is **theoretically cheaper** than direct learning, where we consider a base model equipped with the uniform capability to parity tasks of the same CoT depth. Our case study consider a **general autoregressive reasoning-tree (2S-ART) regime**, which strictly generalizes parity and includes Countdown reasoning (Fig. 1), Markov-chain reasoning (Kim et al., 2025), induction-head–style associative recall (Nichani et al., 2025), causal-graph reasoning (Nichani et al., 2024), and other structured multi-step reasoning operations per discussed in Line 218-226. Thus, parity is merely a **special case** of our broader structure.
> 3. **Differences in how “reasoning” is represented.** Kim & Suzuki use a **parity-specific hierarchical decomposition** over $k = 2^v$ parity, a neighbor-wise binary tree that is not suitable as an autoregressive reasoning outcome. In contrast, our framework uses the standard autoregressive update $z_l = z_{l-1} \oplus x_{i_l}$, which naturally models transformer-style step-by-step reasoning and applies uniformly to all 2S-ART tasks. Also, Kim & Suzuki’s transformer **does not model a sampling process**. In contrast, our model incorporates a **dictionary-based sampling mechanism**, where the attention module must **select the secret index** at each reasoning step while the FFN/MLP simulates the underlying operation. Even though both works involve learning to pick the correct index, the architectures, data flow, and training objectives are fundamentally different.
> 4. **Differences in learning-theoretic assumptions.** Kim & Suzuki:
>    - use a **statistical query argument**;
>    - assume an **$O(\varepsilon)$-approximate gradient oracle**;
>    - optimize an **average $L_2$ teacher-forced loss** over CoT-parity data.
>
> Our work:
>    - uses **standard stochastic gradients**, computed from model-generated trajectories;
>    - considers **outcome-only** CoT generation without internal supervision;
>    - evaluates correctness via pre-EOS validation rather than teacher-forced signals.
> 5. **Theorem 5 provides an additional, distinct case study.**
>    Theorem 5 (Appendix E) demonstrates the curriculum benefit under a **realizable linear softmax parameterization**, providing a second setting that is techinically similar (both validating Eqs (1-3)) to our proofs of Theorem 3-4, but clearly **far absent** from Kim & Suzuki.

---

> ### Author Response · Authors · 2025-11-21
>
> **Summary**
>
> We sincerely thank the reviewer for the time and thoughtful feedback. We understand the concerns regarding accessibility, and we appreciate the opportunity to clarify. In our rebuttal, we have addressed the reviewer’s misunderstandings about Theorem 1, provided the definitions and explicit conditions, and substantially improved the clarity of the relevant text. We hope these clarifications make the argument of Theorem 1 fully checkable.
>
> Regarding Theorems 3 and 4, we have carefully explained how our results differ in motivation, setting, and technical substance from prior work, and why they play a central role in supporting the general curriculum framework developed in Theorem 1. We also revised the original exposition to make these sections easier to follow.
>
> We hope that these revisions and clarifications help the reviewer better understand our contributions and resolve the earlier concerns. We appreciate the constructive feedback and are grateful for your valuable time.
>
> ---
>
> **References**
>
> Parashar et al. (2025) Curriculum reinforcement learning from easy to hard tasks improves llm reasoning
>
> Tong et al. (2025) DART-Math: Difficulty-Aware Rejection Tuning for Mathematical Problem-Solving
>
> Amani et al. (2025) Lower Bounds for Chain-of-Thought Reasoning in Hard-Attention Transformers
>
> Wang et al. (2025) Learning compositional functions with transformers from easy-to-hard data.
>
> Liu et al. (2025)  UFT: Unifying supervised and reinforcement fine-tuning
>
> Kim et al. (2025). Metastable Dynamics of Chain-of-Thought Reasoning: Provable Benefits of Search, RL, and Distillation
>
> Nichani et al (2025). Understanding factual recall in transformers via associative memories
>
> Nichani et al (2024). How transformers learn causal structure with gradient descent

---

### Author Response · Authors · 2025-11-21

Dear Reviewers,

Thank you for your valuable time in reviewing our submission. We are encouraged by the positive feedback, including comments noting that the paper is *well-motivated* (Reviewer PGDS), provides a *unifying abstraction* (Reviewer vyLJ; Reviewer PGDS), offers *precise modeling* (Reviewer GLjV), and has *potential impact* (Reviewer PGDS).

To further enhance accessibility, we have uploaded a revised PDF incorporating several clarity-focused improvements based on the reviewers’ helpful suggestions. All modifications are highlighted in **orange** for ease of inspection.

**Summary of updates:**

* **Introduction:** We added an explicit footnote explaining the likelihood-ratio bound, making the motivation and logical flow more accessible.
* **Notation Section:** We reorganized the exposition and added the explicit definitions to improve readability and reduce cognitive load.
* **Theorem 1:** To streamline the statement, we removed the separate definition of $C_{k',k}$ and instead directly use the likelihood-ratio form $||\frac{\pi'}{\pi}||\_{\infty}$. We also added a footnote clarifying the role of the learning complexity $\mathcal{C}(\pi' \mid \pi)$, emphasizing that Theorem 1 is intentionally stated in a general form to highlight the sufficient conditions under which curriculum benefits arise.

We hope these revisions improve the clarity and accessibility of the paper. We are grateful for the constructive feedback and would be happy to further refine the manuscript if reviewers have additional suggestions.

Thank you very much for your valuable time and attention.

*Authors of Submission 11516*

---

### Comment · Area_Chair_oUGc · 2025-11-28
**Please Reply to Author Concerns**

Dear Reviewers,

A friendly reminder to reply to author concerns.

Best,
AC

---

### Meta-Review · Area_Chair_6Vk4 · 2026-01-14

**Summary:**

According to the reviews, the concerns clustered into the following parts:
1. Accessibility and checkability of the core framework. Multiple reviewers reported that key objects and relationships were hard to locate or verify (e.g., what the abstract “learning complexity” ) denotes, and how it connects to “difficulty”). This matters because the paper’s main structural claim is formalized through Theorem 1’s sufficient condition template. The paper does explicitly frame “difficulty” via a coverage / likelihood-ratio proxy, and then states Theorem 1 in terms of curriculum step ratios and a complexity difficulty alignment condition.
2. Strength of assumptions used to instantiate the template. Even accepting Theorem 1 as a clean sufficient-condition statement, reviewers questioned whether the paper adequately justifies those conditions in settings of interest. Reviewers found this analytically convenient but potentially unrealistic for modern LLM next-token distributions.
3. Relationship to prior curriculum and CoT theory. Some negative reviews argued the main separations resemble prior CoT/curriculum analyses, and asked for clearer differentiation in setup (post-training vs train-from-scratch, outcome-only reward vs process reward). The paper’s intent is to unify curriculum post-training mechanisms under a general condition template and then validate those conditions in concrete regimes.
4. Lack of empirical calibration. One supportive reviewer still emphasized that, even for a theory paper, minimal empirical evidence or calibration would strengthen perceived relevance, while others accepted “pure theory” as a valid scope but noted that the current stylization limits broader community impact.

**Reviewer Concerns:**

Addressed by rebuttal / revision
1. Definition/role of “difficulty” and how it is used in the template.
2. How the abstract template is instantiated in 2S-ART.
3. Concrete learning-theoretic separations for finetuning/test-time scaling.

Still (partially) outstanding:
1. Realism of the 2S-ART base-model assumption
2. Breadth of tasks and trace structure. The “uniqueness” and “legal-set” structure is still idealized relative to many real reasoning problems with multiple near-equivalent trajectories.
3. Empirical calibration. No new empirical evidence is provided, if the venue prioritizes demonstrated practical relevance, this remains a weakness.
4. Perceived novelty to a broad ICLR audience.

**Reviewer Scores:**

1. Reviewer GLjV (score 6). Likely unchanged.
2. Reviewer PGDS (score 4). Their main concern appears to be an interpretation issue, which the rebuttal directly disputes, and the paper’s 2S-ART mechanism ties exponential depth dependence to the coverage/pass-rate structure rather than claiming universal exponential reasoning steps. Likely unchanged or +1.
3. Reviewer vyLJ (score 2).  Their review is dominated by “missing notation/proof pointer/validation of assumptions” concerns. Given the clarified linkage, they may still judge novelty/breadth as limited.
4. Reviewer wQMa (score 2). The “not checkable” complaint is plausibly alleviated once the paper/rebuttal clarifies definitions.
5. Reviewer 7hYf (score 0; identified as spurious by AC).

---

### Decision · Program_Chairs · 2026-01-26

Reject